# Neural similarity predicts whether strangers become friends

Yixuan Lisa Shen [1,6], Ryan Hyon[1,6], Thalia Wheatley[2,3], Adam M. Kleinbaum [4], Christopher L. Welker [2] & Carolyn Parkinson [1,5] ✉

What determines who becomes and stays friends? Many factors are linked to friendship, including physical proximity and interpersonal similarities. Recent work has leveraged neuroimaging to detect similarities among friends by capturing how people process the world around them. However, given the cross-sectional nature of past research, it is unknown if neural similarity precedes friendship or only emerges among friends following social connection. Here we show that similarities in neural responses to movie clips—acquired before participants met one another—predicted proximity in a friendship network eight months later (that is, participants with similar responses were more likely to be friends rather than several degrees of separation apart). We also examined changes in distances between participants in their shared social network, which resulted from the formation, persistence and dissolution of friendships, between two months and eight months after they met each other. Compared with people who drifted further apart, people who grew closer over this six-month period had been more neurally similar as strangers. In addition, analyses controlling for sociodemographic similarities showed that whereas these similarities appeared to drive the differences in pre-existing neural similarities between friends and dyads of a social distance of 3, they did not account for the more extensive links between pre-existing neural similarities and the tendency for people to grow closer together, rather than drift farther apart, over time. Thus, whereas some friendships may initially form due to circumstance and dissolve over time, later-emerging and longer-lasting friendships may be rooted in deeper interpersonal compatibilities that are indexed by pre-existing neural similarities. The localization of these results suggests that pre-existing similarities in how people interpret, attend to and emotionally respond to their surroundings are precursors of future friendship and increased social closeness.

[1]Department of Psychology, University of California, Los Angeles, Los Angeles, CA, USA. [2]Department of Psychological and Brain Sciences, Dartmouth College, Hanover, NH, USA. [3]Santa Fe Institute, Santa Fe, NM, USA. [4]Tuck School of Business, Dartmouth College, Hanover, NH, USA. [5]Brain Research Institute, University of California, Los Angeles, Los Angeles, CA, USA. [6]These authors contributed equally: Yixuan Lisa Shen, Ryan Hyon. ✉e-mail: cparkinson@ucla.edu

● fMRI study participants ● Other participants — Reported social ties

Two months — Six months

Time 1
fMRI study

Time 2
Social network survey

Time 3
Social network survey

**Fig. 1 | Data collection timeline and social network characterization.** A subset of students in a graduate programme (red nodes; $N = 41$) participated in the fMRI study before the start of their first year of the programme (Time 1). Most of the participants were scanned shortly after arriving on the campus (median number of days between arrival and fMRI scan, 3; modal number of days between arrival and fMRI scan, 1; see Methods for details). Two months later (Time 2), every student in the programme participated in a social network survey ($N = 288$; 100% response rate). The majority of these students ($N = 287$; 99.6% response rate)

completed the social network survey again six months after Time 2 (Time 3). Their social networks at Time 2 and at Time 3 were reconstructed using these data. The visualizations of the social networks at Time 2 and Time 3 were generated using the Large Graph Layout algorithm[76] in igraph[64]; note that there is some stochasticity in the determination of nodes' positions within graphs using this algorithm. Each node represents a study participant, and an edge connects two nodes if there is a mutually reported (that is, reciprocal) friendship.

Humans tend to cluster in their real-world social networks according to similarities in their demographic characteristics, such as age, gender and ethnicity, as well as in their behaviours and preferences[1–5]. This tendency to surround oneself with similar others, a phenomenon known as homophily, is a ubiquitous property of human social networks that has been observed across wide-ranging contexts, from modern industrialized societies to hunter-gatherer communities and online communities[1,6–8]. Such inter-individual similarities in demographic characteristics, behaviours and preferences may reflect similarities in how friends think about and respond to the world around them.

To investigate this possibility, prior work has attempted to relate inter-individual similarities in self-report-based measures of personality traits (for example, Big Five personality traits) to how close people are in their social network. However, this approach has yielded null or inconsistent results[9–18], suggesting that traditional approaches using personality surveys or questionnaires may not be sensitive to the types of inter-individual similarities that characterize friends. More recent work has demonstrated that inter-individual similarity in thoughts, feelings and beliefs about the world (that is, a "generalized shared reality") is predictive of social connection between individuals[19]. Consistent with this notion, a growing body of research has demonstrated that people closer together in their real-world social networks (that is, who are fewer degrees of separation from one another) are characterized by similarities in neuroanatomy, neural responses to watching naturalistic stimuli (that is, audiovisual movies) and patterns of neural activity at rest[20–23]. Although these studies demonstrate the utility of neuroimaging in probing the types of similarities that are shared among friends, their cross-sectional nature limits the types of inferences that can be made about the relationship between neural similarity and social network proximity. For example, the previously observed relationships between neural similarity and social network proximity may arise purely from people who were already friends becoming similar to one another due to social influence, contagion or exposure to common environments and experiences as a result of proximity in their social networks. Here we leveraged a longitudinal study design to test whether pre-existing similarities in neural responses to naturalistic

stimuli (prior to meeting one another) predicted future proximity in a real-world friendship network eight months later, and whether pre-existing neural similarity was also associated with increases in social closeness over time.

## Results

A subset of individuals in a graduate student cohort participated in our functional MRI (fMRI) study ($N_{fMRI} = 41$) before the beginning of their programme (Time 1). The majority of fMRI participants were scanned shortly after their arrival on campus to limit the opportunities that they had to interact with and befriend one another (Fig. 1 and Supplementary Fig. 1). During the fMRI study, the participants watched a series of naturalistic audiovisual stimuli (that is, movie clips). Movie clips spanning a wide range of styles (for example, documentary, comedy and debate) and topics (for example, food, science/technology, sports, environment and social events) were selected to fulfil several key criteria. Specifically, we sought clips that participants would be unlikely to have seen before, that would minimize mind-wandering by being sufficiently interesting to keep participants' thoughts and attention focused on the stimuli and that would elicit meaningful individual differences in neural responses—for example, because different people might have different temporal trajectories of emotional reactions or attentional allocation, interpret differently or attend to different aspects of the stimuli (see Methods for more details about stimulus selection). We then characterized the complete social network of this cohort two months later (after the beginning of their academic programme; Time 2) and again six months later (Time 3). For each unique pair of fMRI participants, pre-existing neural similarities were characterized at Time 1, and social distance was characterized at Times 2 and 3 (Fig. 2).

If pre-existing neural similarities truly reflect deep interpersonal compatibilities, we reasoned that such similarities may be particularly predictive of friendships after students have ample time to get to know one another and sort out with whom they are compatible. Our primary analyses therefore involved comparing different levels of social distance and the direction of changes in social distance to test whether Time 1 neural similarities predicted (1) friendship at our

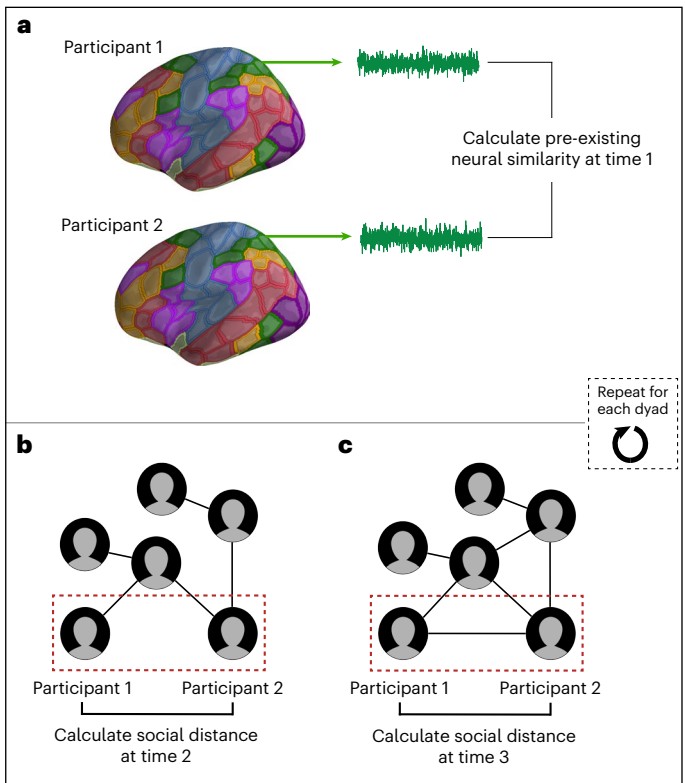

**Fig. 2 | Calculation of neural similarity and social distance. a**, Participants' data were resampled to standard space and parcellated into 200 brain regions using the Schaefer et al.[42] parcellation scheme. Each region is associated with a brain network from the Yeo et al.[43] seven-network parcellation, signified by different colours. For each region, the mean neural response was extracted at each time point during the fMRI scan, yielding a mean neural response time series. This was repeated for each participant. For a given pair of fMRI participants, the similarity of their neural response time series in each brain region at Time 2 was calculated using Pearson's *r*. **b**,**c**, For each pair of fMRI participants, the geodesic distance between them in the social network of their academic cohort was calculated at Time 2 (**b**) and at Time 3 (**c**).

latest time point (Time 3, eight months later) and (2) increases, rather than decreases, in social closeness over time (that is, in the six months between Times 2 and 3).

More specifically, in each brain region, we tested whether individuals who had a social distance of 1 (that is, friends) eight months into the academic programme had greater neural similarity before meeting one another than individuals characterized by greater social distances (that is, friends-of-friends and friends-of-friends-of-friends). Additionally, for each pair of fMRI participants, we determined the direction of their change in social distance over time (that is, from Time 2 to Time 3) and tested whether (and in which brain region(s)) individuals who grew closer in social ties over time had exhibited greater initial neural similarity before meeting one another than individuals who did not grow closer over time (Methods).

### Pre-existing neural similarity predicted future friendship

Mean neural similarities for dyads within each of the three social distance levels are visualized in Fig. 3. There were 93 dyads with a social distance of 1 (that is, friends), 544 dyads with a social distance of 2 (that is, friends-of-friends) and 183 dyads with a social distance of 3 (that is, friends-of-friends-of-friends). In this set of analyses, we compared pre-existing neural similarities between friends to those of dyads with different levels of social distance at Time 3. We first tested whether pre-existing neural similarities between individuals who came to be

characterized by a social distance of 1 (that is, who became friends) eight months later were greater than pre-existing neural similarities between individuals who did not. Specifically, this first analysis, which compared distance level 1 dyads against distance levels 2 and 3 combined, revealed no significant differences in pre-existing neural similarity between those groups (statistical significance was determined using permutation testing; Methods).

We followed up on these analyses, in which all non-friends were collapsed into a single category, with analogous analyses comparing pre-existing neural similarities between friends to those between individuals characterized by a social distance of 2 (that is, friends-of-friends) or 3 (that is, friends-of-friends-of-friends). Whereas no significant differences were observed between friends and friends-of-friends, friends exhibited significantly higher pre-existing neural similarity in a portion of the left orbitofrontal cortex (OFC; Fig. 4) than pairs of individuals characterized by a social distance of 3 (95% bootstrap confidence interval, [0.313, 0.998]; *P* < 0.001, corrected for false discovery rate (FDR)). A similar pattern of results was observed when we excluded the pairs of individuals who reported having interacted with each other (for example, briefly met at a happy hour) prior to the neuroimaging session (that is, excluding 18 dyads of the total 820 dyads; 95% bootstrap confidence interval, [0.314, 1.010]; Supplementary Fig. 5). For completeness, analogous analyses were conducted using social network data at Time 2 (two months into the academic programme; mean neural similarities for dyads within each of the three social distance levels at Time 2 are visualized in Supplementary Fig. 3), and there were no significant differences in pre-existing neural similarity between friends and any other non-friend groups (Supplementary Fig. 4).

In exploratory analyses, we investigated whether accounting for inter-individual similarities in ratings of enjoyment of and interest in the stimuli would fully account for and/or significantly diminish the difference in neural similarity between friends and pairs of individuals characterized by a social distance of 3 (Methods). In our first set of such analyses, which tested whether the difference in neural similarity between friends and pairs of individuals with a social distance of 3 could be fully accounted for by inter-individual similarities in ratings of enjoyment and interest in the stimuli, the effect in the OFC remained significant when we controlled for inter-individual similarities in ratings of enjoyment and interest (Supplementary Fig. 6). This suggests that similarity in neural responding in this brain region captured, at least in part, similarities in friends' responses to the stimuli that the interest and enjoyment ratings did not. Even so, while similar patterns of results were observed with and without controlling for similarities in these behavioural ratings, it is possible that the observed neural differences between groups of dyads might have been partly but significantly driven by participants' interest in and/or enjoyment of the stimuli. If this were the case, we would expect the magnitude of the neural similarity difference between groups to significantly decrease when controlling for such behavioural similarities. Thus, to inform our interpretation of the psychological meaning of the OFC result, we tested whether controlling for inter-individual similarities in ratings of either enjoyment or interest would significantly diminish the difference in neural similarity between friends and pairs of individuals characterized by a social distance of 3 (Methods). Neither controlling for inter-individual similarity in interest ratings nor controlling for inter-individual similarity in enjoyment ratings significantly decreased the difference in neural similarity between friends and individuals with a social distance of 3.

### Neural similarity as strangers predicted growing closer over time

For each dyad, we calculated their direction of change in social distance between Time 2 (that is, two months into their academic programme) and Time 3 (that is, eight months into their academic programme). Dyads were divided into three categories based on whether they grew closer over time (that is, decreased in social distance), grew apart over

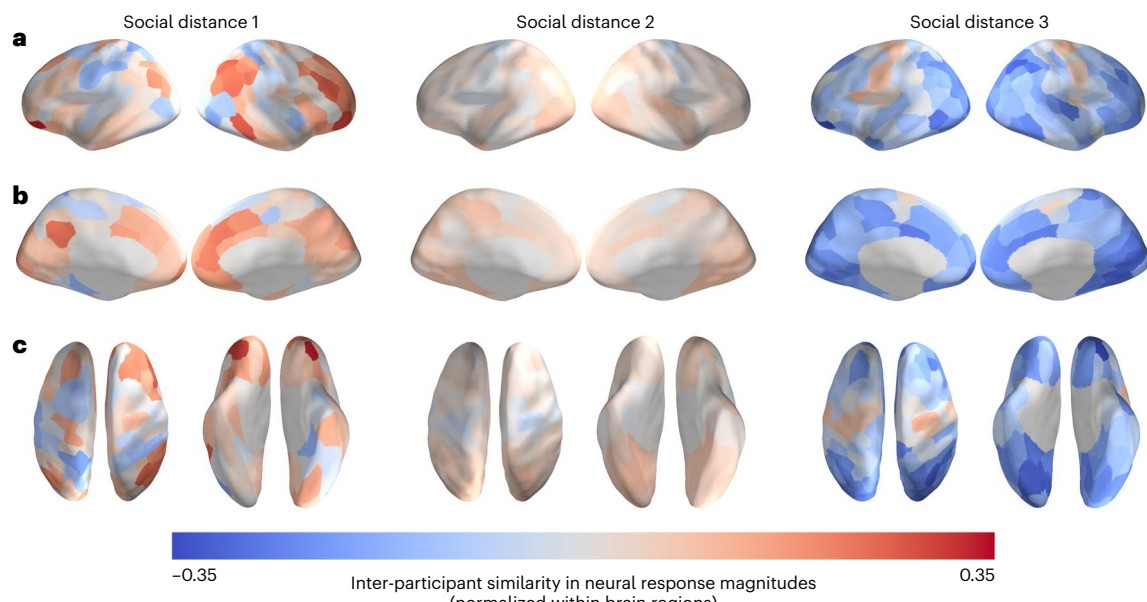

**Fig. 3 | Pre-existing neural similarities averaged within levels of social distance measured eight months later. a–c**, Social distance, ranging from 1 to 3, is the geodesic distance between a pair of individuals in the social network characterized by reciprocal friendship ties. The data are overlaid on a cortical surface model and are shown in lateral (**a**), medial (**b**) and dorsal and ventral (**c**) views. Because these images merely depict the relative mean similarities for each brain region for each group of dyads, rather than the results of statistical tests, no thresholding has been performed. Inter-participant neural similarities were normalized (that is, *z*-scored across dyads for each region; see Supplementary Fig. 2 for non-normalized ISCs), averaged within each social distance level and then projected onto an inflated model of the cortical surface. Warmer colours correspond to relatively similar neural responses for a given region, and cooler colours correspond to relatively dissimilar neural responses for a given region.

time (that is, increased in social distance) or remained at the same social distance. Mean neural similarities across dyads within each of the three change-in-social-distance categories are visualized in Fig. 5. We tested whether pre-existing neural similarities between individuals who grew closer over time were greater than those between individuals who did not grow closer over time (that is, who grew apart or whose social distance did not change). Statistical significance was determined using permutation testing (Methods). Relative to individuals who did not grow closer over the course of six months, people who grew closer did not exhibit significantly greater pre-existing neural similarity.

We followed up on these analyses, in which all pairs of participants that did not grow closer to each other over time were collapsed into a single category, with analogous analyses comparing pre-existing similarities between pairs of individuals who grew closer over time (that is, between Times 2 and 3) to those between pairs of individuals whose social distance did not change or who grew apart over time. Pairs of individuals who grew closer over time did not exhibit significantly higher pre-existing neural similarity than individuals whose social distance did not change over time. However, relative to individuals who grew apart over time, individuals who grew closer over time exhibited significantly higher pre-existing neural similarity in the bilateral thalamus, the left amygdala and 40 cortical regions spanning the visual cortex, ventral temporal cortex, occipitotemporal cortex, superior parietal cortex, angular gyrus, medial frontal cortex and lateral prefrontal cortex ($P < 0.05$, FDR-corrected; Fig. 6). A similar pattern of results was observed when we excluded pairs of individuals who reported having interacted with each other in any way prior to the neuroimaging session (Supplementary Fig. 9) and when we controlled for enjoyment and interest ratings (Supplementary Fig. 10). For each brain region that was statistically significant from the abovementioned analyses, the 95% confidence intervals of the normalized differences in intersubject correlations (ISCs) between the dyads that grew apart versus dyads that grew closer, as well as the associated *P* values, are reported in Supplementary Tables 2–5.

To further investigate why individuals who grew closer over time exhibited significantly higher pre-existing neural similarity than individuals who grew apart over time, we tested whether accounting for inter-individual similarities in ratings of enjoyment and interest in the stimuli would significantly diminish the difference in neural similarity between these two groups (Methods). This analysis was repeated for each of the brain regions in which we observed a significant difference in mean neural similarity (that is, the regions outlined in black in Fig. 6). We reasoned that even though similar patterns of results were obtained with and without statistically accounting for inter-participant similarities in enjoyment and interest ratings, it is possible that the observed group differences in ISCs in some brain regions might have been partially (and significantly) driven by what participants found enjoyable or interesting. We thus conducted this analysis with the goal of informing interpretations of the observed results. Controlling for inter-individual similarities in ratings of enjoyment of the stimuli significantly decreased the extent to which individuals who grew closer over time exhibited higher neural similarity in a portion of the right superior parietal cortex (difference in mean normalized neural similarity, 0.015; $P = 0.036$) than individuals who grew apart over time. However, this effect was not observed in other brain regions when we controlled for inter-individual similarities in ratings of interest in the stimuli.

### Sociodemographic variables partially explained these links

Given that homophily based on sociodemographic variables is widely observed in social networks, we ran analyses to test whether the relationships observed between pre-existing neural similarity and social network phenomena were accounted for by inter-participant similarities in sociodemographic variables (including inter-individual similarities in their age, gender, nationality, hometown size and location, undergraduate alma mater location and institution type, undergraduate major, and industry). When we controlled for these demographic variables, the significant effect in the left portion of the OFC shown in our first set of analyses, comparing pre-existing neural similarity between friends and dyads of a social distance of 3, was no longer significant after FDR correction across 214 brain regions (95% bootstrap confidence interval, [0.162, 0.909]; $P = 0.003$; FDR-corrected $P = 0.589$);

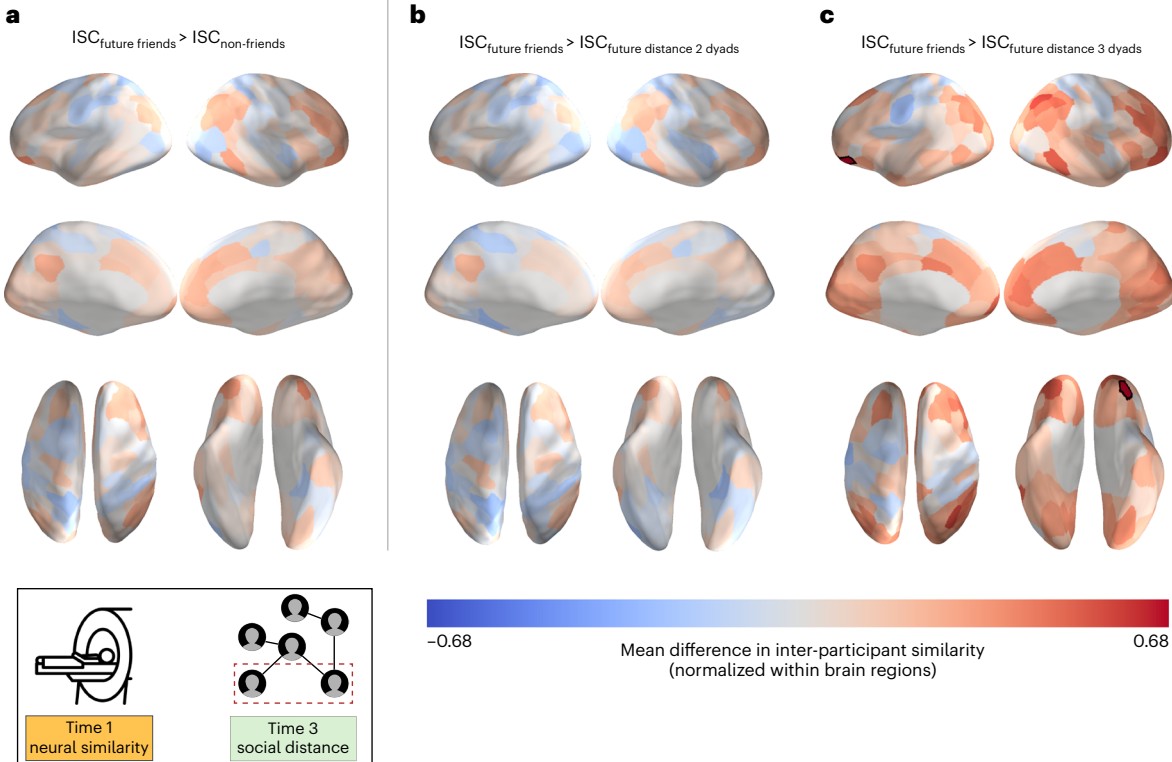

**Fig. 4 | People who became friends with each other showed greater pre-existing neural similarity than those who ended up three degrees of separation from each other eight months later. a–c,** The data are overlaid on a cortical surface model. Warmer colours correspond to relatively greater neural similarity for future friends than for other pairs of participants, and cooler colours correspond to relatively less neural similarity for future friends than for other pairs of participants. Individuals who became friends with each other showed greater pre-existing neural similarity in a portion of the left OFC (*P* < 0.001, FDR-corrected; statistical significance was determined using permutation testing) than individuals who ended up three degrees of separation from each other in the social network eight months after Time 1. The overlaid data are unthresholded; regions with significant differences, after FDR correction for multiple tests, are outlined in black.

---

however, we observed similar patterns of results with and without controlling for these demographic variables in the second set of analyses comparing the neural similarity of dyads that grew closer versus apart (Supplementary Fig. 8 and Supplementary Table 3). To further probe whether accounting for this extensive set of sociodemographic variables significantly reduced the difference in neural similarity between future friends and future friends-of-friends-of-friends, and if so, what sociodemographic variable(s) drove the significant reduction, we conducted analyses analogous to those that we performed to test whether accounting for inter-individual similarities in the extent to which participants found the stimuli enjoyable or interesting significantly diminished links between the neural and social network data. We found that controlling for these demographic variables resulted in a significant reduction of the neural similarity differences between friends and dyads of a social distance of 3 in the aforementioned OFC parcel (difference in mean normalized neural similarity, 0.129; *P* = 0.042), and that only gender similarity resulted in a significant reduction in the difference in neural similarity between groups (difference in mean normalized neural similarity, 0.019; *P* = 0.043).

## Discussion

Do pre-existing neural similarities predict future friendship? The current results provide evidence for neural homophily, such that relative to people who ended up far from one another in social ties, people who ended up becoming friends with one another demonstrated greater pre-existing neural similarity before meeting one another. Additionally, pre-existing neural similarity was particularly strongly linked to changes in inter-individual social distance between two and eight months after entering a new community. Such changes reflected the formation, persistence and dissolution of friendships in the community's social network.

Relative to individuals who had a social distance of 3 after having lived in a new community for eight months, individuals who became friends exhibited greater pre-existing neural similarity in a portion of the left OFC. Given the role of the OFC in processing subjective value[24], neural similarity in this brain region may reflect similarities in tastes and preferences (for example, similarities in what individuals find funny or otherwise appealing), which may become aligned across certain individuals as a function of sociodemographic similarities. Inter-individual similarity in sociodemographic variables (that is, age, gender, nationality, hometown size and location, undergraduate alma mater, undergraduate major, and industry) appeared to drive the differences in pre-existing neural similarity in the OFC that were observed between friends and friends-of-friends-of-friends. In particular, accounting for gender similarity significantly reduced differences in pre-existing neural similarity between friends and distance 3 dyads in this brain region. However, controlling for inter-individual similarity in the extent to which individuals were interested in the stimuli did not significantly diminish the difference in pre-existing neural similarity between friends and individuals characterized by a social distance of 3. Although neural similarity in this brain region may reflect similarity in the processing of subjective value, inter-individual similarity in self-reported ratings of preferences may not completely account for the subjective value processing that may be particularly aligned among friends.

We observed a similar pattern of results when examining the relationship between pre-existing neural similarities and changes in

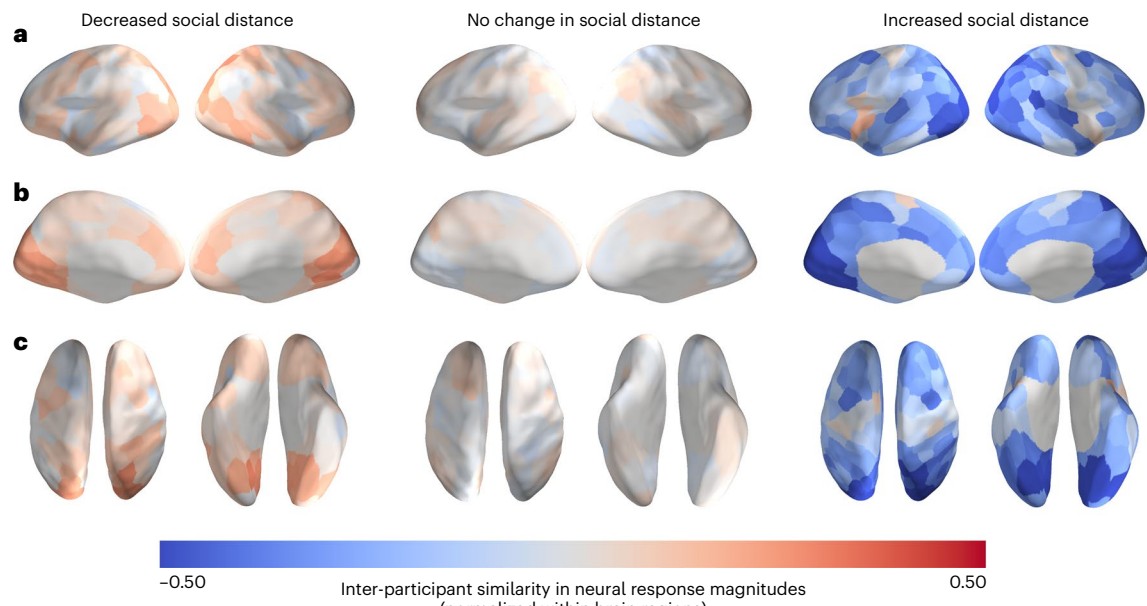

**Fig. 5 | Pre-existing inter-participant neural similarities, averaged within directions of change in social distance over time. a–c**, The data are overlaid on a cortical surface model and are shown in lateral (**a**), medial (**b**) and dorsal and ventral (**c**) views. Because these images merely depict the relative mean similarities for each brain region for each group of dyads, rather than the results of statistical tests, no thresholding has been performed. Inter-participant neural similarities were normalized (that is, *z*-scored across dyads for each region; see Supplementary Fig. 7 for non-normalized ISCs), averaged within directions of change in social distance and then projected onto an inflated model of the cortical surface. Warmer colours correspond to relatively similar neural responses for a given region, and cooler colours correspond to relatively dissimilar neural responses for a given region.

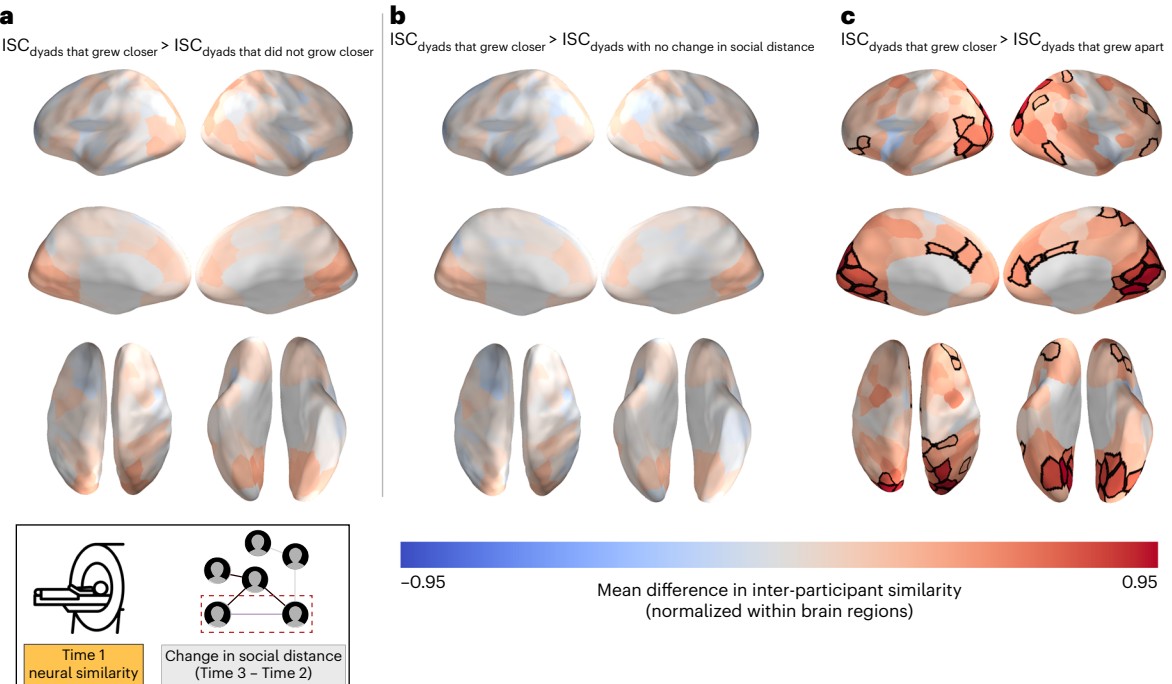

**Fig. 6 | Pairs of individuals who grew closer to each other over time showed greater pre-existing neural similarity than individuals who grew farther apart over time. a–c**, The data are overlaid on a cortical surface model. Warmer colours correspond to relatively greater mean neural similarity for a given brain region, and cooler colours correspond to relatively less mean neural similarity. For a given dyad, change in social distance over time was calculated by subtracting their social distance at Time 2 from their social distance at Time 3. The dyads were then placed into three categories depending on whether their social distance increased, decreased or remained the same. Pairs of individuals who grew closer to each other over time exhibited greater pre-existing neural similarity in 40 cortical regions spanning portions of the visual cortex, ventral temporal cortex, occipitotemporal cortex, superior parietal cortex, angular gyrus, medial frontal cortex and lateral prefrontal cortex than pairs of individuals who grew apart over time. The overlaid data are unthresholded; regions with significant differences ($P < 0.05$; statistical significance was determined using permutation testing; the *P* values for each brain region that showed a significant effect are reported in Supplementary Table 2), after FDR correction for multiple tests, are outlined in black.

inter-individual social distance over the course of six months. Relative to dyads who drifted apart over time, dyads who grew closer in social ties over time were characterized by exceptionally similar neural responses in portions of the bilateral OFC, and this difference in neural similarity was not significantly diminished when we controlled for similarities in individuals' self-reported enjoyment of or interest in the stimuli. Furthermore, dyads who grew closer in social ties over time also exhibited exceptionally similar neural responses in many brain regions spanning the default mode network (DMN), the frontoparietal control network (FPCN) and the dorsal attention network, suggesting that pre-existing neural similarities were particularly predictive of whether friendships formed, persisted or dissolved over time.

In particular, this relationship was observed in canonical DMN regions[25], such as the angular gyrus and medial prefrontal cortex, which have a well-established role in social cognitive functions such as mentalizing and perspective-taking[26–28]. Similarities in neural responding in these DMN regions have been associated with convergent interpretations of and affective responses to complex narratives[29–34]. Similar responding in these regions among people who later became friends may thus reflect pre-existing similarities in how such individuals deployed socio-cognitive processing to interpret the stimuli. Indeed, a recent framework has suggested that the DMN serves as an active "sense-making" network that integrates prior beliefs and long-term memories (for example, intrinsic information) with processing of extrinsic information to create models of real-life situations as they unfold in real time[35]. However, this relationship was not observed in all DMN regions (for example, the temporal poles and precuneus) but was observed in portions of the ventral temporal cortex and occipitotemporal cortex. Together with the angular gyrus, these regions, in part, have been suggested to comprise "gestalt cortex", in which inter-individual similarity in seemingly effortless subjective construal of complex narratives is associated with neural similarity[36]. Neural similarities in these regions may thus capture the creation of high-level meaning across individuals as they watch naturalistic stimuli.

People who grew closer over time also exhibited greater pre-existing neural similarities in several regions of the FPCN, such as the lateral prefrontal cortex, the cingulate cortex and a portion of the inferior temporal gyrus. Recent work has shown that a subsystem of the FPCN couples with the DMN across a multitude of tasks, such as those involving mentalizing, emotional processing, metacognitive awareness, prospective memory, stimulus-independent and abstract thinking, and future planning[37]. Given the well-established role of the FPCN in executive control[38], the FPCN may support top-down management of thought by constraining one's focus on contextually relevant material while simultaneously allowing for introspective thought[37,39], such as the processing of personally salient information or internally focused autobiographical planning[40,41]. In the context of watching naturalistic stimuli, the integration of internal trains of thought (for example, prior beliefs and memories) and processing of extrinsic stimuli is crucial for understanding complex narratives[35], suggesting that the FPCN may work in concert with the DMN to engage in sense-making of complex narratives. This is in line with prior work demonstrating that similarity in neural responding in both the DMN and FPCN is associated with similarity in subjective understanding of a complex narrative[30]. Such similarities in subjective construal of narratives may be particularly conducive to friendship formation, as they may reflect, more generally, alignment in how individuals make sense of the world around them.

We also observed pre-existing neural similarities in the superior parietal lobule and occipitotemporal cortex between friends that were significantly greater than those of people more distant from each other in their social network. Given that both of these regions are associated with the dorsal attention network[42,43], these neural similarities may reflect similarities in attentional allocation and in level of engagement. A growing body of research has demonstrated that attention modulates DMN activity during the processing of complex narratives[35,44,45]. In the context

of viewing naturalistic stimuli, pre-existing neural similarities in the dorsal attention network may reflect inter-individual similarities in how people allocate their attention and what people find particularly engaging, which may lead to convergent interpretations of complex stimuli.

Controlling for inter-individual similarities in ratings of enjoyment of and interest in the stimuli did not significantly diminish the exceptionally large difference in pre-existing neural similarity between individuals who grew closer over time and those who drifted apart. This was observed in nearly every brain region (besides the right superior parietal cortex, where accounting for enjoyment ratings decreased the difference in neural similarity between pairs of people who grew closer and those who drifted apart) in which individuals who grew closer over time exhibited exceptionally higher pre-existing neural similarity. This suggests that pre-existing neural similarities that predict whether people grow closer or drift apart go beyond what is captured by self-reported ratings of interest and enjoyment. A more extensive set of brain regions was implicated in our second set of analyses (which focused on changes in social distance over time) than in our first set of analyses (which compared friends to other dyads within the fMRI sample). This may be due in part to the fact that the second set of analyses are sensitive to changes in dyadic social distance over time that are driven by the formation or dissolution of friendships beyond the dyad, including those where one or both people involved in a friendship did not complete the fMRI study, whereas the first set of analyses may not be as sensitive to such phenomena. It is possible that future studies in which a larger proportion of the social network undergoes neuroimaging would reveal associations between friendship formation and neural similarity in a more extensive set of brain regions.

The current results suggest that individuals who process audio-visual movies in a similar fashion are exceptionally likely to become friends in the future and grow closer in social ties over time. Viewing naturalistic stimuli demands continuous integration of dynamic information streams as they unfold in real time, thereby evoking a wide range of socio-cognitive and emotional processes that characterize everyday mental life[46–48]. Thus, similarities in how people think about and respond to naturalistic stimuli may reflect more general similarities in how people think about and respond to the world around them. Taken together with the current results, this growing body of research is consistent with the possibility that these inter-individual similarities may facilitate social connection and the formation of social ties. Indeed, separate work has shown that a sense of "generalized shared reality" (that is, similarities in feelings, beliefs and concerns about the world in general) is linked to social connection and interpersonal liking[19] and thus may be conducive to friendship formation. The pre-existing neural similarities observed in the current study may also underpin similarities in sociobehavioural tendencies, which can lead individuals to participate in similar activities and frequent similar social spaces. Such sociobehavioural similarities can also foster the formation of affiliative ties by facilitating interpersonal communication and predictability[49–52].

Variability in these pre-existing similarities may have also facilitated the formation, persistence and dissolution of direct social ties between the two-month mark and the eight-month mark after individuals entered their new community. It is possible that some initial friendships at the two-month mark were formed based more on opportunity (for example, befriending others who conveniently sit nearby in class) than on deep interpersonal compatibility (for example, a sense of generalized shared reality)[19]. Subsequently, between the two-month mark and the eight-month mark, neural and social homophily processes may have had more time to unfold. As a result, given sufficient time, pairs of individuals may have formed and/or maintained friendships due to interpersonal compatibilities reflected in pre-existing neural and social similarity. In contrast, friendships born out of mere circumstance may have dissolved due to interpersonal incompatibilities, reflected in pre-existing neural and social dissimilarities, which became apparent only with the extended passage of time.

The current results are consistent with the possibility of neural homophily, such that future friends share pre-existing neural similarity before meeting one another. Future work would benefit from investigating the role of social influence, as it is likely that homophily and social influence processes interact in human social networks. For example, pre-existing similarities in how individuals think about and respond to the world around them may, in part, cause individuals to become friends. Over time, these individuals may undergo repeated, sustained social interactions that lead to further convergence in how they tend to speak, think, feel and behave[53–55], which may increase similarity in neural processing. This effect of social influence can percolate outward to more distal social ties, which allows for even indirectly connected individuals to influence and be influenced by each other[56]. Although the current study attempted to test for both neural homophily and social influence, planned collection of fMRI data two years after the participants entered their new community was prevented by the COVID-19 pandemic. Future work can leverage longitudinal fMRI to further examine such phenomena.

We also note that in the current study, fMRI data had to be collected within a very short time frame (that is, as soon as participants arrived on campus; as described in the Methods, the modal number of days between participants' arrival on campus and their participation in the fMRI study was 1, and the median number of days was 3) to minimize participants' opportunities to begin interacting with and befriending one another beforehand. This constraint, particularly in combination with other logistical limitations (for example, instrument availability during participants' initial arrival period), imposed a limit on sample size. Future studies would benefit from examining the current phenomena in larger samples.

Although the current sample included participants from a wide range of nationalities (Methods), the current results are reflective of a single context, and results may vary across cultures and contexts. Past cross-sectional work has linked neural[23] and behavioural[8] similarities to social network proximity across markedly different sociocultural contexts. However, it is important to note that the current evidence for neural homophily is constrained within a community of graduate students studying business at a private university in a single cultural context. Furthermore, participants self-selected to participate in the fMRI study, and it is possible that systematic differences exist between people who do or do not elect to participate in such a study. Future work examining these effects across diverse cultures and communities, ideally randomly sampling participants from within communities, will yield a more comprehensive understanding of how this phenomenon may differentially unfold across different social networks.

Relatedly, although we attempted to statistically account for inter-participant similarities in some sociodemographic variables in our analyses, it is likely that some unmeasured sociodemographic variables (particularly those related to similar cultural perspectives) causally contribute to both pre-existing neural similarities between participants and subsequent friendship formation. Our results are consistent with a causal model in which a wide array of sociodemographic variables and other factors, including but not limited to those that we were able to measure here (such as geographic proximity, race, nationality, gender, birth cohort, religion, linguistic background, educational background, parenting, political beliefs, genes, gene–environment interactions and so on), give rise to sets of experiences, expectations and social feedback that in turn give rise to particular ways of responding to the world, which are reflected in neural activity and, when similar, facilitate future friendship.

The current results nevertheless demonstrate that even without knowing or measuring all of these possibly innumerable sociodemographic variables and other factors, it is possible to measure one of their likely outcomes (neural similarity), which predicts future friendship. Although the question of the precise causes of such neural similarity may be intractable (given that there are probably numerous interactive causes, whose relative importance may vary across contexts), the current findings suggest that neural similarity predicts both future friendship formation and growing closer to particular people over time. We hope that future research can build on the current work to elucidate what factors contribute to similar ways of processing the world. Although we do not expect any single study to be able to fully address this question, it will be valuable for the field in general to better understand how sociocultural (and other) forces shape neural activity.

More broadly, despite the substantial degree of random assignment to which participants were subjected regarding with whom and where they lived, and regarding with whom they studied and took classes (Methods), we note that this study is inherently limited in its capacity to support decisive causal claims, given that it was an observational study. Thus, while the current findings are consistent with the causal model posited above, in which innumerable demographic and biological factors give rise to shared ways of responding to the world that are reflected in neural similarities and that in turn facilitate friendship, more research is needed to better understand how relationships among these phenomena come to be, to investigate potential moderating factors and to more directly test for such a causal model.

We also note that the current study's stimuli were selected and presented in a manner that we hypothesized would afford sensitivity to detect our predicted effects, as described in the Methods. It remains to be seen what kinds of stimuli and what characteristics of stimuli would evoke ISCs that would be most closely linked to future friendship formation. Similarly, it remains to be seen whether similar results would be observed if inter-participant neural similarities were instead computed on the basis of patterns of functional connectivity measured at rest or during naturalistic stimulation (which has recently been shown to be more predictive of behaviour than rest[57]). It would be valuable for future work to systematically address these questions, as such work has the potential to inform both basic and applied research on predictors of social connection.

A growing body of cross-sectional research integrating neuroimaging and social network analysis has demonstrated that individuals closer together in their real-world social network tend to share similarities in neuroanatomy[22], resting-state functional connectomes[23] and neural responses to watching naturalistic stimuli[20,21]. However, the cross-sectional nature of this work has limited the types of inferences that can be made about the causal relationship between neural similarity and social distance. The current research expands on this work and sheds light on the potential role of neural homophily in shaping human social structures over time.

## Methods

### Participants

**fMRI participants.** All data collection procedures were completed in accordance with the standards of the local ethical review board at Dartmouth College. A subset of 43 individuals from an incoming graduate student (MBA) cohort ($N_{cohort}$ = 288) at a private university in the USA participated in the neuroimaging study. The participants were scanned as soon as possible after their arrival on campus and self-selected to participate in the study after receiving a recruitment email that was sent to all incoming students in the cohort. The majority of fMRI participants were scanned shortly after arriving on campus (Supplementary Fig. 1). The modal number of days between participants' arrival on campus and their participation in the fMRI study was 1, and the median number of days was 3. Two of the fMRI participants were scanned more than one month after their arrival on campus; these participants had been enrolled in two different prior graduate programmes at the university but had not met one another at the time of the fMRI scan. The participants provided informed consent in accordance with the policies of the institution's ethical review board at Dartmouth College. Of the 43 participants, one participant did not complete the scan and was excluded from analysis, and one participant did not participate in the

social network survey administered at Time 3 (see 'Social network survey participants') and thus was excluded from analysis. Data from the resulting 41 participants (14 female) aged 25–34 (mean, 28.63; s.d., 2.15) were used for analysis. Of these participants, two participants had excess movement in only one of the four fMRI runs; thus, these scans were excluded from the analyses involving these two participants. Of these 41 participants, 38 were right-handed, and 3 were left-handed. Twenty-four participants self-identified their nationality as that of the USA; the remainder included five participants from India, two from Australia, two from Peru and one each from Argentina, Iran, Russia, Kyrgyzstan, the United Kingdom, China, Brazil and Canada. Fifteen participants identified as White/Non-Hispanic, two identified as Hispanic/Latino, one identified as Black/Non-Hispanic, one identified as Asian/Asian-American/Pacific-Islander, one identified as multi-racial and 21 chose to not indicate their ethnicity. The neuroimaging study was advertised to all students in the cohort via email. All students who were interested in participating and who passed a standard MRI safety screening participated in the scan. The participants received US$20 per hour for their participation.

**Social network survey participants.** Approximately two months (Time 2) and eight months (Time 3) after the fMRI scan (Time 1), the social network of the academic cohort was characterized (Fig. 1). Participants were from the same cohort of 288 (aged 24–39; mean age, 28.53; 129 females) first-year graduate students who had been recruited for the fMRI study at Time 1. As part of the graduate programme's aim to help students build a professional network, students were randomly assigned to study groups using stratified random sampling, such that students completed all of their coursework with the same group of randomly assigned classmates in each term. Additionally, nearly all students applied to on-campus housing and were randomly assigned to housing units on the basis of a lottery system. (That said, we note that even if housing units and roommates were initially randomly assigned, it is possible that they did not stay that way, given past research demonstrating both individual- and dyad-level predictors of roommate satisfaction and breakup, including various facets of interpersonal similarity[58,59].) All students in the first-year graduate cohort enrolled in a leadership course within the graduate programme, and they were invited to participate in the online social network survey as an optional part of their coursework on leadership. To encourage participation, students who completed the survey were provided with personalized, educational feedback about their social networks based on their survey responses, but they were reminded that participation was voluntary and would not affect their performance in the course (see more details about recruitment and data collection on p. 24 of the Supplementary Information). At Time 2, all students in the cohort completed an online social network survey (that is, 100% response rate). At Time 3, all students except one individual completed the survey (that is, 99.6% response rate).

### Experimental procedures

**fMRI acquisition.** The participants were scanned using a Siemens Prisma 3T scanner. Six functional runs were acquired using an echo-planar sequence (25 ms echo time; 2,000 ms repetition time; 3.0 mm × 3.0 mm × 3.0 mm resolution; 240 mm field of view; 40 interleaved transverse slices with no gap). A high-resolution T1-weighted (T1w) anatomical scan was also acquired for each participant (2.32 ms echo time; 2,300 ms repetition time; 240 mm field of view; 0.9 mm × 0.9 mm × 0.9 mm resolution).

**fMRI paradigm and stimuli.** Before the fMRI study began, the participants were told that they would be watching a set of videos while being scanned, which would vary in content, and that their experience in the study would be akin to watching television while someone else channel surfed. All participants saw the same clips in the same order (as if the clips comprised different scenes of a continuous movie), to avoid inducing response variability between participants related to differences in how the clips were presented. The stimuli consisted of 14 videos presented with sound over the course of six fMRI runs. The videos ranged in duration from 88 to 305 s (see Supplementary Table 1 for brief descriptions of the stimuli). The criteria used to select the stimuli are described in more detail in a previous manuscript that used the same stimuli in a different sample[20]. Briefly, efforts were made to select stimuli that (1) most participants would not have seen before, (2) would be engaging for participants and (3) would evoke diverging inferences and patterns of attentional allocation across viewers, and thus psychologically meaningful variability in neural responding (for example, because different people might attend to, emotionally react to and/or interpret them differently). We selected stimuli that spanned a wide range of styles and topics with the hope of evoking a wide range of cognitive and affective processing that resembles the diverse range of stimulation and associated mental phenomena that arises in everyday life. After the fMRI session, the participants filled out an online survey and provided ratings of the extent to which they enjoyed and were interested in each of the stimuli shown during the fMRI session. For each stimulus, a screenshot was shown, and the participants answered "How interesting did you find this video?" and "How much did you enjoy this video?" on a Likert scale from 1 to 5.

**Social network survey.** The participants followed an emailed link to the study website, where they responded to a survey designed to assess their position in the social network of students in their cohort of the academic programme. The survey question was adapted from Burt[60] and has been previously used in the modified form used here[11,20,60,61]. It read, "Consider the people with whom you like to spend your free time. Since you arrived at [institution name], who are the classmates you have been with most often for informal social activities, such as going out to lunch, dinner, drinks, films, visiting one another's homes, and so on?" A roster-based name generator was used to avoid inadequate or biased recall. Participants indicated the presence of a social tie with an individual by placing a checkmark next to his or her name. Participants could indicate any number of social ties and had no time limit for responding to this question.

We note that in this particular graduate programme, students lived close to each other in an isolated, rural area. Furthermore, they took classes together and frequently ate meals and socialized together. These characteristics of this graduate programme engendered an intense and immersive social experience.

### Data analysis

**fMRI preprocessing and parcellation.** fMRIPrep version 1.4.0 was used for anatomical and functional data preprocessing[62]. The T1w image was corrected for intensity non-uniformity with N4BiasField-Correction, distributed with ANTs version 2.1.0 and used as the T1w reference throughout the workflow. The T1w reference was then skull-stripped with a Nipype implementation of the antsBrainExtraction.sh workflow (from ANTs), using OASIS30ANT as the target template. Brain tissue segmentation of cerebrospinal fluid, white matter and grey matter was performed on the brain-extracted T1w using FSL FAST. Volume-based spatial normalization to MNI152Nlin2009cAsym standard space was performed through nonlinear registration with antsRegistration (ANTs version 2.1.0), using brain-extracted versions of both the T1w reference and the T1w template.

For each of the six blood-oxygenation-level-dependent (BOLD) runs per participant (across all tasks and sessions), the following preprocessing was performed. First, a reference volume and its skull-stripped version were generated using a custom methodology of fMRIPrep. The BOLD reference was then co-registered to the T1w reference using FSL FLIRT with the boundary-based registration cost-function. Co-registration was configured with nine degrees of

freedom to account for distortions remaining in the BOLD reference. Head-motion parameters with respect to the BOLD reference (transformation matrices and six corresponding rotation and translation parameters) were estimated before any spatiotemporal filtering using FSL MCFLIRT. Automatic removal of motion artefacts using independent component analysis (ICA-AROMA) was performed on the preprocessed BOLD run on MNI space time-series after the removal of non-steady-state volumes and spatial smoothing with an isotropic, Gaussian kernel of 6 mm full-width half-maximum. The confounding variables generated by fMRIPrep that were used as nuisance variables in the current study included global signals extracted from the cerebrospinal fluid, white matter and whole-brain masks; framewise displacement; three translational motion parameters; and three rotational motion parameters. These confounds were regressed out of the data for each preprocessed run. Temporal filtering was performed with a band-pass filter between 0.009 and 0.08 Hz.

The Schaefer et al.[42] parcellation scheme (resampled to MNI152Nlin2009cAsym standard space) with 200 parcels was used in the current study to define the 200 cortical regions of interest. Each parcel is associated with one of seven brain networks from the Yeo et al.[43] seven-network parcellation—the visual, somatomotor, dorsal attention, ventral attention, limbic, frontoparietal task control and default mode networks. The Harvard-Oxford subcortical atlas[63] was used to define 14 subcortical regions of interest—the bilateral nucleus accumbens, amygdala, putamen, caudate, thalamus, hippocampus and pallidum.

**Social network analysis.** The following steps were taken to characterize the social networks at Time 2 and at Time 3. Social network data were analysed using igraph (version 1.5.1)[64] in R (version 4.3.1)[65]. An unweighted graph consisting of only mutually reported social ties was used to estimate social distances between individuals. In other words, an undirected edge would connect two actors only if they had both nominated one another as friends, and no edge would connect the two actors if neither nominated the other as a friend or if only one actor nominated the other as their friend. Social distance was defined as the geodesic distance between people in the social network—that is, as the smallest number of intermediary social ties required to connect them in the network. Pairs of individuals who both named one another as friends were assigned a social distance of 1. Individuals were assigned a distance of 2 from one another if they had a mutually reported friendship with a shared friend but were not friends with one another, and so on.

At Time 2, of the 820 dyads of fMRI participants, 63 (7.68%) had a social distance of 1 (that is, they were friends), 436 (53.17%) had a social distance of 2 (that is, they were friends of one another's friends), 280 (34.15%) had a social distance of 3 and 41 (5.00%) had a social distance of 4. Dyads with a social distance of 4 were recoded as dyads with a social distance of 3 for the first set of analyses, given that similarities in neural responses in people four or more degrees of separation apart have previously been found to be highly variable and not significantly different from those of dyads two or three degrees of separation apart[20], although this recoding did not impact results. More generally, a large body of research demonstrates that relationships between interpersonal similarities in a variety of cognitive, emotional and behavioural phenomena (for example, risk perception, cooperation, smoking, depression, loneliness and happiness) and social network proximity disappear beyond three to four degrees of separation[66–73]. At Time 3, of the 820 dyads of fMRI participants, 93 (11.34%) had a social distance of 1, 544 (66.34%) had a social distance of 2 and 183 (22.32%) had a social distance of 3. For each social network, for descriptive purposes, we then calculated the average number of social ties across individuals, the median number of social ties and the reciprocity of the graph, which refers to the probability that person $i$ nominated person $j$ as a friend if person $j$ nominated person $i$ as a friend (mean social ties at Time 2, 72; median social ties at Time 2, 65; reciprocity at Time 2, 0.53; mean social ties at Time 3, 100;

median social ties at Time 3, 91; reciprocity at Time 3, 0.56). Here, the number of social ties refers to degree centrality, where incoming and outgoing ties are summed.

### Variation in neural similarity by future social distance
The following analysis was performed in each of the 200 cortical and 14 subcortical brain regions, and all analyses were conducted in Python (version 3.9.6). For each fMRI participant, all six fMRI scans were concatenated into a single time series (except for two participants who had only three scans that were concatenated due to excess movement in the excluded scans). For each fMRI participant, the neural response time series was spatially averaged across all voxels within a given region. For each unique pair of fMRI participants, we then calculated the Pearson correlation between their neural response time series, yielding a value reflecting neural similarity for a given region. For each region, disproportionately high and low neural similarity values (that is, outliers) were identified for values 1.5 times the interquartile range above the upper quartile (75th percentile) or below the lower quartile (25th percentile), and these outliers were reassigned values equal to the upper quartile plus 1.5 times the interquartile range or the lower quartile minus 1.5 times the interquartile range, respectively. For a given region, these neural similarity values were then Fisher-$z$-transformed and normalized across fMRI dyads using scikit-learn's StandardScaler function[74].

To test whether individuals characterized by a social distance of 1 (that is, friends) initially exhibited significantly higher neural similarity in a given brain region than did individuals characterized by a social distance of 2 or 3 (that is, non-friends), we first bucketed all individuals characterized by a social distance of 2 or 3 into a single category of social distance (that is, non-friends). The mean neural similarity was then calculated for each of the two resulting levels of social distance (that is, friends and non-friends). We then subtracted the mean neural similarity between pairs of individuals who did not become friends from the mean neural similarity between pairs of individuals who did become friends. To assess the statistical significance of this difference, we implemented the following permutation testing procedure. Neuroimaging data were randomly shuffled across fMRI participants, at the individual level, 1,000 times while holding all else in the dataset constant. Specifically, prior to calculating neural similarities, participant labels for the fMRI dataset were randomly shuffled (that is, each set of fMRI data was permuted at the individual level) to ensure that the permuted datasets would have equivalent dependency structures to the observed (non-permuted) data. In addition, we note that node-level permutations are often used for hypothesis testing on social network data across a wide range of contexts, particularly when probing potential links between individual nodes' features (here, neural responses) and their patterns of interactions or associations with one another, given that this method generates null models that break potential links between these variables while preserving the structure of the network[75]. In each permuted dataset, the inter-participant neural similarities for all parcels were computed for each dyad, and the mean neural similarity between individuals characterized by a social distance of 2 or 3 was again subtracted from the mean neural similarity between individuals characterized by a social distance of 1. This procedure yielded a null distribution of 1,000 permuted difference values. We then calculated the extent to which the true difference value was greater than the null distribution to generate a $P$ value. These $P$ values were then corrected for FDR across all 214 regions.

To get the 95% bootstrap confidence intervals, we generated 1,000 bootstrap samples of the fMRI participants at each time point. Then, for each unique dyad from the unique nodes in a given bootstrap sample, we retrieved the normalized ISCs of each brain region. Subsequently, we calculated the mean difference in ISCs between comparison groups (for example, dyads of social distance 1 versus 3) for each bootstrap sample. The 95% confidence intervals for each brain region were obtained by taking the 2.5% and 97.5% cut-offs of the bootstrap sampling distribution of mean differences in ISCs.

We repeated these analyses while controlling for inter-participant similarities in the sociodemographic variables of age, gender, nationality, hometown (in terms of both its location and its population), undergraduate alma mater (in terms of both whether they attended a private or public institution and the institution's location), undergraduate major, and the industry in which they had been employed before enrolling in the MBA programme. More specifically, to statistically account for demographic similarities that may be related to similarities in neural responding and/or social network proximity, we regressed out inter-participant similarities in the aforementioned sociodemographic variables from ISCs in each brain region before repeating the analyses described above.

This analytical procedure was repeated to test whether individuals characterized by a social distance of 1 exhibited significantly higher neural similarity in a given region than did individuals characterized by a social distance of 2. It was also repeated to test whether individuals characterized by a social distance of 1 exhibited significantly higher neural similarity in a given region than did individuals characterized by a social distance of 3.

In each brain region in which a significant difference in neural similarity between groups was observed, we then tested whether inter-individual similarity in self-reported ratings of enjoyment of the stimuli accounted for a significant portion of this difference. Even when similar patterns of results are observed with and without controlling for similarities in behavioural ratings, it is possible that the magnitude of the neural similarity difference between groups would significantly decrease when controlling for such behavioural similarities if the neural similarities are partially driven by them. These exploratory analyses thus aim to inform the psychological interpretation of the significant findings. First, for a given dyad, inter-individual similarity in enjoyment or interest ratings was measured by calculating the Euclidean distance between individuals' vectorized series of enjoyment ratings across the stimuli to yield a single similarity value. We then calculated the between-group difference in mean normalized neural similarity, and we separately calculated the between-group difference in mean normalized neural similarity after controlling for inter-individual similarities in enjoyment ratings. We then subtracted the latter difference value from the former difference value to yield a value capturing the extent to which between-group difference in neural similarity was diminished when controlling for inter-individual enjoyment similarity. To test whether this decrease in between-group difference in neural similarity was statistically significant, we implemented the following permutation testing procedure. Enjoyment ratings were randomly shuffled across fMRI participants 1,000 times while holding all else in the dataset constant. In each permuted dataset, the above-mentioned analytic procedure was repeated to generate a value capturing the decrease in between-group difference in neural similarity when controlling for enjoyment similarity. This procedure yielded a null distribution of 1,000 values capturing the decrease in between-group difference in neural similarity when controlling for enjoyment similarity. We then calculated the extent to which the true value was greater than this null distribution to generate a P value.

This analytical procedure was repeated using self-reported ratings of interest in the stimuli to test whether inter-individual similarity in self-reported ratings of interest in the stimuli accounted for significant portions of the differences in neural similarity between groups. This analytical procedure was also repeated using demographic variables to probe whether inter-individual similarity in demographic variables significantly accounted for the differences in pre-existing neural similarity between groups.

## Variation in neural similarity by the direction of future distance change

For each pair of fMRI participants, the change in social distance between Time 2 and Time 3 was calculated by subtracting their Time 2 social distance from their Time 3 social distance. Given that (1) extensive

changes in social distance from Time 2 to Time 3 were possible for only a small number of dyads (for example, only 5% of dyads were far enough apart in the social network at Time 2 to be able to decrease in social distance by three degrees of separation by Time 3) and (2) extraneous and situational factors may have an especially large impact on these early social distance values (as mentioned in the Discussion), we focused on the overall direction of social distance change (for example, whether participants within a given dyad decreased in social distance over time) rather than more granular variability in social distance changes.

To test whether individuals whose social distance decreased (that is, individuals who grew closer over time) between Times 2 and 3 exhibited greater pre-existing neural similarity in a given brain region than did individuals with no change in social distance between Times 2 and 3 or an increase in social distance (that is, individuals who grew apart over time) between Times 2 and 3, we first bucketed all individuals with no change or an increase in social distance into a single category of individuals who did not grow closer over time. In total, of the 820 dyads of fMRI participants, 279 were characterized by a decrease in social distance, 445 were characterized by no change in social distance and 96 were characterized by an increase in social distance. The following analysis was performed in each of the 200 cortical and 14 subcortical brain regions. The mean neural similarity was calculated for individuals who grew closer over time and separately for individuals who did not grow closer over time. We then subtracted the mean neural similarity between individuals who did not grow closer over time from the mean neural similarity between individuals who did grow closer over time. To assess the statistical significance of this difference, we implemented the following permutation testing procedure. Neuroimaging data were randomly shuffled across fMRI participants, at the individual level, 1,000 times while holding all else in the dataset constant. Specifically, prior to calculating neural similarities, participant labels for the fMRI dataset were randomly shuffled (that is, each set of fMRI data was permuted at the individual level) to ensure that the permuted datasets would have equivalent dependency structures to the observed (non-permuted) data. In each permuted dataset, the inter-participant neural similarities for all parcels were computed for each dyad, and the mean neural similarity between individuals who did not grow closer over time was again subtracted from the mean neural similarity between individuals who did grow closer over time. This procedure yielded a null distribution of 1,000 permuted difference values. We then calculated the extent to which the true difference value was greater than the null distribution to generate a P value. These P values were then corrected for FDR across all 214 regions. To get the 95% bootstrap confidence intervals, we generated 1,000 bootstrap samples of the fMRI participants at each time point. Then, for each unique dyad from the unique nodes in a given bootstrap sample, we retrieved the normalized ISCs of each brain region. Subsequently, we calculated the mean difference in ISCs between comparison groups (for example, dyads who grew closer over time versus those that did not) for each bootstrap sample. The 95% confidence intervals for each brain region were obtained by taking the 2.5% and 97.5% cut-offs of the bootstrap sampling distribution of mean differences in ISCs.

We repeated these analyses while controlling for inter-participant similarities in the demographic variables of age, gender, nationality, hometown (in terms of both its location and its population), undergraduate alma mater (in terms of both whether they attended a private or public institution and the institution's location), undergraduate major and the industry in which they had been employed before enrolling in the MBA programme. More specifically, to statistically account for sociodemographic similarities that may be related to similarities in neural responding and/or social network proximity, we regressed out inter-participant similarities in the aforementioned variables from ISCs in each brain region before repeating the analyses described above.

This analytical procedure was repeated to test whether individuals who grew closer over time exhibited greater pre-existing neural

similarity in a given region than did individuals whose social distance did not change over time or who grew apart over time.

In each brain region in which a significant difference in neural similarity between groups was observed, we then tested whether inter-individual similarity in self-reported ratings of enjoyment of or interest in the stimuli accounted for this significant difference using the same data-analytic and permutation-testing approaches described in the preceding section ('Variation in neural similarity by future social distance').

### Reporting summary

Further information on research design is available in the Nature Portfolio Reporting Summary linked to this article.

## Data availability

The processed data generated in this study (inter-participant correlation and social distance, as well as participants' ratings of enjoyment of and interest in the stimuli and handedness) are publicly available via GitHub at https://github.com/lisashen-syx/Neural-Similarity-Predicts-Whether-Strangers-Become-Friends.git. To ensure participant anonymity, demographic data can be made available upon request to accredited researchers after signing a non-disclosure agreement; some demographic data (regarding participants' hometowns, undergraduate institutions, nationalities and industries of prior employment) can be made available only at the dyadic, rather than individual, level. The repositories used for the two brain parcellation schemes are Schaefer (2018) 200 parcel parcellation (7 Networks)[42,43] (https://github.com/ThomasYeoLab/CBIG/tree/ee58db8b0e43c61468ba744abebcee08f601502b/stable_projects/brain_parcellation/Schaefer2018_LocalGlobal) and the Harvard-Oxford subcortical atlas[63] (https://identifiers.org/neurovault.collection:262). Source data are provided with this paper.

## Code availability

The code used for the analyses, including a demo using a small simulated dataset, is available via GitHub at https://github.com/lisashen-syx/Neural-Similarity-Predicts-Whether-Strangers-Become-Friends.git.

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

## Acknowledgements

This work was supported by startup funds from UCLA, National Institute of Mental Health grant no. R01MH128720 (C.P.) and National Science Foundation grant no. 1835239 (C.P.). The funders had no role in study design, data collection and analysis, decision to publish or preparation of the manuscript.

## Author contributions

C.P., T.W. and A.M.K. designed the research. R.H., C.L.W. and A.M.K. collected the data. Y.L.S. and R.H. analysed the data. Y.L.S., R.H. and C.P. wrote the manuscript with feedback from all authors.

## Competing interests

The authors declare no competing interests.

## Additional information

**Correspondence and requests for materials** should be addressed to Carolyn Parkinson.

# Reporting Summary

## Statistics

For all statistical analyses, confirm that the following items are present in the figure legend, table legend, main text, or Methods section.

| n/a | Confirmed | |
|---|---|---|
| ☐ | ☒ | The exact sample size ($n$) for each experimental group/condition, given as a discrete number and unit of measurement |
| ☐ | ☒ | A statement on whether measurements were taken from distinct samples or whether the same sample was measured repeatedly |
| ☐ | ☒ | The statistical test(s) used AND whether they are one- or two-sided *Only common tests should be described solely by name; describe more complex techniques in the Methods section.* |
| ☐ | ☒ | A description of all covariates tested |
| ☐ | ☒ | A description of any assumptions or corrections, such as tests of normality and adjustment for multiple comparisons |
| ☐ | ☒ | A full description of the statistical parameters including central tendency (e.g. means) or other basic estimates (e.g. regression coefficient) AND variation (e.g. standard deviation) or associated estimates of uncertainty (e.g. confidence intervals) |
| ☐ | ☒ | For null hypothesis testing, the test statistic (e.g. $F$, $t$, $r$) with confidence intervals, effect sizes, degrees of freedom and $P$ value noted *Give P values as exact values whenever suitable.* |
| ☒ | ☐ | For Bayesian analysis, information on the choice of priors and Markov chain Monte Carlo settings |
| ☐ | ☒ | For hierarchical and complex designs, identification of the appropriate level for tests and full reporting of outcomes |
| ☐ | ☒ | Estimates of effect sizes (e.g. Cohen's $d$, Pearson's $r$), indicating how they were calculated |

*Our web collection on statistics for biologists contains articles on many of the points above.*

## Software and code

Policy information about availability of computer code

| Data collection | Participants were scanned using a Siemens Prisma 3T scanner. |
|---|---|
| Data analysis | We used fMRIPrep version 1.4.0 to preprocess our fMRI data. We calculated inter-subject correlations (ISCs) using SciPy, normalized the ISCs using Scikit-learn 0.24.2 and permuted ISCs at subject-level in Python 3.9.6. The social network was characterized using igraph 1.5.1 in Python. Statistical tests (i.e., permutation tests) were conducted in Python 3.9.6. Custom codes developed for the study is available at https://github.com/lisashen-syx/Neural-Similarity-Predicts-Whether-Strangers-Become-Friends.git |

For manuscripts utilizing custom algorithms or software that are central to the research but not yet described in published literature, software must be made available to editors and reviewers. We strongly encourage code deposition in a community repository (e.g. GitHub). See the Nature Portfolio guidelines for submitting code & software for further information.

## Data

Policy information about availability of data

All manuscripts must include a data availability statement. This statement should provide the following information, where applicable:
- Accession codes, unique identifiers, or web links for publicly available datasets
- A description of any restrictions on data availability
- For clinical datasets or third party data, please ensure that the statement adheres to our policy

Source data for this study (specifically, the inter-subject correlation and social distance data that we generated in this study, as well as dyadic dissimilarities in

# Research involving human participants, their data, or biological material

Policy information about studies with <u>human participants or human data</u>. See also policy information about <u>sex, gender (identity/presentation), and sexual orientation</u> and <u>race, ethnicity and racism</u>.

| | |
|---|---|
| Reporting on sex and gender | As part of the social network survey, participants self-reported their gender. As our research question of interest does not concern friendship formation or social network based on gender, gender-based analyses were not performed as part of our primary analyses. Gender was included as a control variable in a set of analyses controlling for sociodemographic variables in which the inter-subject similarities in such variables (e.g., age, gender, nationality) were regressed out from ISCs in each brain region before repeating the methods used in the main analyses. Given that the analyses of our study were conducted on dyad-level data, we have only included a binary variable "gender_similarity" to indicate whether two people in the dyad self-report the same gender or not. Thus, individual-level data on gender was not shared. |
| Reporting on race, ethnicity, or other socially relevant groupings | Participants who completed the social network survey also self-reported their nationality, which is defined as the nation where they hold citizenship. We conducted additional analyses controlling for sociodemographic variables (e.g., age, gender, nationality, size and location of hometown, location and type of undergraduate institution, industry of employment) by regressing out inter-subject similarities in these variables from ISCs in each brain region before repeating the methods used in the main analyses. |
| Population characteristics | The research sample consists of a first-year graduate student (M.B.A) cohort at a private university in the United States (N = 288; 129 female; Mean age = 28.5 years). A subset of 43 individuals (14 female; Mean age= 28.6 ) self-selected to participate in the fMRI study after receiving a recruitment email that was sent to all incoming students in the cohort. Of the 43 participants, one participant did not complete the scan and was excluded from analysis and one participant did not participate in the social network survey administered at Time 3 (see Social network survey participants) and thus was excluded from analysis. Data from the resulting 41 participants (14 female) aged 25-34 (M = 28.63, SD = 2.15) were used for analysis. Of these 41 participants, 38 were right-handed and three were left-handed. Twenty-four participants self-identified their nationality as that of the United States, with five participants from India, two participants from Australia, two participants from Peru, and one participant each from Argentina, Iran, Russia, Kyrgyzstan, the United Kingdom, China, Brazil, and Canada. Fifteen participants identified as White/Non-Hispanic, two identified as Hispanic/Latino, one identified as Black/Non-Hispanic, one identified as Asian/Asian-American/Pacific-Islander, one identified as multi-racial, and 21 chose to not indicate their ethnicity. |
| Recruitment | Social network survey: All students in the first-year graduate cohort enrolled in a leadership course within the graduate program were invited to participate in the online social network survey as an optional part of their coursework on leadership. They were reminded that participation was voluntary and would not affect their performance in the course.<br>fMRI study: A subset of 43 individuals from an incoming graduate student (M.B.A.) cohort (Ncohort = 288) at a private university in the United States participated in the neuroimaging study. Participants self-selected to participate in the study after receiving a recruitment email that was sent to all incoming students in the cohort. |
| Ethics oversight | Institutional Review Board of Dartmouth College |

Note that full information on the approval of the study protocol must also be provided in the manuscript.

# Field-specific reporting

Please select the one below that is the best fit for your research. If you are not sure, read the appropriate sections before making your selection.

☐ Life sciences   ☒ Behavioural & social sciences   ☐ Ecological, evolutionary & environmental sciences

For a reference copy of the document with all sections, see <u>nature.com/documents/nr-reporting-summary-flat.pdf</u>

# Behavioural & social sciences study design

All studies must disclose on these points even when the disclosure is negative.

| | |
|---|---|
| Study description | This is a quantitative study involving tests for associations between fMRI and longitudinal social network data. |
| Research sample | The research sample consists of a first-year graduate student (M.B.A) cohort at a private university in the United States (N = 288; 129 female; Mean age = 28.5 years). A subset of 43 individuals (14 female; Mean age = 28.6) self-selected to participate in the fMRI study after receiving a recruitment email that was sent to all incoming students in the cohort. As the aim of the study involves investigating whether pre-existing neural similarities can predict future social network structure (e.g., whether two people befriend one another or grew closer over time) and thus characterizing the social network over time, we sought to recruit individuals who have opportunities to interact frequently and form connections with one another within a relatively bounded community, where variation in the relationships among individuals may be observed. Therefore, although this sample may not be representative of the general population at this age, deliberate choice to recruit from this graduate cohort is appropriate for research question of interest. |
| Sampling strategy | The sampling procedure is convenience sampling based on their willingness to participate. The neuroimaging study was advertised to all students in the cohort via email, and all students who were interested in participating and who passed a standard MRI safety |

screening participated in the scan. No sample size calculation was performed as we sought to recruit as many participants in the first-year graduate student cohort as possible to be able to characterize the social network accurately and to study our research question of interest.

| | |
|---|---|
| Data collection | The social network survey was administered online and participants completed it privately in the location of their choice. fMRI data was collected using a 3T Siemens Prisma scanner at a private university in the Untied States. Participants provided informed consent in accordance with the policies of the institution's ethical review board. No one was present besides the participant and the researchers, and the researchers were not blind to the study hypothesis during fMRI data collection, but they had no knowledge of the participants' social network data as these data were collected later. |
| Timing | ████████████████████████████████████████████ |
| Data exclusions | Of the 43 fMRI participants, one participant did not complete the scan and was excluded from analysis and one participant did not participate in the social network survey administered at Time 3 (see Social network survey participants) and thus was excluded from analysis. |
| Non-participation | Of the 43 fMRI participants, one participant did not complete the scan and one participant did not participate in the social network survey administered at Time 3. |
| Randomization | Participants were not allocated into experimental groups by the experimenters. That said, students were subject to substantial randomization with respect to their interaction opportunities with other members of the cohort. Specifically, students were randomly assigned to study groups using stratified random sampling, such that students completed all of their coursework with the same group of randomly assigned classmates in each term. Additionally, nearly all students applied to on-campus housing and were randomly assigned to housing units based on a lottery system |

# Reporting for specific materials, systems and methods

We require information from authors about some types of materials, experimental systems and methods used in many studies. Here, indicate whether each material, system or method listed is relevant to your study. If you are not sure if a list item applies to your research, read the appropriate section before selecting a response.

## Materials & experimental systems

| n/a | Involved in the study |
|---|---|
| ☒ | ☐ Antibodies |
| ☒ | ☐ Eukaryotic cell lines |
| ☒ | ☐ Palaeontology and archaeology |
| ☒ | ☐ Animals and other organisms |
| ☒ | ☐ Clinical data |
| ☒ | ☐ Dual use research of concern |
| ☒ | ☐ Plants |

## Methods

| n/a | Involved in the study |
|---|---|
| ☒ | ☐ ChIP-seq |
| ☒ | ☐ Flow cytometry |
| ☐ | ☒ MRI-based neuroimaging |

# Plants

| | |
|---|---|
| Seed stocks | N/A |
| Novel plant genotypes | N/A |
| Authentication | N/A |

# Magnetic resonance imaging

## Experimental design

| | |
|---|---|
| Design type | Naturalistic movie-watching task |
| Design specifications | Six functional runs (ranging from 104 TRs to 272 TRs) per session and subject |
| Behavioral performance measures | No behavioral performance was measured during the scans |

## Acquisition

| | |
|---|---|
| Imaging type(s) | structural and functional MRI |
| Field strength | 3T |
| Sequence & imaging parameters | Functional scans: spin echo; EPI sequence; 25 ms echo time (TE); 2000 ms repetition time (TR); 3.0 mm x 3.0 mm 3.0 mm resolution; 240 mm FOV; 40 interleaved transverse slices with no gap<br>T1-weighted anatomical scan: 2.32 ms TE; 2300 ms TR; 240 mm FOV; 0.9 mm x 0.9 mm x 0.9 mm resolution |
| Area of acquisition | A whole brain scan was used. |
| Diffusion MRI | ☐ Used  ☒ Not used |

## Preprocessing

| | |
|---|---|
| Preprocessing software | fMRIPrep version 1.4.0 |
| Normalization | The T1-weighted (T1w) image was corrected for intensity non-uniformity (INU) with N4BiasFieldCorrection, distributed with ANTs 2.1.0, and used as T1w-reference throughout the workflow. The T1w-reference was then skull-stripped with a Nipype implementation of the antsBrainExtraction.sh workflow (from ANTs), using OASIS30ANT as target template. Brain tissue segmentation of cerebrospinal fluid (CSF), white-matter (WM) and gray-matter (GM) was performed on the brain-extracted T1w using FSL FAST. Volume-based spatial normalization to MNI152NLin2009cAsym standard space was performed through nonlinear registration with antsRegistration (ANTs 2.1.0), using brain-extracted versions of both T1w reference and the T1w template. |
| Normalization template | MNI152NLin2009cAsym standard space (ICBM 152 Nonlinear Asymmetrical template version 2009c) |
| Noise and artifact removal | Automatic removal of motion artifacts using independent component analysis (ICA-AROMA) was performed on the preprocessed BOLD on MNI space time-series after removal of non-steady state volumes and spatial smoothing with an isotropic, Gaussian kernel of 6mm FWHM (full-width half-maximum). The confounding variables generated by fMRIPrep that were used as nuisance variables in the current study included global signals extracted from the CSF, WM, and whole-brain masks, framewise displacement, three translational motion parameters, and three rotational motion parameters. These confounds were regressed out of the data for each preprocessed run. Temporal filtering was performed with a band-pass filter between 0.009 and 0.08 Hz. |
| Volume censoring | N/A |

## Statistical modeling & inference

| | |
|---|---|
| Model type and settings | We calculated inter-subject correlations (ISCs) of fMRI time series of neural responses to capture similarity in neural responses across subjects during the processing of naturalistic stimuli. First, we extracted the mean-response time series across the video-viewing task from (1) each of the 200 cortical parcels in the 200-parcel version of the Schaefer et al. (2018) parcellation scheme and (2) 14 subcortical parcels in the Harvard-Oxford subcortical atlas, which resulted in a total of 214 brain regions across the whole brain. For each unique pair of participants (i.e., dyads) in our fMRI sample, we computed the Pearson correlation between the dyad members' time series of neural responses for each cortical parcel. This yields one correlation coefficient per unique dyad for each brain parcel. We then used a permutation testing procedure to test whether the mean neural similarity differed between levels of social distance at Time 3 and direction of change in social distance from Time 2 to Time 3. |
| Effect(s) tested | We tested (1) if pre-existing neural similarity at Time 1 differed between levels of social distance at Time 3 and (2) if pre-existing neural similarity at Time 1 significantly differed as a function of the direction of change in social distance between Time 2 and Time 3. |
| Specify type of analysis: | ☒ Whole brain  ☐ ROI-based  ☐ Both |
| Statistic type for inference<br><br>(See Eklund et al. 2016) | As described in the "Model type and settings" field above, our analyses used responses within each of 214 anatomically-defined brain regions, and thus are not impacted by the concerns that the Eklund et al. (2016) paper raised regarding inflated false-positive rates in fMRI inferences for spatial extent.<br><br>We used False-Discovery Rate (FDR) correction to correct for multiple comparisons across brain regions. |
| Correction | We used FDR correction for multiple comparisons. |

## Models & analysis

| n/a | Involved in the study |
|---|---|
| ☒ | ☐ Functional and/or effective connectivity |
| ☒ | ☐ Graph analysis |
| ☒ | ☐ Multivariate modeling or predictive analysis |

