## [Peer Review File · Nature Human Behaviour]

Neural Similarity Predicts Whether Strangers Become Friends

Corresponding Author: Dr Carolyn Parkinson

This manuscript has been previously reviewed at another journal. This document only contains information relating to versions considered at Nature Human Behaviour.

A version of this paper was originally rejected for publication by Nature Human Behaviour, however that decision was reconsidered after appeal by the authors.

Version 0:

Decision Letter:

20th November 2023

Dear Dr Parkinson,

Thank you once again for your manuscript, entitled "Neural Similarity Induces Friendship", and for your patience during the peer review process.

Your Article has now been evaluated by 3 referees. You will see from their comments copied below that, although they find your work of considerable potential interest, they have raised quite substantial concerns. In light of these comments, we cannot accept the manuscript for publication, but would be interested in considering a revised version if you are willing and able to fully address reviewer and editorial concerns.

We hope you will find the referees' comments useful as you decide how to proceed. If you wish to submit a substantially revised manuscript, please bear in mind that we will be reluctant to approach the referees again in the absence of major revisions. We are committed to providing a fair and constructive peer-review process. Do not hesitate to contact us if there are specific requests from the reviewers that you believe are technically impossible or unlikely to yield a meaningful outcome.

To guide the scope of the revisions, the editors discuss the referee reports in detail within the team, including with the chief editor, with a view to (1) identifying key priorities that should be addressed in revision and (2) overruling referee requests that are deemed beyond the scope of the current study. We hope that you will find the prioritized set of referee points to be useful when revising your study. Please do not hesitate to get in touch if you would like to discuss these issues further.

In particular,

- 1) Both Reviewer #1 and Reviewer #3 express concerns with statistical tests on non-independent samples. Please address those in full, possibly by running multilevel linear models as suggested by Reviewer #1.
- 2) Reviewer #3 requests additional analyses to increase confidence in the robustness of the results. Please perform those.
- 3) Please clarify how participant sampling and recruitment was performed in detail, as both Reviewer #2 and #3 have concerns about this aspect of the research. In particular, Reviewer #2 has a potentially serious concern about ethics. Please confirm that participation was truly voluntary, or elaborate on the precise conditions of participant recruitment.
- 4) Please expand on what insights the longitudinal design used here can provide above and beyond your 2018 Nature Communications study. We ask that you clarify the relationship between the two pieces of work, including the potential overlap between the two datasets and why the sample size differs.
- 5) Please remove causal claims while working to address Reviewer #2's concerns regarding threats to potential causal inference.

Finally, your revised manuscript must comply fully with our editorial policies and formatting requirements. Failure to do so will result in your manuscript being returned to you, which will delay its consideration. To assist you in this process, I have attached a checklist that lists all of our requirements. If you have any questions about any of our policies or formatting,

please don't hesitate to contact me.

If you wish to submit a suitably revised manuscript, we would hope to receive it within 4 months. I would be grateful if you could contact us as soon as possible if you foresee difficulties with meeting this target resubmission date.

- Include a "Response to the editors and reviewers" document detailing, point-by-point, how you addressed each editor and referee comment. If no action was taken to address a point, you must provide a compelling argument. When formatting this document, please respond to each reviewer comment individually, including the full text of the reviewer comment verbatim followed by your response to the individual point. This response will be used by the editors to evaluate your revision and sent back to the reviewers along with the revised manuscript.
- Highlight all changes made to your manuscript or provide us with a version that tracks changes.

Link Redacted

Thank you for the opportunity to review your work. Please do not hesitate to contact me if you have any questions or would like to discuss the required revisions further.

Sincerely,

Nature Human Behaviour

Reviewer expertise:

Reviewer #1: interpersonal relationships, social cognition

Reviewer #2: social networks, homophily, social cognition

Reviewer #3: neural synchronization, fMRI, naturalistic stimuli

REVIEWER COMMENTS:

Reviewer #1:

Remarks to the Author:

Theoretical Contribution

I believe this paper makes a groundbreaking theoretical contribution to our understanding of what makes humans connect and deepen their relationships. Prior work in this area has been largely cross-sectional, and for the first time this work allows us to establish causality by measuring neural similarity before friendship formation and measuring decreases in social distance over time. I find it impressive that this pattern of results held controlling for demographic similarity and for similarity in interest and enjoyment of the stimuli (suggesting that these effects are not driven by demographic similarity or agreements about valence, but rather deeper interpretations of the stimuli). These results greatly advance not only social neuroscience, social psychology, and interpersonal science, but our understanding of human behavior as a whole. I believe the paper could be improved by running more meaningful analyses (multilevel linear models) and by more clearly and consistently referring to the methods in the rest of the paper.

Analyses

My greatest concern is that the authors analysed their data using permutation testing procedures. Why did the authors not run multilevel linear regressions? Running MLM's (with binary predictors and continuous outcomes) would allow the authors to account for the interdependence of the datapoints within individuals and within networks, etc., and would produce effect sizes and confidence intervals rather than just p-values. If the authors have a reason for conducting permutation testing rather than regression models, this should be explained and references should be provided to support this decision (and the decision not to run multi-level models/ account for the interdependence of datapoints should also be explained), even if in the supplement.

More broadly, the authors only report p-values in the results, which does not give us an idea of effect sizes. Even if linear models are not used, I would recommend adding effect sizes, 95% confidence intervals, and degrees of freedom throughout the results section.

Methods

The paper would benefit from greater clarity in the way the methods (specifically the social distance) are discussed in the paper itself. The methods are clear in the methods section, but they are described inconsistently in the rest of the paper, especially the results. For example, social distance is often described as if it were a continuous variable (i.e., "distance in social network") when it is actually a series of binary comparisons (e.g., whether or not they became friends (i.e., 1 vs levels 2 or 3), or 1 vs. 3, etc.) I don't see an issue with using binary comparisons, but I think it would be clearer to bring the description of the variable in line with the nature of the variable.

How were non-reciprocal friendships dealt with? If reciprocity was .53-.56, this suggests that there were a substantial number of non-reciprocal friendship dyads where one person designated the other as a friend but not vice versa. The authors state "Pairs of individuals who both named one another as friends were assigned a social distance of one. Individuals would be assigned a distance of two from one another if they had a mutually reported friendship with a shared friend, but were not friends with one another, and so on." How were dyads coded if the selection was not mutual?

"Dyads were divided into three categories depending on if they grew closer over time (i.e., decrease in social distance), grew apart over time (i.e., increase in social distance), or remained at the same social distance." - why was distance not examined continuously? I don't mean to imply that this approach would be superior, but merely that the authors should explain their decision-making.

Discussion

I found the discussion thoughtful. It did leave me wondering why the authors found that only the left OFC predicted friendship formation, whereas similarity in so many other brain regions predicted shrinking social distance over time. I love the interpretation of the latter finding, but why would we see that so many brain regions would predict increased/decreased social distance over time, but not initial friendship formation (why only the left OFC for the first finding)? Is this related to the conclusion that "friendships born out of mere circumstance may have dissolved due to interpersonal incompatibilities, reflected in pre-existing neural dissimilarities, which only became apparent with the extended passage of time"?

I understand that examining social influence may lie outside of the purview of the current paper, but the authors imply that they do not have the data to examine social influence processes because they were not able to collect data two years later. Would they not be able to examine social influence processes with their current data? For example, the authors could test whether one person's neural processing changed more/ became more similar to the other person's (whereas the other person's may have stayed the same, suggesting they influenced the other)?

More Minor

This paragraph is redundant (analyses and results stated twice - I would recommend starting the paragraph at the word "Next" and moving the references to Methods and Supplemental Figures).

"In exploratory analyses, we investigated if accounting for inter-individual similarities in ratings of enjoyment and interest in the stimuli would fully account for and/or significantly diminish the difference in neural similarity between friends and pairs of individuals characterized by a social distance of 3 (see Methods). The effect in the OFC remained significant when controlling for inter-individual similarities in ratings of enjoyment and interest (Supplementary Fig. 6), suggesting that similarity in neural responding in this brain region captured similarities in friends' responses to the stimuli that the interest and enjoyment ratings did not. Next, to inform our interpretation of the psychological meaning of the OFC result, we tested if controlling for inter-individual similarities in ratings of enjoyment and/or interest would significantly diminish the difference in neural similarity between friends and pairs of individuals characterized by a social distance of 3. Neither controlling for inter-individual similarity in interest ratings nor controlling for inter-individual similarity in enjoyment ratings significantly decreased the difference in neural similarity between friends and individuals characterized by a social distance of 3. "when controlling for enjoyment and interest ratings (Supplementary Fig. 9)." Is redundant with: "if accounting for inter-individual similarities in ratings of enjoyment and interest in the stimuli would significantly diminish the difference in neural similarity between these two groups (see Methods). This analysis was repeated for each of the brain regions in which we observed a significant difference in mean neural similarity (i.e., regions outlined in black in Fig. 6). Controlling for inter-individual similarities in ratings of enjoyment of the stimuli significantly decreased the extent to which individuals who grew closer over time exhibited higher neural similarity in a portion of right superior parietal cortex (Difference in mean normalized neural similarity = 0.015; $p < 0.05$) relative to individuals who grew apart over time. However, this effect was not observed when controlling for inter-individual similarities in ratings of interest in the stimuli or in other brain regions." "Controlling for inter-individual similarities" ... "The one exception was..." I wouldn't make too much of this - could be noise given the number of analyses conducted.

I hope these suggestions help the authors improve the paper. Again, I believe this paper makes a highly valuable contribution to our understanding of human behavior.

Reviewer #2:

Remarks to the Author:

"Neural Similarity Induces Friendship" is a fascinating manuscript, drawing from sources in psychology, neuroscience, and sociology. The manuscript appropriately employs Ronald Burt's network tie-eliciting method for obtaining information about MBA students' informal social connections to one another at three moments in time during the first year of the graduate program. Strikingly, and this is a matter for further comment, the authors report gathering complete network data for an entire student population numbering in the hundreds, not just once but multiple times.

It is not new for attitudinal homophily or personality homophily (or, for that matter, demographic or cultural homophily) to be discussed; the empirical establishment of the pattern that "birds of a feather flock together," especially in demographic characteristics, is a solid finding of the social network and social-psychological literature. However, what the authors offer here is some evidence of functional involvement of particular areas of the human brain, through which sociocultural similarity may produce observable behavioral acts to maintain and preserve some social ties. This possibility is a novel and important contribution.

I use the word "may" rather than "does" intentionally. The authors assert that alignment of neural characteristics between student dyads in an MBA program "determines," "fosters," "induces," or "causes" the creation or withering of social ties between those dyads. These are strong causal verbs, and the authors refer to their longitudinal study design to justify the strength of their claim. However, the study design is not one that controls for multiple possible confounding social factors. The study admirably shows that neural similarity exists prior to social tie formation, and that time order is important. However, the study design does not restrict variation to the measured aspects of brain activity alone or completely control for other personal or social factors; the authors acknowledge the presence of considerable social diversity among the students being studied, some dimensions of which appear to be measured and some dimensions of which are surely not.

To provide one possible example of a confounding correlation, the authors note in passing that MBA students live and eat close to one another to describe the stimulating environment for tie formation. But surely not all 288 of these students live in the same floor on the same building, and surely they don't all eat at the same table. Given well-known tendencies toward segregation in residence, dining, and other functions of educational institutions, could tie formation and tie dissolution during the time period studied be connected to greater proximity of more sociodemographically similar students, who it just so happens may (due to the localization of culture sociodemographically) have the "generalized shared reality" that would lead to neural similarities? If so, then the neural similarities would be a side effect of a dynamic driven primarily by social availability, only appearing to be the factor that "induces friendship" because it is strongly associated as a variable with the underlying sociological driver, segregated availability. Although we are living in the 21st Century, it is too soon to declare that racial, national, gender, birth cohort, religious, linguistic, disciplinary, and other dimensions of educational segregation have been eliminated.

If the main driver of similarity in association is indeed psychological, perhaps the precedence of neural similarity among those who will form and maintain ties in the future is due to another psychological mechanism, such as inter-category animosity. Animosity might lead to segregation, which would in turn could lead on the one hand to reinforcement of cultural differences that became visible in the kind of neural differences that the researchers observe, and on the other hand to the tie formation that the researchers observe. The neural association with association would in this case be spurious.

Regarding these two possibilities, the authors in supplemental files include two figures showing findings that claim to "control" for "similarities in demographics," the method by which this occurs, or for what demographic categories, is not described in the main paper or any supplements. Regardless, it is quite unlikely that all relevant sociodemographic dimensions of segregation that shape opportunities for contact and tie reinforcement have been measured, since there are simply so many of these dimensions that exist.

Finally, the accurate establishment of clear social network distances, especially beyond a distance of 2, can be threatened by missing data, with multiple missing ties for each person who doesn't report their ties. It is fortunate in a methodological sense, then, that the authors report obtaining network data for every single one of the 288 MBA students at Time 2, and for all but 1 MBA student at Time 3. Given the typically low response rates to study solicitations, this is odd. The methods section regarding the 43 fMRI participants states explicitly that data collection was in accordance with the local ethical review board. The 288 MBA students universally completing a survey reporting personal information about themselves and individuals and their social choices among a set of pre-professional peers is a strange accomplishment. Were students required by educational officials in the MBA program to participate, either explicitly or as a way to curry favor? If so, how is that consistent with the principle established in federal HHS guidelines that participation in human subjects research be voluntary and be driven by informed affirmative consent? If all 288 student freely volunteered once, and 287 freely volunteered twice, that's a considerable accomplishment. Some clarification of this process would be appreciated.

Reviewer #3:

Remarks to the Author:

Hyon and colleagues use a naturalistic longitudinal design to show that pre-existing neural similarity (measured during film-viewing) is related to future friendship status (social distance) 8 months later. The authors find a significant difference in pre-existing neural similarity (measured at the beginning of the school year) between subjects who eight months later became friends (social distance 1) and those who were friends-of-friends-of-friends (social distance 3) in orbitofrontal cortex (OFC).

They also find that the dyads whose social distance decreased (they became closer friends) between 2 months and 8 months post-scan were associated with higher pre-existing neural similarity in a variety of brain areas. This is a concise study—using a very neat longitudinal design!—on the very exciting topic of neural homophily and real-world friendship. The results are a bit mixed, and I think there are a couple places where the authors could do a bit more work to build the reader's confidence. There were also several places in the narrative of the Results section where I had a hard time following the logic of analyses. I articulate some comments below that I hope will strengthen the manuscript.

Comments:

The authors organize the data for statistical tests in particular ways, and my main concern is that readers will worry that these effects might hold up if you slice the data differently. I don't think the slicing is actually that arbitrary, but it doesn't seem well-motivated or thoroughly explained while reading the Results section. For example, a reader might worry that $N = 43$ is a relatively small sample size, without fully appreciating that the fMRI data had to be collected as quickly as possible as soon as subjects arrived on campus to minimize the chance that they would start interacting with each other and becoming friends. A little more motivation and explanation here would go a long way.

In another example, I think we need a little more motivation / explanation in the Results as to how you arrived at social distance groups 1, 2, and 3. For example, how do you determine the threshold between level 2 "friends-of-friends" and level 3 "friends-of-friends-of-friends"? Are these groups of dyads similar in size? Is there a more continuous metric on the friendship network graph that might provide corroborative results? In Figure 3, I would also describe what these social distance groups mean and how dyads are assigned to these groups in the caption.

In one of the core results, the authors find a significant difference in neural similarity between friends (social distance 1) and friends-of-friends-of-friends (social distance 3) in an orbitofrontal cortex (OFC) parcel. This is a difficult part of the brain to capture with fMRI due to susceptibility artifacts—i.e. due to the air/bone boundary in the nearby sinuses—and signal here may be heavily affected by head motion (although 25 ms TE might help!). Can the authors provide any supplementary analyses to increase readers' confidence that we actually have reliable signal in this OFC parcel? For example, how correlated is the time series in this parcel with framewise displacement (FD)? Does preprocessing / nuisance regression mitigate this correlation? Another example: Do we see substantial ISC values in this parcel? This would suggest that there is in fact stimulus-driven signal in this area that's reliable across subjects. Last example: Does regressing out subject-level FD in the group-level test impact this result? (I understand this is tricky because you're dealing with dyads, not individual subjects, here.)

What's the motivation for z-scoring similarities across dyads within each region? I assume this is why the similarity values are generally centered around zero in Figures 3 and 5, and why we might (incorrectly) think the cool values at social distance 3 correspond to negative similarities. Does this change how we interpret the magnitudes across parcels within a group? Was this transformation performed only for visualization? I think it would be helpful to plot the "raw" (not parcelwise z-scored across dyads) ISCs in the Supplementary Materials.

How exactly were the fMRI subjects recruited? Were they randomly sampled, or did they self-select to participate in the fMRI study. It seems like they fairly uniformly sample the social network as measured at time 2 (but not so much at time 3). I think the design is safe from concerns like "the kind of people who sign up for an fMRI study make friends in a particular way"—but it might benefit the authors to explicitly preempt this kind of concern.

In the Figure 1 caption, can you briefly include a sentence describing how similarity is measured and what kind of dimensionality reduction is used to generate these visualizations of the social network structure? Any idea why the red fMRI subjects end up biased toward one side of the projection at time 3?

I think it's generally helpful to plot unthresholded brain maps so that readers can see effects across the whole brain. But I would suggest that the authors make an explicit note in figure captions presenting unthreshold maps (e.g. Figs. 3, 5, Supplementary, etc.) that no statistical threshold or correction for multiple tests is performed.

The authors are performing statistical tests on dyads, which necessarily introduces non-independence among samples: i.e. one subject contributes to multiple dyads and the number of dyads will yield inflated degrees of freedom (Chen et al., 2016; Nastase et al., 2019). Are the authors taking any statistical precautions to address this non-independence?

At the beginning of the Results section, you mention the film clips very briefly in passing. It might be worth adding another sentence here to clarify if there was any particular rhyme or reason to the film clips selected by the experimenters. Similarly, is the content of these particular naturalistic stimuli doing any theoretical work here? Would you expect the same kind of results with resting-state connectivity?

In the Introduction, you mention that prior work relating inter-individual similarities in self-reported personality traits to social network structure "has yielded null or inconsistent results." Do the authors have any further explanation as to why? For example, is it possible that personality research simply uses much larger samples with higher statistical standards (relative to neuroscience)? Or is there some mismatch between behavioral assays and the constructs they aim to capture (e.g. Eisenberg et al., 2019; Dang et al., 2020)?

The authors have analyzed data from Ivy League business school students in prior studies. Obviously this sample is not

guaranteed to be representative of the broader population. Do the authors have any empirical data—e.g. personality metrics—on whether this particular sample systematically deviates from the “general” population in any particular way?

References:

Chen, G., Shin, Y. W., Taylor, P. A., Glen, D. R., Reynolds, R. C., Israel, R. B., & Cox, R. W. (2016). Untangling the relatedness among correlations, part I: nonparametric approaches to inter-subject correlation analysis at the group level. *NeuroImage*, 142, 248–259. <https://doi.org/10.1016/j.neuroimage.2016.05.023>

Dang, J., King, K. M., & Inzlicht, M. (2020). Why are self-report and behavioral measures weakly correlated? *Trends in Cognitive Sciences*, 24(4), 267–269. <https://doi.org/10.1016/j.tics.2020.01.007>

Eisenberg, I. W., Bissett, P. G., Zeynep Enkavi, A., Li, J., MacKinnon, D. P., Marsch, L. A., & Poldrack, R. A. (2019). Uncovering the structure of self-regulation through data-driven ontology discovery. *Nature Communications*, 10, 2319. <https://doi.org/10.1038/s41467-019-10301-1>

Nastase, S. A., Gazzola, V., Hasson, U., & Keysers, C. (2019). Measuring shared responses across subjects using intersubject correlation. *Social Cognitive and Affective Neuroscience*, 14(6), 667–685. <https://doi.org/10.1093/scan/nsz037>

Signed: Samuel A. Nastase

Version 1:

Decision Letter:

28th October 2024

Dear Dr Parkinson,

Thank you once again for your manuscript, entitled "Neural Similarity Induces Friendship," and for your patience during the peer review process.

Your manuscript has now been evaluated by 2 of the original reviewers, whose comments are included at the end of this letter. In the light of their advice, I regret that we cannot offer to publish your manuscript in *Nature Human Behaviour*.

While the reviewers found your manuscript to have improved during revision, Reviewer #2 maintains important outstanding concerns about the results: only three demographic variables are controlled for—and when they are, the relationship between neural similarity and friendship almost disappears. We feel that these reservations are sufficiently important as to preclude publication of this work in *Nature Human Behaviour*.

Although we cannot offer to publish your manuscript in *Nature Human Behaviour*, I have spoken with my colleagues at *Communications Psychology*, and subject to a suitable revision they have agreed to publish your manuscript there (link provided below the reviewers' comments). Your revision would need to fully address the remaining reviewer concerns and would be subject to re-review. In particular, you will need to refrain from any causal language, with any speculation on causality being confined to (a small section of) the Discussion and highlighted as speculative. This limitation on the interpretation, that neural similarity may be a proxy of other factors that determine mid-term friendship, should be mentioned in the Abstract. Conducting the additional analyses suggested by Reviewer #2 is strongly encouraged as these will deepen the insights that can be derived from the work, but completing these analyses is not sufficient to make causal claims, regardless of their outcome.

Please contact the Chief Editor, Marike Schiffer (CC'd here) if you have any questions about the required revisions or the journal.

Please also contact Marike if you wish to discuss the potential of a waiver of the journal's Article Processing Charges upon transfer. While the journal is able to offer waivers to a limited number of post-review transfers, please note the requirement to let the editors know that you wish to explore this option within a month from now.

I am sorry that we cannot be more positive on this occasion but hope that you will find our reviewers' comments helpful when preparing your paper for submission elsewhere.

Sincerely,

Nature Human Behaviour

Reviewer expertise:

Reviewer #2: social networks, homophily, social cognition

Reviewer #3: neural synchronization, fMRI, naturalistic stimuli)

Reviewers' Comments:

Reviewer #2 (Remarks to the Author):

Feedback on First Revision of "Neural Similarity Induces Friendship"
October 2024
Reviewer 2

I appreciate the authors' revisions and detailed comments regarding those revisions. This remains a fascinating piece of research. I identify the following issues as still needing attention:

Editor Comment 1):

Regarding issues of network autocorrelation and non-independence of measurements raised by other reviewers, I am satisfied by the authors' response, which appropriately references the social network research literature and particularly the QAP-style logic identified by the authors.

Editor Comment 3) and R2.2):

Despite the general assurances of the authors, I remain skeptical. A participation rate of 100% by students is far outside the range of participation I am familiar with in wholly voluntary studies, and ethical considerations include implicit as well as explicit pressure. The authors acknowledge that they engaged in an offer of "personalized, educational feedback" to "encourage participation," and that (passively speaking) these students "were invited to participate in the online social network survey as an optional part of their coursework." I am curious regarding that nature of the offer of "personalized, educational feedback." Could that be explained, with inclusion of the text of that offer and a description of who would offer it? I am also curious about who (in the active voice) "invited" participation and using what additional signals of authority. In an MBA program, in which students measure success or failure not simply by the passing of classes but in no small part on the basis of recommendations from and the formation of social ties with faculty and administration, is it reasonable to expect that students might perceive implicit an obligation to engage in the "optional" work of supporting said faculty and administration? Were the names of participants and non-participants available to any faculty or administration members? Supplying the consent forms supplied to students and the text of the recruitment communications would help resolve skepticism on this point.

Editor Comment 5) and R2.1):

Although the word "may" has been added, the words "determines," "fosters," "induces," "causes", "produce," "especially large impact," "facilitate," "conducive to," "due to," and "shaping" are still present in the manuscript. Lines 461-463 involve the claim that a result "sheds light" on a "causal relationship." This is all causal language, and while the word "may" is added the implied message is that there's something causal going on here. There is absolutely no "may" about the title, which asserts openly that "Neural Similarity Induces Friendship." I am not convinced that the authors have thoroughly followed the editors' request to "remove causal claims."

I am glad, to see that the authors have indicated in the discussion, without using the word "spurious," that there's quite a possibility that "Neural Similarity does NOT Induce Friendship," and that neural similarity might be epiphenomenal, with social forces playing the shaping roles. Perhaps neural similarity does induce friendship. Perhaps neural similarity is a conduit but not independent. The best title here, given the uncertainty, would be "Neural Similarity Might or Might Not Induce Friendship" or perhaps "Social and Neural Similarity Precede Friendship."

Particularly regarding living proximity in R2.1), that "nearly all students applied to on-campus housing and were randomly assigned to housing units" might lessen demographic homophily in housing, but should not be expected to have eliminated it. Sorting toward similarity is a well-known first-year student phenomenon. I'll quote Bahns et al 2013:

"Research supports both individual-level and relationship-level predictors of roommate satisfaction and breakup. Similarity—an emergent property of the roommate dyad—has been repeatedly shown to affect the decision to stay with or leave a roommate, including matching on personality (Carli, Ganley, & Pierce-Otay, 1991; Heckert et al., 1999; cf. Lapidus, Green, & Baruh, 1985), values (Jones, McCaa, & Martecchini, 1980; cf. Lapidus et al., 1985), sleeping habits, study habits, and neatness (Fuller & Hall, 1996; Jones et al., 1980; Lapidus et al., 1985), as well as shared activities (Lovejoy, Perkins, & Collins, 1995) and same-race compared to mixed-race roommates (Shook & Fazio, 2008)."

In contrast, it is helpful to know that students were randomly assigned to study groups for an entire term.

The authors write in the revision that "to statistically account for demographic similarities that may be related to similarities in

neural responding and/or social network proximity, we regressed out inter-subject similarities in age, gender and nationality from ISCs in each brain region...", with what appears based on my reading to be a comparison between a baseline model without the three demographic controls (with the difference between those growing closer and those growing apart as delta-ISC in Supplementary Table 2) and a model that includes controls for age similarity, gender similarity, and nationality similarity (showing as delta-ISC in Supplementary Table 3).

It's worth noting that the authors statistically very partially accounts for demographic similarities, considering that it leaves out a large number of social dimensions that if other research bears out should expect to exhibit homophily: region/geography, economic class, legacy status, disability status, religion, race, ethnicity, political party, first language, major, undergraduate alma mater, and immediately prior occupation.

What trend in the data does the inclusion of these three variables provoke? If my interpretation accurately reflects the meaning of Supplementary Tables 2 and 3, it appears that the difference in ISC between those dyads growing closer versus those dyads growing apart increases in 11 brain parcels when controlling for the 3 demographic dimensions, stays the same in 1 brain parcel, and decreases in 30 brain parcels.

Would the authors concur in that interpretation of results? If so, this at least suggests that a portion of differences between the two sets of dyads might be explained by the meager 3 controls for similarity, and that if we included more dimensions (many among those I've listed above exhibiting quite strong homophily in replicated research), the differences might further erode. There's a whole lot of unobserved heterogeneity here, regarding variables known to have a very strong effect on both networks and neuron-shifting culture, and that lessens my confidence that we're seeing nonspurious neuronal similarity as an independent dimension of homophily. Without full controls of meaningful sociological variables, this isn't a slam dunk and maybe not even a basket. While intriguing, it looks like some more controls are needed. Is there a way to find at least some more of these demographic variables in student records? Speaking hopefully, the authors' consent forms might have already given permission to access the school's records to obtain individual-level variables.

These are the remaining concerns I have with the article, and I wish to reiterate my appreciation of the cleverness of this method in uncovering physical markers at least correlated with social outcomes.

Sincerely,
James Cook, Associate Professor of Sociology, University of Maine at Augusta

References

Bahns, A. J., Crandall, C. S., Canevello, A., & Crocker, J. (2013). Deciding to dissolve: Individual-and relationship-level predictors of roommate breakup. *Basic and Applied Social Psychology*, 35(2), 164-175.

Fuller, B. E., & Hall, F. J. (1996). Differences in personality type and roommate compatibility as predictors of roommate conflict. *Journal of College Student Development*, 37, 510-517.

Heckert, T. M., Mueller, M. A., Roberts, L. L., Hannah, A. P., Jones, M. A., Masters, S., . . . Bergman, S. M. (1999). Personality similarity and conflict among female college roommates. *Journal of College Student Development*, 40, 79-81.

Jones, L. M., McCaa, B. B., Jr., & Martecchini, C. A. (1980). Roommate satisfaction as a function of similarity. *Journal of College Student Personnel*, 21, 229-234

Lapidus, J., Green, S. K., & Baruh, E. (1985). Factors related to roommate compatibility in the residence hall: A review. *Journal of College Student Personnel*, 26, 420-434. doi: 10.1111/j.1475-6811.1999.tb00202.x

Lovejoy, M. C., Perkins, D. V., & Collins, J. E. (1995). Predicting fall semester breakups in college roommates: A replication using the social satisfaction questionnaire. *Journal of College Student Development*, 36, 594-602.

Shook, N. J., & Fazio, R. H. (2008). Roommate relationships: A comparison of interracial and same-race living situations. *Group Processes and Intergroup Relations*, 11, 425-437. doi: 10.1177/1368430208095398

Reviewer #2 (Remarks on code availability):

The code is useful and helpfully annotated. The full data .csv file has been helpful to examine in the completion of this review.

Reviewer #3 (Remarks to the Author):

I think the authors have adequately addressed all of my comments and the manuscript is considerably improved. I

appreciate their thorough answers to the other reviewers' very thoughtful comments as well. In particular, I think the supplementary statistical analyses and the more measured language around causality are important improvements. Two minor notes from my final read-through:

Missing a word here: "which has recently been [shown?] to be more predictive of behavior than rest"

Cosmetic note: In Figures S13 and S15, due to the large number of parcels in DMN, the purple color from the legend basically looks black; maybe you could just remove the black edges around each bar?

Reviewer #3 (Remarks on code availability):

The authors provide code and a Jupyter Notebook demo—looks useful.

Following suitable revisions, you may want to consider transferring your manuscript. Although we cannot offer to publish your manuscript, I have consulted with my colleagues at Communications Psychology, and they have agreed to continue the review of your manuscript. To transfer your manuscript please use our manuscript transfer portal. You will not have to re-supply manuscript metadata and files, unless you wish to make modifications. For more information, please see our [manuscript transfer FAQ](http://www.nature.com/authors/author_resources/transfer_manuscripts.html?WT.mc_id=EMI_NPG_1511_AUTHORTRANSF&WT.ec_id=AUTHOR) page.

Version 2:

Decision Letter:

Dear Dr Parkinson,

Thank you for your correspondence asking us to reconsider our decision on your Article, "Neural Similarity Induces Friendship". After careful consideration we have decided that we would be willing to consider a revised version of your manuscript.

Along with your revised manuscript, you should also submit a separate point-by-point response to all of the concerns raised by the referees, in each case describing what changes have been made to the manuscript or, alternatively, if no action has been taken, providing a compelling argument for why that is the case.

Please note that, in addition to addressing the remaining analytical concerns, we ask that you also provide a full response to the ethical concerns raised by Reviewer #2 regarding participant enrollment. If we feel that a substantial attempt has been made to address the referees' comments, this response will be sent back to the referees - along with the revised manuscript - so that they can judge whether their concerns have been addressed satisfactorily or otherwise.

I should stress, however, that we would be reluctant to trouble our referees again unless we thought that their comments had been addressed in full.

- ensure it complies with our format requirements as set out in our [Guide to Authors](http://www.nature.com/nathumbehav/info/gta).

- state in a cover note the length of the text, methods and figure legends; the number of references and the number of display items.

Please ensure that all correspondence is marked with your Nature Human Behaviour reference number in the subject line.

Please use the following link to submit your revised manuscript:

Link Redacted

We hope to receive your revised paper within four weeks. If you cannot send it within this time, please let us know so that we can close your file. In this event, we will still be happy to reconsider your paper at a later date so long as nothing similar has been accepted for publication at Nature Human Behaviour or published elsewhere in the meantime. Should you miss the

four-week deadline and your paper is eventually published, the received date will be that of the revised, not the original, version.

I look forward to hearing from you soon.

Best regards,

[REDACTED]
[REDACTED]
Nature Human Behaviour

Version 3:

Decision Letter:

Our ref: NATHUMBEHAV-23082716C

19th February 2025

Dear Dr Parkinson,

Thank you for submitting your revised manuscript "Neural Similarity Predicts Whether Strangers Become Friends" (NATHUMBEHAV-23082716C). It has now been seen by one of the original referees. Reviewer #2 was unavailable, but Reviewer #3 evaluated your revisions and endorsed publication. We will therefore be happy in principle to publish it in Nature Human Behaviour, pending minor revisions to comply with our editorial and formatting guidelines.

We are now performing detailed checks on your paper and will send you a checklist detailing our editorial and formatting requirements within two weeks. Please do not upload the final materials and make any revisions until you receive this additional information from us.

Sincerely,

[REDACTED]
[REDACTED]
Nature Human Behaviour

Responses to the Editor and Reviewers

Thank you for providing the list of prioritized referee points to help guide the revision process. In this document, we provide detailed responses to all reviewer concerns, which are preceded by a summary of the key revisions that were completed in response to the editor's prioritized set of referee points. For clarity and concision, we have enumerated reviewer concerns such that Reviewer #1's first point is labeled R1.1, Reviewer #1's second point is labeled R1.2, Reviewer #2's first point is labeled R2.1, and so on; we use these labels when referring to particular reviewer comments throughout the document. Our responses to editor and reviewer concerns are provided in blue text and text quoted from the revised manuscript is provided in *italics* with added text underlined and ~~deleted text~~ struck through.

Responses to Comments from the Editor:

1) Both Reviewer #1 and Reviewer #3 express concerns with statistical tests on non-independent samples. Please address those in full, possibly by running multilevel linear models as suggested by Reviewer #1.

We have clarified and provided evidence that the specific permutation testing approach used in the manuscript accounts for the non-independent nature of the data, as all permuted versions of the dataset preserve the dependency structure of the observed data. We apologize for not having made this sufficiently clear in the original version of the paper. We now discuss this in greater detail on pp. 8-9 and p. 12 of the revised Methods section, where we indicate that, whereas permuting the data at the level of dyads would not be sufficiently conservative (as the reviewers allude to, since the dependencies among dyads in the permuted and observed datasets would not be equivalent using such an approach), the current approach – in which the brain data corresponding to individual participants (i.e., nodes in the social network) is shuffled in each permuted version of the dataset, while keeping the structure of the social network constant, before similarities in participants' brain data are computed and related to their distances in the social network – produces permuted datasets with dependency structures equivalent to that of the observed (non-permuted) data. We also point out that this approach has been used frequently in prior social network research. As suggested by the reviewers, we have run multilevel linear models to complement our current analyses and provided the corresponding results in the supplement. Please see R1.1, R1.2, and R3.8 for detailed discussions that address this concern.

2) Reviewer #3 requests additional analyses to increase confidence in the robustness of the results. Please perform those.

We performed the additional analyses as suggested and include the results in R3.3.

3) Please clarify how participant sampling and recruitment was performed in detail, as both Reviewer #2 and #3 have concerns about this aspect of the research. In particular, Reviewer #2 has a potentially serious concern about ethics. Please confirm that participation was truly voluntary, or elaborate on the precise conditions of participant recruitment.

In the revised manuscript, we confirm that all participation was voluntary and elaborate on the recruitment process, as described in more detail in R2.2. Specifically, all participants were recruited from

a first-year graduate cohort that later enrolled in a leadership course within their program, and as a part of their course, they were invited to participate in the online social network survey. They were reminded that participation was voluntary and would not affect their performance in the course. We responded to Reviewer #3's concern about participant sampling and the generalizability of the results in R3.11.

4) Please expand on what insights the longitudinal design used here can provide above and beyond your 2018 Nature Communications study. We ask that you clarify the relationship between the two pieces of work, including the potential overlap between the two datasets and why the sample size differs.

Since the 2018 Nature Communications study was cross-sectional, its findings could be explained by neural similarity simply having *resulted from* proximity in a friendship network. Put another way, the results of that study could arise purely from phenomena such as social influence, contagion, or exposure to common environments unfolding among people *who were already friends*. By acquiring neural data before participants met one another (typically within *one day* of when they arrived in a new city and academic institution, as shown in Supplementary Figure 1), and then following those individuals and their broader social network over time, the current study is able to test, for the first time, if and how pre-existing neural similarity may contribute to subsequent social connection. Moreover, because the current dataset contains multiple waves of social network data, it also allows us to test for links between pre-existing neural similarities and reductions in social distance over time. We have clarified these distinguishing features of the current dataset on pp. 3-4 of the revised manuscript: *“For example, the previously observed relationships between neural similarity and social network proximity may arise purely from people who were already friends becoming similar to one another due to social influence, contagion, or exposure to common environments or experiences as a result of proximity in their social networks. Here, we leveraged a longitudinal study design to test, for the first time, whether pre-existing similarities in neural responses to naturalistic stimuli (prior to meeting one another) predicted future proximity in a real-world friendship network eight months later, and whether pre-existing neural similarity was also associated with increases in social closeness over time.”* Finally, we note that although both samples from the current study and the 2018 Nature Communications study were recruited from the same graduate program of the same school, they were from two separate cohorts separated by several years. Thus, the two datasets do not overlap and their sample sizes differ. We now explicitly clarify that these were different samples on p. 3 of the revised Methods section (*“Criteria used to select stimuli are described in more detail elsewhere in a previous manuscript that used the same stimuli in a different sample⁶⁰.”*)

5) Please remove causal claims while working to address Reviewer #2's concerns regarding threats to potential causal inference.

We have softened and qualified the causal claims and addressed Reviewer #2's concern in detail in R2.1. For example, we added text to clarify relevant methodological details (which we regret having omitted previously) regarding the fact that participants were subjected to a substantial amount of random assignment regarding the people with whom they studied and lived. Thus, although this is an observational study that cannot provide decisive evidence for a causal relationship, participants underwent significant random assignment regarding the composition of their social spheres. We have also clarified what demographic variables we controlled for in our supplementary analyses. Additionally,

Reviewer #2's comments made us realize that we had previously neglected to give sufficient attention to the potential role of demographic similarities in the paper's Discussion section. Therefore, we have added new text to the Discussion section to clarify our thinking on this topic, which is consistent with what Reviewer #2 posits in their review ("*what the authors offer here is some evidence of functional involvement of particular areas of the human brain, through which sociocultural similarity may produce observable behavioral acts to maintain and preserve some social ties*"); we expect that dyadic similarities in wide-ranging and numerous demographic attributes and other factors give rise to neural similarities, which in turn, impact the propensity for people to become and stay friends, as detailed in R2.1. We also indicate that additional studies will be necessary to test for and elucidate causal relationships among these variables.

Responses to Comments from Reviewer #1:

Theoretical Contribution

I believe this paper makes a groundbreaking theoretical contribution to our understanding of what makes humans connect and deepen their relationships. Prior work in this area has been largely cross-sectional, and for the first time this work allows us to establish causality by measuring neural similarity before friendship formation and measuring decreases in social distance over time. I find it impressive that this pattern of results held controlling for demographic similarity and for similarity in interest and enjoyment of the stimuli (suggesting that these effects are not driven by demographic similarity or agreements about valence, but rather deeper interpretations of the stimuli). These results greatly advance not only social neuroscience, social psychology, and interpersonal science, but our understanding of human behavior as a whole. I believe the paper could be improved by running more meaningful analyses (multilevel linear models) and by more clearly and consistently referring to the methods in the rest of the paper.

Thank you for your encouraging feedback on this manuscript.

Analyses

R1.1) My greatest concern is that the authors analysed their data using permutation testing procedures. Why did the authors not run multilevel linear regressions? Running MLM's (with binary predictors and continuous outcomes) would allow the authors to account for the interdependence of the datapoints within individuals and within networks, etc., and would produce effect sizes and confidence intervals rather than just p-values. If the authors have a reason for conducting permutation testing rather than regression models, this should be explained and references should be provided to support this decision (and the decision not to run multi-level models/ account for the interdependence of datapoints should also be explained), even if in the supplement.

Thank you for your thoughtful review and suggestion on data analysis methods. We apologize for previously not clearly conveying how our data analytic approach accounts for the interdependence of datapoints. As the reviewer points out, a simple permutation testing approach in which, for example, neural similarities were shuffled at the level of dyads, would fail to account for the interdependence of data points and would thus be insufficiently conservative. However, the approach that we adopt in the current manuscript generates permuted datasets that have equivalent interdependence structures to that of the observed (non-permuted) data. Specifically, in each permutation, prior to calculating inter-subject neural similarities, participant labels for the fMRI datasets are randomly shuffled (i.e., each set of fMRI observations is permuted at the individual level, rather than the dyadic level). Next, within each permuted dataset, inter-subject neural similarities for all parcels are computed and related to the participants' (unshuffled) social network data. We clarify these points in the revised Methods section:

- *“To assess the statistical significance of this difference, we implemented the following permutation testing procedure. Neuroimaging data were randomly shuffled across fMRI participants, at the individual level, 1,000 times while holding all else in the dataset constant. Specifically, prior to calculating neural similarities, participant labels for the fMRI dataset were randomly shuffled (i.e., each set of fMRI data was permuted at the individual level) to ensure that*

the permuted datasets would have equivalent dependency structures to the observed (non-permuted) data. In addition, we note that node-level permutations are often used for hypothesis testing on social network data across a wide range of contexts, particularly when probing potential links between individual nodes' features (here, neural responses) and their patterns of interactions or associations with one another, given that this method generates null models that break potential links between these variables while preserving the structure of the network⁷⁸. In each permuted dataset, the inter-subject neural similarities for all parcels were computed for each dyad, and the mean neural similarity among individuals characterized by a social distance of 2 or 3 (~~shuffled~~) was again subtracted from the mean neural similarity among individuals characterized by a social distance of 1 (~~shuffled~~)." (pp. 8-9 of the revised Methods section)

- *"To assess the statistical significance of this difference, we implemented the following permutation testing procedure. Neuroimaging data were randomly shuffled across fMRI participants, at the individual level, 1,000 times while holding all else in the dataset constant. Specifically, prior to calculating neural similarities, participant labels for the fMRI dataset were randomly shuffled (i.e., each set of fMRI data was permuted at the individual level) to ensure that the permuted datasets would have equivalent dependency structures to the observed (non-permuted) data. In each permuted dataset, the inter-subject neural similarities for all parcels were computed for each dyad, and the mean neural similarity among individuals who did not grow closer over time was again subtracted from the mean neural similarity among individuals who did grow closer over time."* (p. 12 of the revised Methods section)

As noted within the first passage of revised text quoted above, similar permutation testing procedures have been frequently used in social networks research, given that this approach to permutation testing accounts for the non-independent nature of social network data, and dyadic data more generally, by generating permuted datasets that correspond to the null hypothesis but preserve the same dependency structure of the non-permuted data. For example, Wey & Blumstein (2010) used a node-label shuffling approach to test for links between dyadic similarity and the frequency of particular kinds of interactions in social networks of marmots. In social networks research, a particularly common implementation of node-label permutation is referred to as the Quadratic Assignment Procedure (QAP). One presentation of QAP (Simpson, 2001) specifically states that this approach "has been used in social network analysis, and is useful for analyzing dyadic data sets, i.e. data sets where pairs of entities are analyzed. Examples include: ... Are people more likely to be friends if they share similar characteristics, such as being about the same age, or using the same statistical software?" Variants of this node-label shuffling approach are discussed in detail in Krackhardt (1987) and Krackhardt (1988). The current manuscript's node-label shuffling approach is logically equivalent to common implementations of QAP with respect to how null distributions are estimated and how the non-independence among dyads is preserved in those null distributions. We hope that our expanded explanation of our permutation testing approach and its appropriateness in the revised Methods section (quoted above) helps to clarify these points.

As suggested, we also ran multilevel linear regressions with crossed random effects to complement our current analyses using permutation testing (see p. 25 of the revised supplement for details on analyses using linear mixed models). Aligning with results from permutation testing, we found that pre-existing similarity in left OFC significantly predicted future friendship (i.e., a social distance of 1) when compared to dyads characterized by a social distance of 3 ($\beta = 0.47$, $SE = 0.12$, $p < 0.001$, FDR-corrected). For our

second set of main analyses on social distance change, we also fit a linear mixed effect model with crossed random effects and performed a planned contrast analysis (i.e., $ISC_{\text{dyads that grew closer}} \text{ vs } ISC_{\text{dyads that grew apart}}$; $ISC_{\text{dyads that grew closer}} \text{ vs } ISC_{\text{dyads that have no change in their social distance}}$; $ISC_{\text{dyads that grew apart}} \text{ vs } ISC_{\text{dyads that have no change in their social distance}}$). For the contrast $ISC_{\text{dyads that grew closer}} \text{ vs } ISC_{\text{dyads that grew apart}}$, although none of the brain parcels reached significance after correcting for multiple tests, we observed a similar pattern of results such that dyads that grew closer generally tended to show higher neural similarity across brain regions. Results from these multilevel linear regressions are included in Supplementary Figs. 12 and 13 (pp. 12-13 of the revised supplement). Supplementary Fig. 13, which depicts the above-described trend, is also provided below for convenience:

Supplementary Figure 13. Dyads that grew closer, relative to dyads that grew apart, tended to show higher neural similarity across brain regions spanning multiple brain networks. Regression coefficients for each brain parcel from the linear mixed model and planned contrast analysis for $ISC_{\text{dyads that grew closer}} \text{ vs } ISC_{\text{dyads that grew apart}}$, grouped by their corresponding brain networks, are shown.

R1.2) More broadly, the authors only report p-values in the results, which does not give us an idea of effect sizes. Even if linear models are not used, I would recommend adding effect sizes, 95% confidence intervals, and degrees of freedom throughout the results section.

Thank you for your recommendation. We have included 95% bootstrap confidence intervals when reporting results. For instance, 95% bootstrap confidence intervals were added when reporting results regarding pre-existing neural similarity in a portion of the left OFC predicting friendship 8 months later (p. 9 of the revised manuscript). For the analyses comparing pre-existing neural similarity for dyads who grew closer versus apart, we have added 95% bootstrap confidence intervals to Supplementary Tables 2-5.

We added the following section in the revised Methods to describe how we bootstrap the nodes to yield the 95% bootstrap confidence intervals: *“To get the 95% bootstrap confidence intervals, we generated 1,000 bootstrap samples of the fMRI participants at each time point. Then, for each unique dyad from the unique nodes in a given bootstrap sample, we retrieved the ISCs of each brain region. Subsequently, we calculated the mean difference in ISCs, normalized within each brain region, between comparison groups... for each bootstrap sample. The 95% confidence intervals for each brain region were obtained by taking the 2.5% and 97.5% cutoff of the bootstrap sampling distribution of mean differences in ISCs.”* (p. 9 and p. 13 of the revised Methods)

There were no effect sizes or degrees of freedom associated with permutation testing, but we report effect sizes in the form of beta coefficients for the results of multilevel regression analyses in Supplementary Figures 12-15 and describe our procedure for determining degrees of freedom for multilevel regression models in the supplement (p. 25 of the supplement).

Methods

R1.3) The paper would benefit from greater clarity in the way the methods (specifically the social distance) are discussed in the paper itself. The methods are clear in the methods section, but they are described inconsistently in the rest of the paper, especially the results. For example, social distance is often described as if it were a continuous variable (i.e., “distance in social network”) when it is actually a series of binary comparisons (e.g., whether or not they became friends (i.e., 1 vs levels 2 or 3), or 1 vs. 3, etc.) I don’t see an issue with using binary comparisons, but I think it would be clearer to bring the description of the variable in line with the nature of the variable.

Thank you for your suggestion. Throughout the paper, whenever appropriate, we have modified descriptions of social distances or changes in social distances to reflect the nature of our analytic methods (i.e., binary comparisons). Specifically, we have added the following text to the revised manuscript:

- *“As such, our primary analyses involve comparing different levels of social distance and the direction of changes in social distance to test if Time 1 neural similarities predict (1) friendship at our latest time point (Time 3; 8 months later) and (2) increases, rather than decreases, in social closeness over time (i.e., in the 6 months between Times 2 and 3).”* (p. 7 of the revised manuscript)
- *“Additionally, for each pair of fMRI participants, we determined the direction of their change in social distance over time (i.e., from Time 2 to Time 3) and tested if (and in which brain region(s)) individuals who grew closer in social ties over time had exhibited greater initial neural similarity before meeting one another relative to individuals who did not grow closer over time...”* (p. 7 of the revised manuscript)
- *“More specifically, in each brain region, we tested if individuals who came to be characterized by a social distance of 1 (i.e., friends) 8 months into the academic program had greater neural similarity before meeting one another than individuals characterized by greater social distances (i.e., friends-of-friends and friends-of-friends-of-friends)”.* (p. 7 of the revised manuscript)
- *“In this set of analyses, we compared pre-existing neural similarities among friends to those of dyads that came to be characterized by different levels of social distance at Time 3. We first*

tested whether pre-existing neural similarities among individuals who came to be characterized by a social distance of 1 (i.e., who were friends) 8 months later were greater than pre-existing neural similarities among individuals who did not.” (pp. 8-9 of the revised manuscript)

R1.4) How were non-reciprocal friendships dealt with? If reciprocity was .53-.56, this suggests that there were a substantial number of non-reciprocal friendship dyads where one person designated the other as a friend but not vice versa. The authors state “Pairs of individuals who both named one another as friends were assigned a social distance of one. Individuals would be assigned a distance of two from one another if they had a mutually reported friendship with a shared friend, but were not friends with one another, and so on.” How were dyads coded if the selection was not mutual?

Thank you for giving us a chance to clarify this. In this paper, we made the choice to focus exclusively on reciprocally reported friendships to maximize the likelihood that each “friendship” in the dataset reflects a true relationship among members of the dyad. Specifically, there only exists a tie between members of a dyad if they both nominate one another as friends and the social distance measures were derived from this social network comprised of reciprocal friendships. Thus, for a given dyad, if the selection was not mutual, they were not coded as friends and would be assigned a social distance greater than 1.

To ensure that this point comes across clearly to readers, we have added the following underlined text in the Methods section: *“An unweighted graph consisting of only mutually reported social ties was used to estimate social distances between individuals. In other words, an undirected edge would connect two actors only if they had both nominated one another as friends, and no edge would connect the two actors if neither nominated one another as friends or if only one actor nominated the other as their friend.”* (p. 6 of the revised Methods section)

R1.5) “Dyads were divided into three categories depending on if they grew closer over time (i.e., decrease in social distance), grew apart over time (i.e., increase in social distance), or remained at the same social distance.” - why was distance not examined continuously? I don’t meant to imply that this approach would be superior, but merely that the authors should explain their decision-making.

Thank you for bringing up this point and giving us an opportunity to explain why social distance change was binned into three categories. When examining how neural similarity may predict change in social distance over time, we believe that dividing social distance change into three categories (i.e., growing closer, growing apart, and no change) reflects distinctions that are qualitatively meaningful to our research question of interest. We note that relatively large increases in social distance from Time 2 to Time 3 were only possible for a very small subset of dyads, since, for example, only 5% of dyads were characterized by a social distance of 4 at Time 2. Given this, as well as the fact that extraneous factors (e.g., who one happens to live nearby or sit close to in class during the first few weeks of the program) are likely to shape social distances early on in the academic program (as discussed in the Discussion section and consistent with a relatively weaker relationship between similarity and friendship at Time 2 compared to Time 3), we reasoned that relatively granular distinctions (e.g., between decreases in social distance of 3 vs. 2 “degrees of separation”) would be less theoretically meaningful than the direction of the distance change. Therefore, we decided to bin dyads in terms of the direction of social distance change, rather than taking into account more granular changes in social distance. In addition, as the majority of the social

distance between dyads remains unchanged (i.e., 445 out of 820 dyads had a social distance change of 0, as shown in the figure below), categorizing social distance change increases the power of our analysis.

As suggested, we have added text to the revised Methods section to explain our thinking on this topic: *“Given that (1) extensive changes in social distance from Time 2 to Time 3 were only possible for a small number of dyads (e.g., only 5% of dyads were far enough apart in the social network at Time 2 to be able to decrease in social distance by 3 ‘degrees of separation’ by Time 3), and (2) extraneous and situational factors may have an especially large impact on these early social distance values, as mentioned in the Discussion section, we focused on overall direction of social distance change (e.g., if participants within a given dyad decreased in social distance over time), rather than more granular variability in social distance changes.”* (p. 11 of the revised Methods section)

We also ran additional analyses using a linear mixed model with crossed random effects in which change in social distance was examined continuously. Using this approach, there were two parcels that exhibited significant links between pre-existing neural similarity and future changes in social distance ($p < .05$ after FDR correction; see Supplementary Figure 14). There was also a trending negative relationship between neural similarity and change in social distance across the majority of the brain parcels such that higher neural similarity is associated with a decrease in social distance (see Supplementary Figure 15).

Supplementary Figure 14. Pre-existing neural similarity in regions of the ventromedial prefrontal cortex and lateral temporal cortex predicted decreases in social distance over time. There existed a trending negative

association between pre-existing neural similarity and continuous change in social distance across brain regions. β is the standardized regression coefficient from the linear mixed effect model with crossed random effects for participants. Regions outlined in black demonstrated where significant associations ($p < 0.05$, FDR-corrected) were observed between continuous change in social distance and ISC.

Supplementary Figure 15. Pre-existing neural similarity across brain regions spanning multiple brain networks showed a trending negative association with change in social distance over time. Regression coefficients for each brain parcel from the linear mixed model examining the relationship between pre-existing neural similarity and continuous change in social distance, grouped by their corresponding brain networks, are shown.

Discussion

R1.6) I found the discussion thoughtful. It did leave me wondering why the authors found that only the left OFC predicted friendship formation, whereas similarity in so many other brain regions predicted shrinking social distance over time. I love the interpretation of the latter finding, but why would we see that so many brain regions would predict increased/decreased social distance over time, but not initial friendship formation (why only the left OFC for the first finding)? Is this related to the conclusion that “friendships born out of mere circumstance may have dissolved due to interpersonal incompatibilities, reflected in pre-existing neural dissimilarities, which only became apparent with the extended passage of time”?

Thank you for your thoughtful comments regarding the finding that the left OFC predicts friendship formation.

One possible set of reasons why we might see more extensive results in our second set of analyses (looking at social distance changes among dyads in the fMRI subsample) compared with our first set of analyses (comparing friend dyads to more distant dyads within the fMRI subsample) pertains to differences in the range of phenomena that each data analytic approach is able to capture. For instance,

consider the possibility that there is a general tendency for more similar people to later become friends, but sometimes various factors disrupt the formation of friendships between similar individuals (e.g., not encountering one another sufficiently frequently during the 8 months of the study to become friends, limitations on one or both individuals' available time or general interest in socializing, other factors that might lead to incompatibility). If, for example, Persons A, B, and C were all quite similar to one another before entering the academic program, but one or more of the factors listed above (or other factors) led Person A and Person B *not* to become friends, despite their similarity, Persons A and B would still be relatively likely to become friends with other similar people, such as Person C (who is similar to both Person A and Person B). This could lead Person A and Person B, who in this example would have high pre-existing neural similarity, to decrease in social distance over time (e.g., going from a social distance of 3 or more at Time 2 to a social distance of 2 at Time 3). Our second set of analyses would be able to capture this, as it would present as a relationship between pre-existing neural similarity and future decreases in social distance between Persons A and B, even if Person C, who indirectly connects Persons A and B at Time 3 in this example, were among the 245 students who were in the social network sample but did not participate in the fMRI study. However, our first set of analyses would not be able to capture this kind of phenomenon.

Along the same lines, consider another scenario where two fMRI participants, Persons D and E, were indirectly connected by a mutual friendship with Person F (who was also among the 245 students in the social network sample but not the fMRI subsample) at Time 2, but either one or both of Persons D and E's friendship(s) with Person F were merely borne out of circumstance and dissolved by Time 3 due to incompatibilities that only became apparent with time, reflected in pre-existing neural dissimilarities between Person F and Persons D and/or E. Again, our second set of analyses, but not our first, would be sensitive to this kind of phenomenon, as it would contribute to the association between social distance change and pre-existing neural similarity that we observe in our second set of analyses: Persons D and E would have low pre-existing neural similarity and would increase in social distance over time (in contrast to Persons A and B in the previous example, who would have high pre-existing neural similarity and would decrease in social distance over time). However, our first set of analyses, which focus on how friends within the neuroimaging sample compare to other dyads within the neuroimaging subsample, would not be sensitive to this sort of phenomenon.

To succinctly convey these points, we have added the following to the revised Discussion section: *“A more extensive set of brain regions was implicated in our second set of analyses (which focused on changes in social distance over time) compared to our first set of analyses (which compared friends to other dyads within the fMRI sample). This may be due in part to the fact that the second set of analyses are sensitive to changes in dyadic social distance over time that are driven by the formation or dissolution of friendships beyond the dyad, including those where one or both people involved in a friendship did not complete the fMRI study, whereas the first set of analyses would not be sensitive to such phenomena, in part because it compares friends within the fMRI subsample to other dyads within the fMRI subsample. It is possible that future studies in which a larger proportion of the social network undergoes neuroimaging would reveal associations between friendship formation and neural similarity in a more extensive set of brain regions.”* (pp. 19-20 of the revised manuscript)

We would be happy to revise the paper to extend our discussion of this point if you believe that it would further improve the manuscript.

R1.7) I understand that examining social influence may lie outside of the purview of the current paper, but the authors imply that they do not have the data to examine social influence processes because they were not able to collect data two years later. Would they not be able to examine social influence processes with their current data? For example, the authors could test whether one person's neural processing changed more/ became more similar to the other person's (whereas the other person's may have stayed the same, suggesting they influenced the other)?

Due to the COVID-19 pandemic, the current dataset only included one wave of neural data collected at Time 1. A second wave of neuroimaging data collection was planned for 2 years after Time 1, but it was unfortunately not possible to collect the second wave of neuroimaging data due to the suspension of in-person learning and research early on during the pandemic. Thus, it is impossible to assess the change in neural processing to examine the social influence processes with the current data.

More Minor

R1.8) This paragraph is redundant (analyses and results stated twice - I would recommend starting the paragraph at the word "Next" and moving the references to Methods and Supplemental Figures). "In exploratory analyses, we investigated if accounting for inter-individual similarities in ratings of enjoyment and interest in the stimuli would fully account for and/or significantly diminish the difference in neural similarity between friends and pairs of individuals characterized by a social distance of 3 (see Methods). The effect in the OFC remained significant when controlling for inter-individual similarities in ratings of enjoyment and interest (Supplementary Fig. 6), suggesting that similarity in neural responding in this brain region captured similarities in friends' responses to the stimuli that the interest and enjoyment ratings did not. Next, to inform our interpretation of the psychological meaning of the OFC result, we tested if controlling for inter-individual similarities in ratings of enjoyment and/or interest would significantly diminish the difference in neural similarity between friends and pairs of individuals characterized by a social distance of 3. Neither controlling for inter-individual similarity in interest ratings nor controlling for inter-individual similarity in enjoyment ratings significantly decreased the difference in neural similarity between friends and individuals characterized by a social distance of 3.

We apologize for the lack of clarity regarding these analyses in the original version of the manuscript. The paragraph in question summarizes two distinct but related (and similar-sounding) sets of analyses. We now realize that we had previously only explicitly articulated the rationale for doing both of these sets of analyses in the Methods section (*"Even when a similar pattern of results is observed with and without controlling for similarities in behavioral ratings, it is possible that the magnitude of the neural similarity difference between groups would significantly decrease when controlling for such behavioral similarities if the neural similarities are partially driven by them"* (p. 10 of the Methods section)). We have expanded our description of these analyses within the aforementioned paragraph of the main text to clarify this:

"In exploratory analyses, we investigated if accounting for inter-individual similarities in ratings of enjoyment and interest in the stimuli would fully account for and/or significantly diminish the difference

in neural similarity between friends and pairs of individuals characterized by a social distance of 3 (see Methods). In our first set of such analyses, which tested if the difference in neural similarity between friends and pairs of individuals characterized by a social distance of 3 could be fully accounted for by inter-individual similarities in ratings of enjoyment and interest in the stimuli, the effect in the OFC remained significant when controlling for inter-individual similarities in ratings of enjoyment and interest (95% bootstrap CI: [0.290, 0.994]; Supplementary Fig. 7). This suggests that similarity in neural responding in this brain region captured, at least in part, similarities in friends' responses to the stimuli that the interest and enjoyment ratings did not. Even so, while a similar pattern of results was observed with and without controlling for similarities in these behavioral ratings, it is possible that the observed neural differences between groups of dyads might have been partly but significantly driven by participants' interest in and/or enjoyment of the stimuli. If this were the case, we would expect the magnitude of the neural similarity difference between groups to significantly decrease when controlling for such behavioral similarities. Thus, to inform our interpretation of the psychological meaning of the OFC result, we tested if controlling for inter-individual similarities in ratings of either enjoyment or interest would significantly diminish the difference in neural similarity between friends and pairs of individuals characterized by a social distance of 3 (see Methods). Neither controlling for inter-individual similarity in interest ratings nor controlling for inter-individual similarity in enjoyment ratings significantly decreased the difference in neural similarity between friends and individuals characterized by a social distance of 3.” (pp. 9-10 of the revised manuscript)

R1.9) “when controlling for enjoyment and interest ratings (Supplementary Fig. 9).” Is redundant with: “if accounting for inter-individual similarities in ratings of enjoyment and interest in the stimuli would significantly diminish the difference in neural similarity between these two groups (see Methods). This analysis was repeated for each of the brain regions in which we observed a significant difference in mean neural similarity (i.e., regions outlined in black in Fig. 6). Controlling for inter-individual similarities in ratings of enjoyment of the stimuli significantly decreased the extent to which individuals who grew closer over time exhibited higher neural similarity in a portion of right superior parietal cortex (Difference in mean normalized neural similarity = 0.015; $p < 0.05$) relative to individuals who grew apart over time. However, this effect was not observed when controlling for inter-individual similarities in ratings of interest in the stimuli or in other brain regions.”

As indicated in our response to the immediately previous point (R1.8), we regret not having made it sufficiently clear in the prior version of the manuscript that these statements are referring to two distinct but related sets of analyses. We hope that the changes noted in response to R1.8 (which refer to analogous analyses involving comparisons of dyads based on social distance levels rather than direction of social distance change) help to clarify this. In addition, we have added the following text to the paragraph quoted in the second part of this comment to help clarify the distinction between the two different kinds of analyses performed involving enjoyment and interest ratings and the rationale for doing both: “We reasoned that even though similar patterns of results were obtained with and without statistically accounting for inter-subject similarities in enjoyment and interest ratings, it is possible the observed group differences in ISCs in some brain regions might have been partially (and significantly) driven by what participants found enjoyable or interesting. Thus, this analysis was conducted with the goal of informing interpretations of the observed results.” (p. 14 of the revised manuscript)

R1.10) “Controlling for inter-individual similarities” ... “The one exception was...” I wouldn’t make too much of this - could be noise given the number of analyses conducted.

Thank you for the suggestion. The information in the sentence that previously started with “*The one exception was...*” is now integrated into the parentheses of the preceding sentence to avoid over-emphasizing this result: “*This was observed in nearly every brain region (besides the right superior parietal cortex, where accounting for enjoyment ratings decreased the difference in neural similarity between pairs of people who grew closer and those who drifted apart) in which individuals who grew closer over time exhibited exceptionally higher pre-existing neural similarity.*” (p. 19 of the revised manuscript)

I hope these suggestions help the authors improve the paper. Again, I believe this paper makes a highly valuable contribution to our understanding of human behavior.

Thank you so much for your kind words and for your suggestions to help improve the paper.

Responses to Comments from Reviewer #2:

"Neural Similarity Induces Friendship" is a fascinating manuscript, drawing from sources in psychology, neuroscience, and sociology. The manuscript appropriately employs Ronald Burt's network tie-eliciting method for obtaining information about MBA students' informal social connections to one another at three moments in time during the first year of the graduate program. Strikingly, and this is a matter for further comment, the authors report gathering complete network data for an entire student population numbering in the hundreds, not just once but multiple times.

It is not new for attitudinal homophily or personality homophily (or, for that matter, demographic or cultural homophily) to be discussed; the empirical establishment of the pattern that "birds of a feather flock together," especially in demographic characteristics, is a solid finding of the social network and social-psychological literature. However, what the authors offer here is some evidence of functional involvement of particular areas of the human brain, through which sociocultural similarity may produce observable behavioral acts to maintain and preserve some social ties. This possibility is a novel and important contribution.

Thank you for your encouraging comments on our manuscript.

R2.1) I use the word "may" rather than "does" intentionally. The authors assert that alignment of neural characteristics between student dyads in an MBA program "determines," "fosters," "induces," or "causes" the creation or withering of social ties between those dyads. These are strong causal verbs, and the authors refer to their longitudinal study design to justify the strength of their claim. However, the study design is not one that controls for multiple possible confounding social factors. The study admirably shows that neural similarity exists prior to social tie formation, and that time order is important. However, the study design does not restrict variation to the measured aspects of brain activity alone or completely control for other personal or social factors; the authors acknowledge the presence of considerable social diversity among the students being studied, some dimensions of which appear to be measured and some dimensions of which are surely not.

To provide one possible example of a confounding correlation, the authors note in passing that MBA students live and eat close to one another to describe the stimulating environment for tie formation. But surely not all 288 of these students live in the same floor on the same building, and surely they don't all eat at the same table. Given well-known tendencies toward segregation in residence, dining, and other functions of educational institutions, could tie formation and tie dissolution during the time period studied be connected to greater proximity of more sociodemographically similar students, who it just so happens may (due to the localization of culture sociodemographically) have the "generalized shared reality" that would lead to neural similarities? If so, then the neural similarities would be a side effect of a dynamic driven primarily by social availability, only appearing to be the factor that "induces friendship" because it is strongly associated as a variable with the underlying sociological driver, segregated availability. Although we are living in the 21st Century, it is too soon to declare that racial, national, gender, birth cohort, religious, linguistic, disciplinary, and other dimensions of educational segregation have been eliminated.

If the main driver of similarity in association is indeed psychological, perhaps the precedence of neural similarity among those who will form and maintain ties in the future is due to another psychological mechanism, such as inter-category animosity. Animosity might lead to segregation, which would in turn could lead on the one hand to reinforcement of cultural differences that became visible in the kind of neural differences that the researchers observe, and on the other hand to the tie formation that the researchers observe. The neural association with association would in this case be spurious.

Regarding these two possibilities, the authors in supplemental files include two figures showing findings that claim to "control" for "similarities in demographics," the method by which this occurs, or for what demographic categories, is not described in the main paper or any supplements. Regardless, it is quite unlikely that all relevant sociodemographic dimensions of segregation that shape opportunities for contact and tie reinforcement have been measured, since there are simply so many of these dimensions that exist.

Thank you for bringing up these important considerations. Although the majority of neuroimaging data was collected within days (Mode = 1 day, Median = 3 days) of participants' arrival on campus (Supplementary Fig. 1) to limit the potential confound that neural similarity observed at baseline may result from any social interactions after their arrival on campus, including interactions between socio-demographically proximal or similar individuals, although participants were subject to substantial random assignment regarding whom they lived with or nearby and whom they studied with, and although supplementary analyses showed that the observed pattern of results was similar when excluding dyads who had prior encounters before the fMRI study and when statistically accounting for the demographic variables that we measured, it is still possible that some unmeasured sociodemographic similarities led to both neural similarities and subsequent friendship formation. Indeed, we expect that individual differences in neural processing are impacted by a wide array of sociodemographic variables and other factors, which impact how people respond to the world around them, and in turn, their future friendships. Therefore, we have revised the manuscript to discuss this possibility, to scale-back causal claims regarding relationships between our variables, to clarify what demographic variables were measured and incorporated into analyses, to include more information regarding the role of random assignment in determining with whom and where participants lived and with whom participants studied, and to suggest a possible causal model (which a body of future research will be needed to test), in line with what the reviewer suggests here, whereby wide-ranging sociodemographic variables and other factors shape neural processing over the course of one's life, which in turn, impacts the social connections that one makes, as described in more detail below.

First, we regret that the previous version of the manuscript omitted important relevant details regarding the substantial amount of random assignment to which participants were subjected with respect to both (1) whom they took classes and worked on group projects with (as part of the academic program) and (2) whom they lived with and near (as a result of the lottery system employed by on-campus housing). Thus, we have added the following text to the Methods section to clarify the random assignment of participants into study groups and housing units: "*As part of the graduate program's aim to help students build a professional network, students were randomly assigned to study groups using stratified random sampling, such that students completed all of their coursework with the same group of randomly assigned classmates in each term. Additionally, nearly all students applied to on-campus housing and were randomly assigned to housing units based on a lottery system.*" (p. 2 of the revised Methods section)

Second, in places where we described the results of the supplementary analyses that controlled for demographic characteristics, we have included the text “(i.e., age, gender, and nationality)” (see pp. 9 and 13 of the revised manuscript and captions in Supplementary Figs. 4 and 7). We have also added the following text to clarify how these findings “control” for similarities in demographic variables: “We repeated these analyses while controlling for inter-subject similarities in the demographic variables of age, gender, and nationality. More specifically, to statistically account for demographic similarities that may be related to similarities in neural responding and/or social network proximity, we regressed out inter-subject similarities in age, gender, and nationality from ISCs in each brain region before repeating the analyses described above.” (pp. 9 and 13 of the revised Methods section)

Third, we fully agree with the Reviewer that “*what the authors offer here is some evidence of functional involvement of particular areas of the human brain, through which sociocultural similarity may produce observable behavioral acts to maintain and preserve some social ties.*” We agree that it is likely that similarities in some unmeasured sociodemographic variable(s) lead to both similar ways of processing the world and a greater propensity to befriend one another. We have now tried to clarify (1) that these functional brain patterns are indeed likely to be the product of myriad, complex, and interacting influences, including but not limited to those that the Reviewer mentions, and (2) our thinking, which is consistent with what the Reviewer posits here, regarding how these variables might impact one another (and indicate that substantial additional research is needed to test for and elucidate causal relationships between these variables). More specifically, we have added the following text to the revised discussion section:

“Relatedly, although we attempted to statistically account for intersubject similarities in some sociodemographic variables in our analyses, it is likely that some unmeasured sociodemographic variables (particularly those related to similar cultural perspectives) causally contribute to both pre-existing neural similarities between participants and subsequent friendship formation. Our results are consistent with a causal model in which a wide array of sociodemographic variables and other factors, including, but not limited to, those that we were able to measure here (such as geographic proximity, race, nationality, gender, birth cohort, religion, linguistic background, educational background, parenting, political beliefs, genes, gene-environment interactions, and so on) give rise to sets of experiences, expectations, and social feedback that in turn give rise to particular ways of responding to the world, which are reflected in neural activity and when similar, facilitate future friendship.

Of note, the current results demonstrate that even without knowing or measuring all of these possibly innumerable sociodemographic variables and other factors, it is possible to measure one of their likely outcomes (neural similarity), which predicts future friendship. Although the question of the precise causes of such neural similarity may be intractable (given that there are likely numerous interactive causes, whose relative importance may vary across contexts), the current findings suggest that neural similarity predicts both future friendship formation and growing closer to particular people over time. We hope that over time, future research can build upon the current work to elucidate what factors contribute to similar ways of processing the world. Although we do not expect any single study to be able to fully address this question, it will be valuable for the field in general to better understand how sociocultural (and other) forces shape neural activity.” (pp. 22-23 of the revised manuscript)

Fourth, we have also revised phrasing throughout the manuscript to avoid making strong causal claims based on the current results. Where causal relationships are discussed, they are now discussed merely as possibilities that remain to be fully tested instead of making causal inferences throughout the paper. Such changes are listed below (new text is underlined):

- *“These results provide evidence for neural homophily, such that pre-existing neural similarities may foster friendship and increased social closeness.”* (p. 2 of the revised manuscript)
- *“The current research expands upon this work and sheds light on the potential ~~causal~~ role of neural homophily in shaping human social structures over time.”* (p. 24 of the revised manuscript)
- We have also added the following underlined text in the Discussion section acknowledging that the current study cannot support decisive causal claims, and encouraging future work to further elucidate the phenomena studied here: *“More broadly, despite the substantial degree of random assignment to which participants were subjected regarding with whom and where they lived, and regarding with whom they studied and took classes (see Methods), we note that this study is inherently limited in its capacity to support decisive causal claims, given that it was an observational study. Thus, while the current findings are consistent with the causal model posited above, in which innumerable demographic and biological factors give rise to neural similarities that in turn facilitate friendship, more research is needed to better understand how relationships among these phenomena come to be, as well as potential moderating factors.”* (pp. 23-24 of the revised manuscript)

We would be happy to revise our phrasing further or to expand on any of these points if needed.

R2.2) Finally, the accurate establishment of clear social network distances, especially beyond a distance of 2, can be threatened by missing data, with multiple missing ties for each person who doesn't report their ties. It is fortunate in a methodological sense, then, that the authors report obtaining network data for every single one of the 288 MBA students at Time 2, and for all but 1 MBA student at Time 3. Given the typically low response rates to study solicitations, this is odd. The methods section regarding the 43 fMRI participants states explicitly that data collection was in accordance with the local ethical review board. The 288 MBA students universally completing a survey reporting personal information about themselves and individuals and their social choices among a set of pre-professional peers is a strange accomplishment. Were students required by educational officials in the MBA program to participate, either explicitly or as a way to curry favor? If so, how is that consistent with the principle established in federal HHS guidelines that participation in human subjects research be voluntary and be driven by informed affirmative consent? If all 288 student freely volunteered once, and 287 freely volunteered twice, that's a considerable accomplishment. Some clarification of this process would be appreciated.

Thank you for pointing out that it would be helpful to clarify the recruitment process. We have added the following to the revised Methods section: *“All students in the first-year graduate cohort enrolled in a leadership course within the graduate program, and they were invited to participate in the online social network survey as an optional part of their coursework on leadership. To encourage participation, students who completed the survey were provided with personalized, educational feedback about their social networks based on their survey responses, but they were reminded that participation was voluntary and would not affect their performance in the course.”* (p. 2 of the revised Methods section)

We would be happy to further expand on any aspects of the procedure if it would be helpful.

Responses to Comments from Reviewer #3:

Hyon and colleagues use a naturalistic longitudinal design to show that pre-existing neural similarity (measured during film-viewing) is related to future friendship status (social distance) 8 months later. The authors find a significant difference in pre-existing neural similarity (measured at the beginning of the school year) between subjects who eight months later became friends (social distance 1) and those who were friends-of-friends-of-friends (social distance 3) in orbitofrontal cortex (OFC). They also find that the dyads whose social distance decreased (they became closer friends) between 2 months and 8 months post-scan were associated with higher pre-existing neural similarity in a variety of brain areas. This is a concise study—using a very neat longitudinal design!—on the very exciting topic of neural homophily and real-world friendship. The results are a bit mixed, and I think there are a couple places where the authors could do a bit more work to build the reader’s confidence. There were also several places in the narrative of the Results section where I had a hard time following the logic of analyses. I articulate some comments below that I hope will strengthen the manuscript.

Thank you for your encouraging words about the study.

Comments:

R3.1) The authors organize the data for statistical tests in particular ways, and my main concern is that readers will worry that these effects might hold up if you slice the data differently. I don’t think the slicing is actually that arbitrary, but it doesn’t seem well-motivated or thoroughly explained while reading the Results section. For example, a reader might worry that $N = 43$ is a relatively small sample size, without fully appreciating that the fMRI data had to be collected as quickly as possible as soon as subjects arrived on campus to minimize the chance that they would start interacting with each other and becoming friends. A little more motivation and explanation here would go a long way.

Thank you for your thoughtful comment and suggestion. We have worked to address the points raised in this comment in three key ways.

First, we have revised the manuscript to provide more explicit rationale regarding the slicing of the dataset. For example, in our second set of analyses, social distance change was binned into three categories (i.e., dyads that grew closer, grew further apart, and did not change in terms of social distance from Time 2 to Time 3) that we believe reflect distinctions that are qualitatively meaningful to our research question. We note that relatively large increases in social distance from Time 2 to Time 3 were only possible for a very small subset of dyads, since, for example, only 5% of dyads were characterized by a social distance of 4 at Time 2. Given this, as well as the fact that extraneous factors (e.g., who one happens to live nearby or sit close to in class during the first few weeks of the program) are likely to shape social distances early on in the academic program (as discussed in the Discussion section), we reasoned that relatively granular distinctions (e.g., between decreases in social distance of 3 vs. 2 “degrees of separation”) would be less theoretically meaningful than the direction of the distance change. Therefore, we decided to bin dyads in terms of the direction of social distance change, rather than taking into account more granular changes in social distance. We have added the following text to the revised Methods section to explain our rationale:

“Given that (1) extensive changes in social distance from Time 2 to Time 3 were only possible for a small number of dyads (e.g., only 5% of dyads were far enough apart in the social network at Time 2 to be able to decrease in social distance by 3 ‘degrees of separation’ by Time 3), and (2) extraneous and situational factors may have an especially large impact on these early social distance values, as mentioned in the Discussion section, we focused on overall direction of social distance change (e.g., if participants within a given dyad decreased in social distance over time), rather than more granular variability in social distance changes.” (p. 11 of the revised Methods section)

We have also worked to clarify how levels of social distance were calculated and thus, how and why data was binned in the manner that it was in our first set of analyses, as discussed in more detail in our response to the next comment (R3.2).

Second, as discussed in our response to one of Reviewer 1’s points (R1.5) earlier in this document, we have also added additional analyses using linear mixed models with crossed random effects for participants, in which change in social distance was examined continuously. In those analyses, two parcels displayed significant relationships between pre-existing neural similarity and social distance change in the expected direction at an FDR-corrected significance threshold of $p < .05$ and no parcels had significant results in the unexpected direction (see Supplementary Figure 14). We also note that there was a trending relationship between pre-existing neural similarity and change in social distance in the expected direction across the majority of brain parcels such that higher pre-existing neural similarity was associated with future decreases in social distance (see Supplementary Figure 15).

Supplementary Figure 14. Pre-existing neural similarity in regions of the ventromedial prefrontal cortex and lateral temporal cortex predicted decreases in social distance over time. There existed a trending negative association between pre-existing neural similarity and continuous change in social distance across brain regions. β is the standardized regression coefficient from the linear mixed effect model with crossed random effects for participants. Regions outlined in black demonstrated where significant associations ($p < 0.05$, FDR-corrected) were observed between continuous change in social distance and ISC.

Supplementary Figure 15. Pre-existing neural similarity across brain regions spanning multiple brain networks showed a trending negative association with change in social distance over time. Regression coefficients for each brain parcel from the linear mixed model examining the relationship between pre-existing neural similarity and continuous change in social distance, grouped by their corresponding brain networks, are shown.

Third, in addition to including the results described above and more explicitly explaining the rationale for categorizing data in the way that we did, we have also followed the reviewer’s suggestion to more fully explain the constraints on sample size for the fMRI study. More specifically, we have added the following text to the Discussion section:

“We also note that in the current study, fMRI data had to be collected within a very short time frame (i.e., as soon as participants arrived on campus; as described in the Methods, the modal number of days between participants’ arrival on campus and their participation in the fMRI study was 1 and the median number of days was 3) to minimize participants’ opportunities to begin interacting with and befriending one another beforehand. This constraint, particularly in combination with other logistical limitations (e.g., instrument availability during participants’ initial arrival period) imposed a limitation on sample size. Future studies would benefit from examining the current phenomena in larger samples.” (pp. 21-22 of the revised manuscript)

We have also added the following text in the Results section to further highlight the purpose of scanning participants shortly after their arrival on campus: *“The majority of fMRI participants were scanned shortly after their arrival on campus to limit the opportunities that participants may have to interact with one another extensively and befriend one another (see Fig. 1 and Supplementary Fig. 1).”* (pp. 4-5 of the revised manuscript)

R3.2) In another example, I think we need a little more motivation / explanation in the Results as to how you arrived at social distance groups 1, 2, and 3. For example, how do you determine the threshold between level 2 “friends-of-friends” and level 3 “friends-of-friends-of-friends”? Are these groups of dyads similar in size? Is there a more continuous metric on the friendship network graph that might provide corroborative results? In Figure 3, I would also describe what these social distance groups mean and how dyads are assigned to these groups in the caption.

Thank you for giving us an opportunity to further explain how the social distance groups were defined. As mentioned above in our response to Reviewer 1 (i.e., response to R1.4), we focused exclusively on reciprocal friendship to ensure that there is a strong and meaningful relationship between each dyad. Specifically, the social network was characterized such that there only existed a tie between the dyad if they both nominated one another as friends (i.e., if the selection was not mutual, they were not coded as friends and would be assigned a social distance greater than 1), and the social distance measures were derived from this social network comprised of reciprocal friendship. Therefore, dyads had a social distance of 1 if they both nominated each other as friends, dyads had a social distance of 2 if they didn’t have a reciprocal friendship between them but shared a mutual friend (i.e., friends-of-friends), and dyads had a social distance of 3 if they were not friends themselves but had friends who were friends (i.e., friends-of-friends-of-friends). We include the information about the sizes of these social distance groups in the results section: *“There were 93 dyads with a social distance of 1 (i.e., friends), 544 dyads with a social distance of 2 (i.e., friends-of-friends), and 183 dyads with a social distance of 3 (i.e., friends-of-friends-of-friends)”* (p. 8 of the revised manuscript). To further clarify how edges in the network were defined, we have revised relevant text in the Methods section (new text is underlined): *“An unweighted graph consisting of only mutually reported social ties was used to estimate social distances between individuals. In other words, an undirected edge would connect two actors only if they had both nominated one another as friends, and no edge would connect the two actors if neither nominated one another as friends or if only one actor nominated the other as their friend.”* (p. 6 of the revised Methods section)

As suggested, we have also added the following sentence in the Figure 3 caption to clarify the social distance metric: *“Social distance, ranging from 1 to 3, is the geodesic distance between a pair of individuals in the social network characterized by reciprocal friendship ties.”* (p. 8 of the revised manuscript).

We made a deliberate choice to not use weighted edges to estimate distance in the current study, as tools to calculate distances in weighted networks (e.g., Dijkstra’s algorithm; Dijkstra, 1959) may potentially lead to characterizations that do not mesh well with our conceptualizations and hypotheses (e.g., by making indirectly connected people appear closer than directly connected people; see Box 1 of Baek et al., 2021). Therefore, given our research questions, we opted to simply use a stringent threshold for defining friendship by characterizing a friendship tie between two individuals only if the friendship was

mutually reported and then calculated the geodesic distances based on the characterized unweighted network of reciprocal friendship ties.

R3.3) In one of the core results, the authors find a significant difference in neural similarity between friends (social distance 1) and friends-of-friends-of-friends (social distance 3) in an orbitofrontal cortex (OFC) parcel. This is a difficult part of the brain to capture with fMRI due to susceptibility artifacts—i.e. due to the air/bone boundary in the nearby sinuses—and signal here may be heavily affected by head motion (although 25 ms TE might help!). Can the authors provide any supplementary analyses to increase readers' confidence that we actually have reliable signal in this OFC parcel? For example, how correlated is the time series in this parcel with framewise displacement (FD)? Does preprocessing / nuisance regression mitigate this correlation? Another example: Do we see substantial ISC values in this parcel? This would suggest that there is in fact stimulus-driven signal in this area that's reliable across subjects. Last example: Does regressing out subject-level FD in the group-level test impact this result? (I understand this is tricky because you're dealing with dyads, not individual subjects, here.)

Thank you for bringing up this point. To show that there is reliable signal in this OFC parcel, we first assessed if there were significant ISC values in this parcel. Consistent with the notion that there is indeed stimulus-driven signal in this parcel that is reliable across participants, testing ISC values against a null distribution generated using bootstrap hypothesis testing indicated significant ISCs in this region, $ISC_{mean} = 0.073$, $p < 0.001$.

To further examine the relationship between ISCs in this parcel with head motion, we looked at the mean correlations across participants between each of the 200 brain parcels and framewise displacement (FD). Compared to other parcels, this parcel in the OFC did not correlate especially strongly with FD (see figure below for the distribution of the mean correlation between 200 brain parcels and FD among all fMRI participants; the dotted line indicates this parcel's mean correlation of .047 with FD, which was on the lower end of the distribution).

We also conducted additional analyses to further ensure that the link between ISCs in the OFC and social distance in our main analysis could not be attributable merely to head motion. We repeated the same permutation testing procedure used in the main analyses to look at the relationship between dyadic similarities in FD (where dyadic similarities in FD were characterized both in terms of inter-subject correlations of FD time series and in terms of the absolute difference in mean FD between members of each dyad) and dependent variables of interest. FD similarities did not significantly differ between friends and friends-of-friends at 8 months ($p = .766$ when characterizing dyadic similarities in FD in terms of inter-subject correlations of FD time series; $p = .862$ when characterizing dyadic similarities in FD in terms of absolute difference in mean FD between members of each dyad). FD similarities also did not significantly differ between dyads who grew closer versus farther apart in social distance over time ($p = .944$ when characterizing dyadic similarities in FD in terms of inter-subject correlations of FD time series; $p = .434$ when characterizing dyadic similarities in FD in terms of absolute difference in mean FD between members of each dyad).

R3.4) What's the motivation for z-scoring similarities across dyads within each region? I assume this is why the similarity values are generally centered around zero in Figures 3 and 5, and why we might (incorrectly) think the cool values at social distance 3 correspond to negative similarities. Does this change how we interpret the magnitudes across parcels within a group? Was this transformation performed only for visualization? I think it would be helpful to plot the “raw” (not parcelwise z-scored across dyads) ISCs in the Supplementary Materials.

The motivation to z-score similarities across dyads within each region is to use the same color scale for all parcels while still facilitating visual comparison across social distance groups. We reasoned that since the range of ISCs differs substantially across brain regions, if ISC values were not first z-scored across dyads within brain region, differences in colors within figures would primarily reflect general differences in ISC magnitude across brain regions, rather than differences in ISCs between groups of dyads within brain regions, which are much more central to our research question. Additionally, this z-scoring yields visualizations where warmer color indicate higher-than-average levels of neural similarity for a given brain region, while cooler color suggests lower-than-average levels of neural similarity for a given brain region, which we believe makes the resultant figures easier for readers to quickly and easily interpret.

As suggested, we have added figures to the supplement that shows the “raw” ISCs for each group of dyads (Supplementary Figures 2 and 8). We hope that this helps to clarify that the cool values in Figures 3 and 5 do not reflect negative similarities and we thank the reviewer for this suggestion.

We also note that the analyses were also done on the ISCs that had been standardized within each region, and as the analyses were all done within parcels, this transformation should not systematically bias the results observed.

R3.5) How exactly were the fMRI subjects recruited? Were they randomly sampled, or did they self-select to participate in the fMRI study. It seems like they fairly uniformly sample the social network as measured at time 2 (but not so much at time 3). I think the design is safe from concerns like “the kind of people who sign up for an fMRI study make friends in a particular way”—but it might benefit the authors to explicitly preempt this kind of concern.

Thank you for raising this point. fMRI subjects were recruited by self-selection, which we have clarified in the revised Methods section, which now states, *“Participants were scanned as soon as possible after their arrival on campus and self-selected to participate in the study after receiving a recruitment email that was sent to all incoming students in the cohort.”* (p. 1 of the revised Methods).

As described in more detail in our next response (R3.6), we also note that although the social network visualization in Fig. 1 might give the appearance that nodes corresponding to fMRI participants were biased towards one side of the network in Time 3, there is considerable stochasticity involved in determining the position of nodes on the graph with the force-directed graph drawing algorithm used here. Thus, the location of particular nodes in the network varies substantially when the algorithm is initialized using different random seeds. To clarify this, we have added the following text to the Fig. 1 caption, *“The visualization for social networks at Time 2 and Time 3 were generated using the Large Graph Layout (LGL) algorithm²⁸ in igraph²⁹; note that there is some stochasticity in the determination of nodes’ positions within graphs using this algorithm.”* (p. 4 of the revised manuscript)

We acknowledge that it is still possible that “the kind of people who sign up for an fMRI study make friends in a particular way”. While we do not view this as a likely explanation for the observed results, we agree that this remains a possibility. Thus, we have added the following text to the revised Discussion section, *“...It is important to note that the current evidence for neural homophily is constrained within a community of graduate students studying business in a private university in a single cultural context. Further, participants self-selected to participate in the fMRI study and it is possible that systematic differences exist between people who do or do not elect to participate in such a study. Thus, future work examining these effects across diverse cultures and communities, ideally randomly sampling participants from within communities, will yield a more comprehensive understanding of how this phenomenon may differentially unfold across different social networks.”* (p. 22 of the revised manuscript)

R3.6) In the Figure 1 caption, can you briefly include a sentence describing how similarity is measured and what kind of dimensionality reduction is used to generate these visualizations of the social network structure? Any idea why the red fMRI subjects end up biased toward one side of the projection at time 3?

Thank you for your questions.

An edge connecting two nodes in the social network graphs indicates that there exists a mutually reported friendship, and we have included the following text in the Fig.1 caption (p. 4 of the revised manuscript) to clarify what the edges represent in the visualization: *“...each node represents a study participant and there exists an edge connecting two nodes if there is a mutually reported (i.e., reciprocal) friendship.”* We hope that this clarifies that edges in this network reflect mutually reported social ties rather than similarities. Because these are network visualizations (generated using a force-directed graph drawing algorithm, as described in more detail in the paragraph below), no dimensionality reduction was involved in their preparation. We hope that this clarifies that these graph visualizations differ, for example, from visualizations depicting inter-subject similarities, which would very likely involve dimension reduction of

some sort. We hope that more explicitly indicating what edges represent in the visualization, as well as how these images were created (see below), helps to clarify this distinction.

Visualization for social network graphs was generated using a force-directed graph drawing algorithm (i.e., large graph layout, LGL; Adai et al., 2004) in the *igraph* R package and there is no dimensionality reduction applied. We have added the following text in the Fig. 1 caption (p. 4 of the revised manuscript) to clarify how the visualization is generated: “The visualization for social networks at Time 2 and Time 3 were generated using the Large Graph Layout (LGL) algorithm in igraph”. The LGL algorithm, in which nodes are arranged on a two-dimensional space and edges act as springs to pull together identified sets of clustered nodes, is developed to visualize large-scale networks. There is some stochasticity in determining the exact positions of nodes in the graphs. Given that where the node ends up exactly in the graph is rather inconsistent (see below for examples of Time 3 social network visualization generated using LGL with different random seeds), it may be not appropriate to interpret the patterns of individual nodes’ positions. Therefore, as noted in our response to the previous comment (R3.5), we have also added the following text to the Fig. 1 caption: “...Note that there is some stochasticity in the determination of nodes’ positions within graphs using this algorithm” (p. 4 of the revised manuscript). We would be happy to add further discussions of these points if it would improve the clarity of this aspect of the manuscript.

R3.7) I think it's generally helpful to plot unthresholded brain maps so that readers can see effects across the whole brain. But I would suggest that the authors make an explicit note in figure captions presenting unthreshold maps (e.g. Figs. 3, 5, Supplementary, etc.) that no statistical threshold or correction for multiple tests is performed.

Thank you for pointing this out.

For figures that depict the results of statistical tests (Figs. 4 and 6, and Supplementary Figs. 4-7 and 9-11), we have added the underlined text to the relevant portion of the caption: *"The overlaid data is unthresholded; regions with significant differences, after FDR correction for multiple tests, are outlined in black."*

Figs. 3 and 5 and Supplementary Figs. 2, 3, and 8 show the mean relative neural similarities for each brain region for each group of dyads. Thus, these figures merely depict descriptive statistics (means) rather than the results of statistical tests. To clarify this, we have added the following to the corresponding captions: *"Because these images merely depict the relative mean similarities for each brain region for each group of dyads, rather than the results of statistical tests, no thresholding has been performed."*

R3.8) The authors are performing statistical tests on dyads, which necessarily introduces non-independence among samples: i.e. one subject contributes to multiple dyads and the number of dyads will yield inflated degrees of freedom (Chen et al., 2016; Nastase et al., 2019). Are the authors taking any statistical precautions to address this non-independence?

Thank you for your thoughtful comment. As addressed in our response to Reviewer #1 (i.e., R1.1), the particular permutation testing procedures used here yielded permuted datasets for the null distribution that preserved the same dependency structure as the original dataset, and thus accounted for the non-independent nature of samples in this dataset.

R3.9) At the beginning of the Results section, you mention the film clips very briefly in passing. It might be worth adding another sentence here to clarify if there was any particular rhyme or reason to the film clips selected by the experimenters. Similarly, is the content of these particular naturalistic stimuli doing any theoretical work here? Would you expect the same kind of results with resting-state connectivity?

Thank you for your thoughtful comment on the content of the movie stimuli. The following sentences have been added to the main text of the manuscript to describe the content of the movie clips that were selected and the reasons for their selection: “Movie clips spanning a wide range of styles (e.g., documentary, comedy, debate) and topics (e.g., food, science/technology, sports, environment, social events) were selected to fulfill several key criteria. Specifically, we sought clips that participants would be unlikely to have seen before, that would minimize mind-wandering by being sufficiently interesting to keep participants’ thoughts and attention focused on the stimuli, and that would elicit meaningful individual differences in neural responses, e.g., because different people might have different temporal trajectories of emotional reactions and/or attentional allocation, interpret differently, and/or attend to different aspects of the stimuli (see Methods for more details about stimuli selection).” (p. 5 of the revised manuscript)

In the Methods section, brief descriptions of why these movie clips were selected have been listed in the following paragraph (new text is underlined): “Criteria used to select stimuli are described in more detail in a previous manuscript that used the same stimuli in a different sample⁶⁰. Briefly, efforts were made to select stimuli that (1) most participants would not have seen before, (2) would be engaging for participants, and (3) would evoke diverging inferences and patterns of attentional allocation across viewers, and thus, psychologically meaningful variability in neural responding (e.g., because different people might attend to, emotionally react to, and/or interpret them differently). We selected stimuli that spanned a wide range of styles and topics with the hope of evoking a wide range of cognitive and affective processing that resembles the diverse range of stimulation and associated mental phenomena that arises in everyday life.” (p. 3 of the revised Methods section)

We used the stimulus selection criteria noted above because we reasoned that stimuli that fulfilled those criteria would provide the greatest sensitivity to observe our hypothesized effects. Thus, we speculate that the above-described characteristics of the selected stimuli are doing some of the work here, as the reviewer suggests might be the case. The current study’s design is not well-suited to testing which stimuli (or which aspects of stimuli) would evoke ISCs that are most closely associated with friendship and social distance changes over time (since all stimuli were presented in a fixed order to avoid inducing response variability between participants related to differences in how clips were presented). However, we view this as an important direction for future research and thus have added the following text to the Discussion section: “Additionally, we note that the current study’s stimuli were selected and presented in a manner that we hypothesized would afford sensitivity to detect our hypotheses, as described in the Methods section. It remains to be seen what kinds of stimuli and/or what characteristics of stimuli would evoke ISCs that would be most closely linked to future friendship formation. Similarly, it remains to be seen if similar results would be observed if inter-subject neural similarities were instead computed based on patterns of functional connectivity measured at rest or during naturalistic stimulation (which has recently been to be more predictive of behavior than rest⁵⁹). It would be valuable for future work to systematically

address these questions, as such work has the potential to inform both basic and applied research on predictors of social connection.” (p. 24 of the revised manuscript)

As the scope of this research is dependent on the participant’s engagement with external stimuli (i.e., resting state data was not collected), it is hard to ascertain whether resting-state connectivity would be similarly linked to future friendship formation and social distance changes. However, based on prior research that showed functional connectivity assessed during naturalistic viewing task outperformed resting-state connectivity when predicting cognitive and affective traits (Finn & Bandettini, 2021), we speculate that pre-existing similarity in functional connectivity observed during a movie-watching task such as that used here, and possibly during rest, may be able to predict future social distance or change in social distance. In the revised Discussion section, we point to this as an important direction for future research, as noted in the newly added text quoted in the paragraph above.

R3.10) In the Introduction, you mention that prior work relating inter-individual similarities in self-reported personality traits to social network structure “has yielded null or inconsistent results.” Do the authors have any further explanation as to why? For example, is it possible that personality research simply uses much larger samples with higher statistical standards (relative to neuroscience)? Or is there some mismatch between behavioral assays and the constructs they aim to capture (e.g. Eisenberg et al., 2019; Dang et al., 2020)?

Thank you for your thoughtful comment. It is possible that investigating similarities in fluctuations of brain activity over time to naturalistic stimuli may be able to capture some meaningful inter-individual similarities (e.g., similar ways of perceiving and understanding the world) beyond similarities in personality traits estimated based on relatively brief self-report measures, that influence subsequent friendship formation and thus social network structure. For example, in addition to capturing temporal trajectories of many different aspects of mental processing in parallel as they unfold over time, using ISCs of neural responses can circumvent the need to rely on self-report. This can be advantageous, for example, because people are often unable to accurately introspect about their own mental processes (Wilson & Nisbett, 1978) and lack conscious access to many aspects of their mental processing (Wilson, 2002), which could in some cases hinder the ability of self-report measures to capture psychological phenomena of interest. Relatedly, compared to self-report measures, neural measures would also likely be less affected (and potentially distorted) by participants trying to present themselves in socially desirable ways (King & Bruner, 2000). As you have suggested, it is also possible that failures of some behavioral measures to effectively capture their constructs of interest may also play a role. As this point is mentioned in the introduction of the manuscript as one motivation for the current study and is not the main focus of this work, we have tentatively made the choice to not include further discussion on this in the main manuscript, but we are open to add the discussion if you think that would make the paper more valuable.

R3.11) The authors have analyzed data from Ivy League business school students in prior studies. Obviously this sample is not guaranteed to be representative of the broader population. Do the authors have any empirical data—e.g. personality metrics—on whether this particular sample systematically deviates from the “general” population in any particular way?

Thank you for raising this important question about this sample. We acknowledge that this sample of Ivy League business school students may not be representative of the “general” population. The following text (underlined) was added in the Discussion section to address this limitation: *“However, it is important to note that the current evidence for neural homophily is constrained within a community of graduate students studying business in a private university in a single cultural context. Further, participants self-selected to participate in the fMRI study and it is possible that systematic differences exist between people who do or do not elect to participate in such a study. Thus, future work examining these effects across diverse cultures and communities, ideally randomly sampling participants from within communities, will yield a more comprehensive understanding of how this phenomenon may differentially unfold across different social networks.”* (p. 22 of the revised manuscript)

As part of another study, students in this cohort also completed the HEXACO-60, a 60-item self-reported personality scale (Ashton & Lee, 2009). The norms (i.e., means) observed in our sample for each of the six personality dimensions ($M_{\text{Honesty-Humility}} = 3.40$; $M_{\text{Emotionality}} = 3.06$; $M_{\text{Extraversion}} = 3.54$; $M_{\text{Agreeableness}} = 3.15$; $M_{\text{Conscientiousness}} = 3.58$; $M_{\text{Openness to Experience}} = 3.61$) were similar to the published norms for each dimension of the scale (see table from Ashton & Lee, 2009 below for the published norms of HEXACO-60 based on a college and community sample). Given that the empirical data on personality metrics are consistent with the published norms, it does not appear that our sample deviates systematically from the “general” population, at least in the dimensions of personality assessed by HEXACO-60. However, it would be fruitful if future research adopting a similar paradigm could be carried out in different communities to study whether the pattern of results observed here is generalizable, as we suggest in the revised Discussion section in the text quoted above.

TABLE 2.—Descriptive and internal consistency statistics for the HEXACO–60 scales in self-report.

	r	α	Women M (SD)	Men M (SD)	d _(W–M)
College sample					
Honesty-Humility	.27	.79	3.30 (0.66)	3.04 (0.71)	0.38
Emotionality	.27	.78	3.64 (0.55)	2.93 (0.61)	1.23
Extraversion	.29	.80	3.49 (0.62)	3.47 (0.63)	0.02
Agreeableness	.25	.77	3.10 (0.58)	3.19 (0.65)	–0.15
Conscientiousness	.26	.78	3.58 (0.59)	3.31 (0.62)	0.46
Openness to Experience	.26	.77	3.54 (0.64)	3.51 (0.68)	0.05
Community sample					
Honesty-Humility	.23	.74	3.98 (0.50)	3.76 (0.55)	0.41
Emotionality	.21	.73	3.37 (0.54)	2.87 (0.49)	0.96
Extraversion ^a	.28	.73	3.32 (0.65)	3.26 (0.59)	0.10
Agreeableness	.23	.75	3.38 (0.54)	3.23 (0.56)	0.28
Conscientiousness	.24	.76	3.73 (0.51)	3.73 (0.52)	0.00
Openness to Experience	.28	.80	3.59 (0.65)	3.62 (0.64)	–0.04

Note. W = women; M = Men. For college sample, $N_W = 645$, $N_M = 283$. For community sample, $N_W = 413$, $N_M = 321$.

^aExtraversion scores in the community sample are based on seven items only (see text for details); *r* is mean interitem correlation; *d* is difference (Cohen's *d*) between mean scores of men and women.

References

- Adai, A. T., Date, S. V., Wieland, S., & Marcotte, E. M. (2004). LGL: creating a map of protein function with an algorithm for visualizing very large biological networks. *Journal of Molecular Biology*, *340*(1), 179–190. <https://doi.org/10.1016/j.jmb.2004.04.047>
- Ashton, M. C., & Lee, K. (2009). The HEXACO-60: a short measure of the major dimensions of personality. *Journal of Personality Assessment*, *91*(4), 340–345. <https://doi.org/10.1080/00223890902935878>
- Baek, E. C., Porter, M. A., & Parkinson, C. (2021). Social network analysis for social neuroscientists. *Social Cognitive and Affective Neuroscience*, *16*(8), 883–901. <https://doi.org/10.1093/scan/nsaa069>
- Dijkstra, E. W. (1959). A note on two problems in connexion with graphs. *Numeriske Mathematik*, *1*, 269–271.
- Finn, E. S., & Bandettini, P. A. (2021). Movie-watching outperforms rest for functional connectivity-based prediction of behavior. *Neuroimage*, *235*, 117963. <https://doi.org/10.1016/j.neuroimage.2021.117963>
- Hobson, E. A., Silk, M. J., Fefferman, N. H., Larremore, D. B., Rombach, P., Shai, S., & Pinter-Wollman, N. (2021). A guide to choosing and implementing reference models for social network analysis. *Biological Reviews of the Cambridge Philosophical Society*, *96*(6), 2716–2734. <https://doi.org/10.1111/brv.12775>
- King, M. F. & Bruner, G. C. (2000) Social desirability bias: A neglected aspect of validity testing. *Psychology & Marketing*, *17*(2), 79–103.
- Krackhardt, D. (1988). Predicting with networks: Nonparametric multiple regression analysis of dyadic data. *Social Networks*, *10*(4), 359-381.
- Simpson, William B. "QAP: The Quadratic Assignment Procedure." Paper presented at the North American Stata Users' Group Meeting, March 20–21, 2001. <http://fmwww.bc.edu/RePEc/nasug2001/simpson.pdf>
- Wey, T. W., & Blumstein, D. T. (2010). Social cohesion in yellow-bellied marmots is established through age and kin structuring. *Animal Behaviour*, *79*(6), 1343-1352.
- Wilson, T. D. (2002). *Strangers to Ourselves: Discovering the Adaptive Unconscious*. Belknap Press, Cambridge, MA.
- Wilson, T. D. & Nisbett, R. R. E. (1978). The accuracy of verbal reports about the effects of stimuli on evaluations and behavior. *Social Psychology*, *41*, 118–131.

Response to Reviewers

Thank you for taking the time to review our manuscript and provide valuable feedback. In this document, we provide detailed responses to address all remaining concerns from the reviewers. For clarity and concision, we have enumerated reviewer concerns such that Reviewer #2's first point is labeled R2.1, Reviewer #2's second point is labeled R2.2, Reviewer #3's first point is labeled R3.1, and so on; we use these labels when referring to particular reviewer comments throughout the document. Our responses to reviewer concerns are provided in blue text and text quoted from the revised manuscript is provided in *italics* with the added text underlined and ~~deleted text~~ struck through.

Response to Comments from Reviewer 2:

I appreciate the authors' revisions and detailed comments regarding those revisions. This remains a fascinating piece of research. I identify the following issues as still needing attention:

Thank you for your encouragement on our research and we appreciate your time and effort in thoroughly reviewing our manuscript.

[R2.1]

Editor Comment 1):

Regarding issues of network autocorrelation and non-independence of measurements raised by other reviewers, I am satisfied by the authors' response, which appropriately references the social network research literature and particularly the QAP-style logic identified by the authors.

Thank you.

[R2.2]

Editor Comment 3) and R2.2):

Despite the general assurances of the authors, I remain skeptical. A participation rate of 100% by students is far outside the range of participation I am familiar with in wholly voluntary studies, and ethical considerations include implicit as well as explicit pressure. The authors acknowledge that they engaged in an offer of "personalized, educational feedback" to "encourage participation," and that (passively speaking) these students "were invited to participate in the online social network survey as an optional part of their coursework." I am curious regarding that nature of the offer of "personalized, educational feedback." Could that be explained, with inclusion of the text of that offer and a description of who would offer it? I am also curious about who (in the active voice) "invited" participation and using what additional signals of authority. In an MBA program, in which students measure success or failure not simply by the passing of classes but in no small part on the basis of recommendations from and the formation of social ties with faculty and administration, is it reasonable to expect that students might perceive implicit an obligation to engage in the "optional" work of supporting said faculty and administration? Were the names of participants and non-participants available to any faculty or administration members? Supplying the consent forms supplied to students and the text of the recruitment communications would help resolve skepticism on this point.

Thank you for raising this concern. To provide more context on the social network data, these data were collected as part of the course pedagogy on social networks, which are integral to success in business and, therefore, a part of the core course in organizational behavior. These social network survey data were collected for every cohort of this business school, and the current fMRI study recruited participants from a cohort in which everyone participated in the social network survey, but there have been students from other cohorts who have declined to complete the survey with no formal or informal consequences. The survey was set up by the course instructor as an optional class exercise and all data were collected using student ID numbers, which were later deidentified. Personalized feedback reports were provided by the instructor of the course for each student who completed the survey. As every email account at the university has one email address based on the student's name and another email address based on their student ID number, the personalized report was sent to the email address based on their ID number. The class later discussed the overall network structure of the business school community, to which everyone taking the course belonged and could relate, whether or not they had completed the survey.

The personalized feedback report started with the following paragraphs:

“Below, please find your personalized report on the structure of your network, how it has changed since the fall, and how it compares to those of your classmates. You'll also find a diagram of your immediate network – you and the people directly tied to you.

As you interpret these results, please remember that there are no answers that are either good or bad. Your network is a means to achieving your social and professional goals, so the ideal network for you is whatever network will help you to do that. While these data describe your socializing network among your classmates, research shows that people tend to exhibit the same network-building tendencies across domains – so your professional network may bear some similarity in structure to your socializing network.

The purpose of this exercise is to give you feedback that will help you to treat your social capital as a resource worth investing in, as discussed in Personal Leadership. If you look at these results and find them to be consistent with your prior beliefs, then you have a realistic sense of your network. If, however, you find one or more of the results surprising, it may be worth thinking about how you might work more effectively to build the network you want to have.”

Following the introduction, the report included definitions of key network characteristics (e.g., size, churn, diversity, and brokerage), the student's personal network characteristics as well as how their metrics compared to the average of their cohort, and a network diagram. Anecdotally, students have found the information in the report interesting and valuable. In addition, the data was collected in a small business school located in a remote part of New Hampshire, and the culture of the school is known to be extremely friendly and collegial, which may in part contribute to the high response rate.

Some additional evidence may help resolve the skepticism around ethical considerations. First, it is very uncommon for faculty members to write recommendations for MBA students, as such letters are not prioritized by employers in their evaluations for potential hires. Further, a grade non-disclosure policy, which forbids students from disclosing their grades to prospective employers under penalty of the school's Honor Code, was implemented to promote collaboration among students. As a result, students at this institution have little incentive to be concerned about their grades. This policy is well-established at this business school and several peer institutions' business schools such that students and employers are

aware of the policy and understand the implication behind the policy that a student's success in the program is not measured by their grades.

We would like to reiterate that the experimental procedures were in compliance with the guidelines set by the Internal Review Board of the institution, and we hope that this detailed description of recruitment and data collection can mitigate the concerns surrounding ethical considerations.

[R2.3]

Editor Comment 5) and R2.1):

Although the word "may" has been added, the words "determines," "fosters," "induces," "causes", "produce," "especially large impact," "facilitate," "conducive to," "due to," and "shaping" are still present in the manuscript. Lines 461-463 involve the claim that a result "sheds light" on a "causal relationship." This is all causal language, and while the word "may" is added the implied message is that there's something causal going on here. There is absolutely no "may" about the title, which asserts openly that "Neural Similarity Induces Friendship." I am not convinced that the authors have thoroughly followed the editors' request to "remove causal claims."

I am glad, to see that the authors have indicated in the discussion, without using the word "spurious," that there's quite a possibility that "Neural Similarity does NOT Induce Friendship," and that neural similarity might be epiphenomenal, with social forces playing the shaping roles. Perhaps neural similarity does induce friendship. Perhaps neural similarity is a conduit but not independent. The best title here, given the uncertainty, would be "Neural Similarity Might or Might Not Induce Friendship" or perhaps "Social and Neural Similarity Precede Friendship."

Thank you for raising this point. We have revisited the uses of such language in the manuscript and made several revisions, as described below. We would like to clarify that in some cases, these words were used with the intention of explaining potential mechanisms underlying the observed effects, while making clear that these were speculative based on the results from the current study or prior research by using them after "may". Therefore, we did not intend to imply that the current findings demonstrate the existence of causal relationships. Below, we carefully review each case of these above-mentioned words in the previous version of the manuscript:

- We have modified the title of the manuscript from "Neural Similarity **Induces** Friendship" to "Neural Similarity Predicts Whether Strangers Become Friends" to remove the causal claim inherent in the previous title.
- "What *determines* who becomes and stays friends?" (p. 2 of the revised manuscript): The word "*determines*" here is used to introduce a broad question that motivates the current study and is followed by the sentence "Many factors are linked to friendship, including physical proximity and interpersonal similarities". Therefore, there is no implication of a causal relationship between pre-existing neural similarity and the observed social network phenomena.
- "However, given the cross-sectional nature of past research, it is unknown if neural similarity is the *cause* of friendship or its consequence" (p. 2 of the manuscript): Because here the word "cause" was used in the context of stating that it is unknown whether or not neural similarity causes friendship, we had left it in the previous draft. However, we have now changed this

sentence to “*However, given the cross-sectional nature of past research, it is unknown if neural similarity ~~is the cause of friendship or its consequence~~ precedes friendship or only emerges among friends following social connection” in order to more directly lay the groundwork for the scope of the research question that the current investigation can decisively answer.*

- “*Although this handful of studies demonstrate the utility of neuroimaging in probing the types of similarities that are shared among friends, their cross-sectional nature limits the types of inferences that can be made about **a causal relationship** between neural similarity and social network proximity,*” (pp. 3-4 of the manuscript): To avoid inducing an expectation in readers that the current study’s results will provide decisive evidence for a causal relationship, we have revised this sentence to remove the phrase “*a causal relationship.*” This sentence now reads, “*Although this handful of studies demonstrate the utility of neuroimaging in probing the types of similarities that are shared among friends, their cross-sectional nature limits the types of inferences that can be made about ~~a causal~~the relationship between neural similarity and social network proximity.*” (pp. 3-4 of the manuscript)
- “*The localization of these results suggests that pre-existing similarities in how people interpret, attend to, and emotionally respond to their surroundings **facilitate friendship**. These results provide evidence for neural homophily, such that pre-existing neural similarities **may foster** friendship and increased social closeness.*” (p. 2 of the revised manuscript): We have revised and consolidated these two sentences to remove causal claims. This section now simply reads: “*The localization of these results suggests that pre-existing similarities in how people interpret, attend to, and emotionally respond to their surroundings ~~facilitate friendship. These results provide evidence for neural homophily, such that pre-existing neural similarities may foster friendship~~ and are precursors of future friendship and increased social closeness.”*
- “*Such sociobehavioral similarities **can also foster** the formation of affiliative ties by facilitating interpersonal communication and predictability⁵¹⁻⁵⁴.*” (p. 21 of the revised manuscript): the word “*foster*” is used here to describe the potential role of sociobehavioral similarities in the formation of affiliative ties and is followed by citations supporting these claims. Therefore, we have left this use of the word “*foster*” in the manuscript.
- “*Future work would benefit from investigating the role of social influence, as it is likely that homophily and social influence processes interact in human social networks. For example, pre-existing similarities in how individuals think about and respond to the world around them **may, in part, cause** individuals to become friends,~~as suggested by the current results.~~”* (p. 22 of the revised manuscript): the word “*cause*” is used here to explain the concept of homophily in general, including as it pertains to similarities in demographic variables and neural processing, both of which relate to how people interpret their experiences and behave. Therefore, we have left this use of the word “*cause*” in the manuscript. However, we have removed “*as suggested by the current results*” to avoid implying that the current results suggest that the current results provide evidence for a causal relationship.
- “*Our results are consistent with a causal model in which a wide array of sociodemographic variables and other factors, including, but not limited to, those that we were able to measure here (such as geographic proximity, race, nationality, gender, birth cohort, religion, linguistic background, educational background, parenting, political beliefs, genes, gene-environment interactions, and so on) give rise to sets of experiences, expectations, and social feedback that in turn give rise to particular ways of responding to the world, which are reflected in neural activity*

*and when similar, **facilitate** future friendship.*” (p. 24 of the revised manuscript): In this instance, the word “*facilitate*” is used in the context of suggesting that similarities in a broad array of socio-demographic variables could lead to friendship by way of inducing similarities in people’s experiences, expectations, and the social feedback they receive, which also become reflected in neural data. We believe the phrase “*Our results are consistent with*” makes clear that this is one possibility with which our results are consistent (but of course, not the only possible explanation for the phenomena observed here). Therefore, we did not think that this use of the phrase “*facilitate*” was problematic and have left it in the paper. That said, we would be happy to revise this text if the Editor and/or Reviewer disagree.

- “*More broadly, despite the substantial degree of random assignment to which participants were subjected regarding with whom and where they lived, and regarding with whom they studied and took classes (see Methods), we note that this study is inherently limited in its capacity to support decisive causal claims, given that it was an observational study. Thus, while the current findings are consistent with the causal model posited above, in which innumerable demographic and biological factors give rise to neural similarities that in turn **facilitate** friendship, more research is needed to better understand how relationships among these phenomena come to be, as well as potential moderating factors.*” (pp. 24-25 of the revised manuscript): We have revised the last sentence of this excerpt to read, “*Thus, while the current findings are consistent with the causal model posited above, in which innumerable demographic and biological factors give rise to shared ways of responding to the world that are reflected in neural similarities and that in turn facilitate friendship, more research is needed to better understand how relationships among these phenomena come to be, as well as potential moderating factors, and to more directly test for such a causal model.*” (pp. 24-25 of the revised manuscript) We hope that these changes, together with the context provided by the preceding sentence, provide greater consistency with preceding portions of the Discussion section, no longer suggest a causal relationship between neural similarity and friendship, and make clear that more work is needed before causal claims can be made.
- “*Taken together with the current results, this growing body of research continues to demonstrate that these inter-individual similarities may **facilitate** social connection and the formation of social ties.*” (p. 21 of the revised manuscript): To clarify that inter-individual similarities facilitating social connection are one possibility (but not the only possibility) for explaining the current results, we have replaced “*continues to demonstrate*” with “*is consistent with the possibility*”. This excerpt now reads: “*Taken together with the current results, this growing body of research ~~continues to demonstrate~~ is consistent with the possibility that these inter-individual similarities may facilitate social connection and the formation of social ties.*” (p. 21 of the revised manuscript)
- “*Additionally, variability in these pre-existing neural similarities may have also **facilitated** the formation, persistence, and dissolution of direct social ties between the two-month mark and the eight-month mark after individuals entered their new community*” (p. 22 of the revised manuscript): In this case, we preceded “*facilitated*” with “*may also have*” with the intention of making clear that this is an instance of speculation and that we are not claiming that the current findings provide decisive evidence supporting causal claims. We also note that we have now deleted “*neural*” from this sentence (it now reads, “*Additionally, variability in these pre-existing ~~neural~~ similarities may have also facilitated the formation, persistence, and dissolution of direct*

social ties between the two-month mark and the eight-month mark after individuals entered their new community") since the similarities discussed in the preceding paragraph, to which this sentence refers, encompass not only neural similarities but sociobehavioral similarities that other work has linked to neural similarities. We have also amended the subsequent sentences in this paragraph to describe both neural and social similarities (previously, only neural similarities had been mentioned) to make clear that the speculation later in this paragraph is not specific to neural similarities.

- *"This is in line with prior work demonstrating that similarity in neural responding in both the DMN and FPCN is associated with similarity in subjective understanding of a complex narrative⁶. Such similarities in subjective construal of narratives **may be particularly conducive to** friendship formation, as they may reflect, more generally, alignment in how individuals make sense of the world around them."* (pp. 19-20 of the revised manuscript): In this case, the word "conductive" is used in the context of pointing to past work linking interpersonal neural similarity in particular brain networks to interpersonal similarity in aspects of mental processing, and speculating about a potential role for such similarities in friendship formation. We think that this particular phrasing, as well as its broader context in the Discussion section, makes clear that this is mere speculation. However, we would be happy to revise this phrasing if the Editor and/or Reviewer disagree.
- *"Indeed, separate work has shown that a sense of "generalized shared reality" (i.e., similarities in feelings, beliefs, and concerns about the world in general) is linked to social connection and interpersonal liking and thus **may be conducive to** friendship formation."* (p. 21 of the revised manuscript): This use of the phrase "conductive to" introduces prior work by other authors that has investigated generalized shared reality using behavioral studies and suggested that generalized shared reality may play an important role in fostering social connection. Because here we are using "conductive" within a description of findings and interpretations from other authors' cited research, rather than the current findings, the word "conductive" in this instance does not imply any causal relationships between neural similarity and friendship formation. Therefore, we have left this use of the word "conductive" in the manuscript.
- *"As a result, given sufficient time, pairs of individuals may have formed and/or maintained friendships **due to** interpersonal compatibilities reflected in pre-existing neural similarity. On the other hand, friendships born out of mere circumstance may have dissolved **due to** interpersonal incompatibilities, reflected in pre-existing neural dissimilarities, which only became apparent with the extended passage of time."* (p. 22 of the revised manuscript): the use of "due to" here suggests interpersonal compatibilities may underlie friendship formation/maintenance or dissolution. While we suggest that these broader compatibilities are "reflected in" neural similarity, we do not suggest that neural similarity itself causes the formation, dissolution, or maintenance of friendships. Therefore, we have tentatively left this largely as-is, but have added "and social" ("*However, between the two-month mark and the eight-month mark, neural and social homophily processes may have had more time to unfold. As a result, given sufficient time, pairs of individuals may have formed and/or maintained friendships due to interpersonal compatibilities reflected in pre-existing neural and social similarity. On the other hand, friendships born out of mere circumstance may have dissolved due to interpersonal incompatibilities, reflected in pre-existing neural and social dissimilarities, which only became*

apparent with the extended passage of time.”) to convey that incompatibilities are reflected not only in neural dissimilarities but also in socio-demographic data.

- “*The current research expands upon [past cross-sectional] work and sheds light on the potential role of neural homophily in **shaping** human social structures over time.*” (p. 26 of the revised manuscript): “*shaping*” here is used in conjunction with “*sheds light on the potential role of*”. Therefore, we think that it is clear that this is a speculation, rather than a claim that the current research demonstrates that neural similarity causes social connection. As such, we have tentatively left this sentence as-is. However, we would be happy to revise this phrasing if the Editor and/or Reviewer disagree.
- We could not find instances of using the words “produce”, “especially large impact”, or the example given that “a result ‘sheds light’ on a ‘causal relationship’” in the previously revised manuscript, so we would like to note that these sentences may have already been revised in the previous round of revision.

[R2.4]

Particularly regarding living proximity in R2.1), that “nearly all students applied to on-campus housing and were randomly assigned to housing units” might lessen demographic homophily in housing, but should not be expected to have eliminated it. Sorting toward similarity is a well-known first-year student phenomenon. I’ll quote Bahns et al 2013:

“Research supports both individual-level and relationship-level predictors of roommate satisfaction and breakup. Similarity—an emergent property of the roommate dyad—has been repeatedly shown to affect the decision to stay with or leave a roommate, including matching on personality (Carli, Ganley, & Pierce-Otay, 1991; Heckert et al., 1999; cf. Lapidus, Green, & Baruh, 1985), values (Jones, McCaa, & Martecchini, 1980; cf. Lapidus et al., 1985), sleeping habits, study habits, and neatness (Fuller & Hall, 1996; Jones et al., 1980; Lapidus et al., 1985), as well as shared activities (Lovejoy, Perkins, & Collins, 1995) and same-race compared to mixed-race roommates (Shook & Fazio, 2008).”

In contrast, it is helpful to know that students were randomly assigned to study groups for an entire term.

Thank you for pointing us to this relevant research. We have revised the relevant portion of the Methods section to point to some of this research and to acknowledge that even if housing units were randomly assigned, homophily with respect to demographic variables or other factors could still unfold:

“Additionally, nearly all students applied to on-campus housing and were randomly assigned to housing units based on a lottery system. (That said, we note that even if housing units and roommates were initially randomly assigned, it is possible that they would not have stayed that way, given past research demonstrating both individual- and dyad-level predictors of roommate satisfaction and breakup, including various facets of interpersonal similarity.)^{60,61}” (p. 2 of the revised Methods section)

[R2.5]

The authors write in the revision that “to statistically account for demographic similarities that may be related to similarities in neural responding and/or social network proximity, we regressed out inter-subject

similarities in age, gender and nationality from ISCs in each brain region...", with what appears based on my reading to be a comparison between a baseline model without the three demographic controls (with the difference between those growing closer and those growing apart as delta-ISC in Supplementary Table 2) and a model that includes controls for age similarity, gender similarity, and nationality similarity (showing as delta-ISC in Supplementary Table 3).

It's worth noting that the authors statistically very partially accounts for demographic similarities, considering that it leaves out a large number of social dimensions that if other research bears out should expect to exhibit homophily: region/geography, economic class, legacy status, disability status, religion, race, ethnicity, political party, first language, major, undergraduate alma mater, and immediately prior occupation.

What trend in the data does the inclusion of these three variables provoke? If my interpretation accurately reflects the meaning of Supplementary Tables 2 and 3, it appears that the difference in ISC between those dyads growing closer versus those dyads growing apart increases in 11 brain parcels when controlling for the 3 demographic dimensions, stays the same in 1 brain parcel, and decreases in 30 brain parcels.

Would the authors concur in that interpretation of results? If so, this at least suggests that a portion of differences between the two sets of dyads might be explained by the meager 3 controls for similarity, and that if we included more dimensions (many among those I've listed above exhibiting quite strong homophily in replicated research), the differences might further erode. There's a whole lot of unobserved heterogeneity here, regarding variables known to have a very strong effect on both networks and neuron-shifting culture, and that lessens my confidence that we're seeing nonspurious neuronal similarity as an independent dimension of homophily. Without full controls of meaningful sociological variables, this isn't a slam dunk and maybe not even a basket. While intriguing, it looks like some more controls are needed. Is there a way to find at least some more of these demographic variables in student records? Speaking hopefully, the authors' consent forms might have already given permission to access the school's records to obtain individual-level variables.

We apologize for not having presented the results clearly and we appreciate the opportunity to further clarify them here and in the revised manuscript. When controlling for three demographic variables (i.e., age, gender, and nationality), we observed similar patterns of results to those that were obtained without controlling for those variables. Specifically, in the first set of analyses presented in the main text (Fig. 4), the same brain region emerges as significant with (Fig. 4) and without (Supplementary Fig. 5c in the prior version of the manuscript) controlling for those demographic variables. In the second set of analyses presented in the main text (Fig. 6), when we repeated our analyses controlling for demographic variables (Supplementary Fig. 9c in the prior version of the manuscript), of the 43 brain regions that had been significant in our main analyses, only one brain region was no longer significant after controlling for demographic variables, and 9 additional regions became significant after controlling for demographic variables (for a total of 52 brain regions showing a significant effect when controlling for demographic variables shown in Table S3, as compared to the original 43 shown in Table S2). Thus, controlling for these 3 demographic variables resulted in a net increase in the number of brain regions that showed significant effects. We would caution against simply comparing the magnitudes of Δ_{ISC} or p -values in corresponding brain regions between Table S2 and Table S3, as doing so cannot provide evidence for or

against a significant difference between the results presented in the two tables, and does not suggest a lack of significant results when controlling for these demographic variables.

We acknowledge that our initial attempt to control for sociodemographic variables only incorporated a small number of these variables. Following the list you provided, we repeated the analyses when controlling for a wider range of sociodemographic variables that we were later able to obtain, including intersubject similarities in their age, gender, nationality, hometown (both in terms of location and population size), undergraduate alma mater (both in terms of private versus public institution and location), undergraduate major, and industry of prior employment. When controlling for all of these variables, the significant effect in the left OFC shown in our first set of analyses, comparing pre-existing neural similarity between friends and dyads of a social distance of 3, was no longer significant after FDR correction across 214 brain regions (95% bootstrap CI: [0.162, 0.909], $p = 0.003$, FDR-corrected $p = 0.589$). However, in the second set of analyses comparing the neural similarity of dyads that grew closer versus apart, we observed a similar pattern of results to what we had previously observed when controlling for this more extensive set of demographic variables. Specifically, of the 43 brain regions that had been significant in our main analyses, 37 remained significant with 2 additional regions becoming significant after controlling for demographic variables (for a total of 39 brain regions showing a significant effect when controlling for this more extensive set of nine sociodemographic variables). We included these results in Table S3, replacing the table of results from the previous analyses which only controlled for age, gender, and nationality (pp. 18-19 of the revised supplement).

In summary, the results from the analyses conducted when controlling for a more extensive set of sociodemographic variables showed that these variables accounted for links between neural similarity and social network phenomena observed for the analysis contrasting the neural similarity between future friends versus future dyads of a social distance of 3, but not for the more extensive set of results found in the analysis comparing dyads that grew closer together to dyads that grew further apart over time. This may in part be due to the range of phenomena these two sets of analyses may have been able to capture, which is related to the reason why we speculate that a more extensive set of regions was implicated for the second set of analyses comparing dyads who grew apart versus those that grew closer over time, as mentioned in our response to Reviewer 1 in the prior response letter (i.e., R1.6). The relevant portion of the previous response is quoted here for your convenience:

“For instance, consider the possibility that there is a general tendency for more similar people to later become friends, but sometimes various factors disrupt the formation of friendships between similar individuals (e.g., not encountering one another sufficiently frequently during the 8 months of the study to become friends, limitations on one or both individuals’ available time or general interest in socializing, other factors that might lead to incompatibility). If, for example, Persons A, B, and C were all quite similar to one another before entering the academic program, but one or more of the factors listed above (or other factors) led Person A and Person B *not* to become friends, despite their similarity, Persons A and B would still be relatively likely to become friends with other similar people, such as Person C (who is similar to both Person A and Person B). This could lead Person A and Person B, who in this example would have high pre-existing neural similarity, to decrease in social distance over time (e.g., going from a social distance of 3 or more at Time 2 to a social distance of 2 at Time 3). Our second set of analyses would be able to capture this, as it would present as a relationship between pre-existing neural similarity

and future decreases in social distance between Persons A and B, even if Person C, who indirectly connects Persons A and B at Time 3 in this example, were among the 245 students who were in the social network sample but did not participate in the fMRI study. However, our first set of analyses would not be able to capture this kind of phenomenon.

Along the same lines, consider another scenario where two fMRI participants, Persons D and E, were indirectly connected by a mutual friendship with Person F (who was also among the 245 students in the social network sample but not the fMRI subsample) at Time 2, but either one or both of Persons D and E's friendship(s) with Person F were merely borne out of circumstance and dissolved by Time 3 due to incompatibilities that only became apparent with time, reflected in pre-existing neural dissimilarities between Person F and Persons D and/or E. Again, our second set of analyses, but not our first, would be sensitive to this kind of phenomenon, as it would contribute to the association between social distance change and pre-existing neural similarity that we observe in our second set of analyses: Persons D and E would have low pre-existing neural similarity and would increase in social distance over time (in contrast to Persons A and B in the previous example, who would have high pre-existing neural similarity and would decrease in social distance over time). However, our first set of analyses, which focus on how friends within the neuroimaging sample compare to other dyads within the neuroimaging subsample, would not be sensitive to this sort of phenomenon."

To further probe whether this set of demographic variables resulted in a significant reduction of the difference in neural similarity between groups (i.e., dyads of future social distance of 1 versus 3) in the parcel where we observed a significant effect without controlling for any demographic variables, and if so, what demographic variables significantly accounted for this difference, we conducted analogous analyses as described on pp. 8-10 of the Methods section for this more extensive set of sociodemographic variables and then each of the nine sociodemographic variables. We found that (1) controlling for this set of demographic variables resulted in a significant reduction of the neural similarity differences between groups in the OFC parcel, and (2) out of the nine sociodemographic variables we now controlled for, only gender similarity resulted in a significant reduction in the difference in neural similarity.

To incorporate these new findings when controlling for a more extensive set of sociodemographic variables, we now make the following changes to the manuscript:

- We added a sentence to the abstract to highlight these findings: "*In addition, analyses controlling for sociodemographic similarities showed that whereas these similarities appeared to drive the differences in pre-existing neural similarities between friends and dyads of a social distance of 3, they did not account for the more extensive links between pre-existing neural similarities and the tendency for people to grow closer together, rather than drift farther apart, over time.*" (p. 2 of the revised manuscript).
- We have updated our descriptions of the relevant analyses in the Methods section:
 - "*We repeated these analyses while controlling for inter-subject similarities in the sociodemographic variables of age, gender, ~~and~~ nationality, hometown (both in terms of its location and population), undergraduate alma mater (both in terms of whether they attended a private or public institution and the institution's location), undergraduate major, and the industry in which they had been employed before enrolling in the MBA program. More specifically, to statistically account for demographic similarities that may*

be related to similarities in neural responding and/or social network proximity, we regressed out inter-subject similarities in the aforementioned sociodemographic variables from ISCs in each brain region before repeating the analyses described above.” (p. 10 of the revised Methods section)

- *“This analytical procedure was repeated using self-reported ratings of interest in the stimuli to test if inter-individual similarity in self-reported ratings of interest in the stimuli accounted for significant portions of the differences in neural similarity between groups. This analytical procedure was also repeated using demographic variables to probe if inter-individual similarity in demographic variables significantly accounted for the differences in pre-existing neural similarity between groups.”* (p. 11 of the revised Methods section)
- *“We repeated these analyses while controlling for inter-subject similarities in the demographic variables of age, gender, nationality, hometown (both in terms of its location and population), undergraduate alma mater (both in terms of whether they attended a private or public institution and the institution’s location), undergraduate major, and the industry in which they had been employed before enrolling in the MBA program. More specifically, to statistically account for sociodemographic similarities that may be related to similarities in neural responding and/or social network proximity, we regressed out inter-subject similarities in the aforementioned variables from ISCs in each brain region before repeating the analyses described above.”* (pp. 13-14 of the revised Methods section)
- We add a subsection titled “Relationships between neural similarity and future social closeness were partially driven by similarity in sociodemographic variables” to discuss these new findings when controlling for a more extensive set of sociodemographic variables (pp. 14-15 of the revised manuscript):

“Given that homophily based on sociodemographic variables is widely observed in social networks, we ran analyses to test if the relationships observed between pre-existing neural similarity and social network phenomena were accounted for by intersubject similarities in sociodemographic variables (including inter-individual similarities in their age, gender, nationality, hometown size and location, undergraduate alma mater location and institution type, undergraduate major, and industry). When controlling for these demographic variables, the significant effect in the left portion of OFC shown in our first set of analyses, comparing pre-existing neural similarity between friends and dyads of a social distance of 3, was no longer significant after FDR correction across 214 brain regions (95% bootstrap CI: [0.162, 0.909], $p = 0.003$, FDR-corrected $p = 0.589$); however, we observed a similar pattern of results with and without controlling for these demographic variables in the second set of analyses comparing the neural similarity of dyads that grew closer versus apart (see Supplementary Fig. 8 and Table S3). To further probe whether accounting for this extensive set of sociodemographic variables significantly reduced the difference in neural similarity between future friends and future friends-of-friends-of-friends, and if so, what sociodemographic variable(s) drove the significant reduction, we conducted analyses analogous to those that we performed to test if accounting for inter-individual similarities in the extent to which participants found the stimuli enjoyable or interesting significantly diminished links between the neural and social network data. We found that controlling for these demographic variables resulted in a significant reduction of the neural

similarity differences between friends and dyads of a social distance of 3 in the aforementioned OFC parcel (difference in mean normalized neural similarity = 0.129; $p < 0.05$), and that only gender similarity resulted in a significant reduction in the difference in neural similarity between groups (difference in mean normalized neural similarity = 0.019; $p < 0.05$).” (pp. 14-15 of the revised manuscript)

- We revised the following paragraph in the Discussion section to incorporate these results: “*Given the role of OFC in processing of subjective value, neural similarity in this brain region may reflect similarities in tastes and preferences (e.g., similarities in what individuals find funny or otherwise appealing), which may become aligned across certain individuals as a function of sociodemographic similarities. Inter-individual similarity in sociodemographic variables (i.e., age, gender, nationality, hometown size and location, undergraduate alma mater, undergraduate major, and industry) appeared to drive the differences in pre-existing neural similarity in the OFC that were observed between friends and friends-of-friends-of-friends. In particular, gender similarity accounted for a significant reduction in differences in pre-existing neural similarity between friends and distance 3 dyads in this brain region. On the other hand, controlling for inter-individual similarity in the extent to which individuals were interested in the stimuli did not significantly diminish the difference in pre-existing neural similarity between friends and individuals characterized by a social distance of 3.*” (p. 17 of the revised manuscript)

More broadly, we note that our brains are profoundly shaped by a wide range of life experiences, social contexts, and individual traits—many of which are captured by, or correlated with, these sociodemographic variables. The fact that brain patterns predict future friendship highlights the profound role that similar experiences (e.g., cohort effects) and perspectives likely play in fostering social connections. Rather than undermining the importance of the neural similarity-friendship link, we believe that the updated results underscore how our brains are shaped by these variables (among many others) such that we can use neural activity alone—as a powerful, integrative marker that reflects the convergence of diverse personal and environmental factors—to predict who people will connect with. Rather than diminishing the importance of our findings, this highlights that friendship is likely determined in part by shared demographics and other influences that shape how people see the world, and that neural signatures of such phenomena predict future friendship.

We attempt to make these points on p. 24 of the Discussion section, and we would be happy to make further revisions to the text if it would be helpful. We include an excerpt here for convenience:

“Relatedly, although we attempted to statistically account for intersubject similarities in some sociodemographic variables in our analyses, it is likely that some unmeasured sociodemographic variables (particularly those related to similar cultural perspectives) causally contribute to both pre-existing neural similarities between participants and subsequent friendship formation. Our results are consistent with a causal model in which a wide array of sociodemographic variables and other factors, including, but not limited to, those that we were able to measure here (such as geographic proximity, race, nationality, gender, birth cohort, religion, linguistic background, educational background, parenting, political beliefs, genes, gene-environment interactions, and so on) give rise to sets of experiences, expectations, and social feedback that in turn give rise to particular ways of responding to the world, which are reflected in neural activity and when similar, facilitate future friendship.”

Of note, the current results demonstrate that even without knowing or measuring all of these possibly innumerable sociodemographic variables and other factors, it is possible to measure one of their likely outcomes (neural similarity), which predicts future friendship... ” (p. 24 or the revised manuscript)

These are the remaining concerns I have with the article, and I wish to reiterate my appreciation of the cleverness of this method in uncovering physical markers at least correlated with social outcomes.

Thank you again for your thoughtful comments that have helped to improve our manuscript.

Sincerely,

James Cook, Associate Professor of Sociology, University of Maine at Augusta

Response to Comments from Reviewer 3:

I think the authors have adequately addressed all of my comments and the manuscript is considerably improved. I appreciate their thorough answers to the other reviewers’ very thoughtful comments as well. In particular, I think the supplementary statistical analyses and the more measured language around causality are important improvements. Two minor notes from my final read-through:

Thank you again for your thoughtful comments to help us improve the manuscript.

[R3.1] Missing a word here: “which has recently been [shown?] to be more predictive of behavior than rest”

Thank you for pointing this out. We have added the missing word to the sentence: “*Similarly, it remains to be seen if similar results would be observed if inter-subject neural similarities were instead computed based on patterns of functional connectivity measured at rest or during naturalistic stimulation (which has recently been shown to be more predictive of behavior than rest⁵⁹).*”

[R3.2] Cosmetic note: In Figures S13 and S15, due to the large number of parcels in DMN, the purple color from the legend basically looks black; maybe you could just remove the black edges around each bar?

Thank you for your suggestion. We have made the black edges around each bar thinner to make the figures more readable. These figures (now Figures S12 and S14) are provided below for convenience:

Supplementary Figure 12. Dyads that grew closer, relative to dyads that grew apart, tended to show higher neural similarity across brain regions spanning multiple brain networks. Regression coefficients for each brain parcel from the linear mixed model and planned contrast analysis for $ISC_{\text{dyads that grew closer}}$ vs $ISC_{\text{dyads that grew apart}}$, grouped by their corresponding brain networks, are shown.

Supplementary Figure 14. Pre-existing neural similarity across brain regions spanning multiple brain networks showed a trending negative association with change in social distance over time. Regression coefficients for each brain parcel from the linear mixed model examining the relationship between pre-existing neural similarity and continuous change in social distance, grouped by their corresponding brain networks, are shown.